# Representational integration and differentiation in the human hippocampus following goal-directed navigation

**Corey Fernandez[1,2]\*, Jiefeng Jiang[3], Shao-Fang Wang[4], Hannah Lee Choi[4], Anthony D Wagner[2,4]**

[1]Graduate Program in Neurosciences, Stanford University, Stanford, United States; [2]Wu Tsai Neurosciences Institute, Stanford University, Stanford, United States; [3]Department of Psychological and Brain Sciences, University of Iowa, Iowa City, United States; [4]Department of Psychology, Stanford University, Stanford, United States

**Abstract** As we learn, dynamic memory processes build structured knowledge across our experiences. Such knowledge enables the formation of internal models of the world that we use to plan, make decisions, and act. Recent theorizing posits that mnemonic mechanisms of differentiation and integration – which at one level may seem to be at odds – both contribute to the emergence of structured knowledge. We tested this possibility using fMRI as human participants learned to navigate within local and global virtual environments over the course of 3 days. Pattern similarity analyses on entorhinal cortical and hippocampal patterns revealed evidence that differentiation and integration work concurrently to build local and global environmental representations, and that variability in integration relates to differences in navigation efficiency. These results offer new insights into the neural machinery and the underlying mechanisms that translate experiences into structured knowledge that allows us to navigate to achieve goals.

\*For correspondence:
coreyf@stanford.edu

**Competing interest:** The authors declare that no competing interests exist.

## Editor's evaluation

This is a carefully designed and analysed fMRI study investigating how neural representations in the hippocampus, entorhinal cortex, and ventromedial prefrontal cortex change as a function of spatial learning. These important and compelling results provide new insight into how local and global knowledge about our environment is represented. It will be of great interest to researchers studying the differentiation and integration of memories and the formation of cognitive maps.

## Introduction

Memory is central to who we are, providing the foundation for our sense of self, understanding of the world, and predictions about the future. While we experience life as a series of events, our ability to build knowledge across events – to abstract common themes and infer relationships between events – enables us to construct internal models of the world, or 'cognitive maps', that we use to plan, make decisions, and act (**Tolman, 1948**; **O'Keefe and Nadel, 1978**; **McClelland et al., 1995**; **Epstein et al., 2017**). One goal of memory science is to understand how neural systems integrate information across experiences, building structured knowledge about the world.

Interactions between a network of brain regions in the medial temporal lobe and neocortex are known to support episodic memory, or memory for events. Central to this network is the hippocampus, whose computations allow us to encode and recall specific episodes from the past and to abstract

relationships across experiences that share common elements (*McClelland et al., 1995*; *Eichenbaum, 2004*). In subserving rapid learning over co-occurring features, the hippocampus plays a critical role in building episodic memories and structured knowledge by binding together inputs from distributed cortical regions, forming conjunctive representations of experiences (*McClelland et al., 1995*; *Teyler and DiScenna, 1986*; *Libby et al., 2014*). As we learn, neural populations in the hippocampus that represent the traces for individual events can vary in their relations, becoming more distinct (differentiation) or more similar (integration) to each other (*Ritvo et al., 2019*; *Brunec et al., 2020*; *Wammes et al., 2022*). A central question is how these two seemingly contradictory mechanisms of representational change are employed during the construction of structured knowledge.

One hypothesized hippocampal mechanism is pattern separation, in which the hippocampus creates distinct neural representations for highly similar events (*O'Reilly and McClelland, 1994*; *Leutgeb et al., 2007*; *Yassa and Stark, 2011*). Through pattern separation, the hippocampus avoids the blending of similar experiences in memory, reducing across-event interference and forgetting. Tests of the pattern separation hypothesis include functional MRI (fMRI) studies manipulating the extent to which features of events are similar. Such studies have found intriguing evidence for pattern separation when experiences have high overlap or lead to similar outcomes (*Bakker et al., 2008*; *Schlichting et al., 2015*; *Kyle et al., 2015*; *Ballard et al., 2019*), with links to later remembering (*LaRocque et al., 2013*). Moreover, some fMRI studies have found evidence for pattern differentiation, where event overlap appears to trigger a 'repulsion' of hippocampal representations beyond baseline similarity to reduce interference (*Wammes et al., 2022*; *Favila et al., 2016*; *Chanales et al., 2017*; *Schapiro et al., 2012*; *Kim et al., 2017*; *Jiang et al., 2020*; *Hulbert and Norman, 2015*).

Despite evidence supporting the hippocampus's role in forming distinct memories for events, decades of work also suggest it is essential for building structured knowledge and forming cognitive maps that capture relations between spatial, perceptual, and conceptual features that were encountered in separate events that share common elements (*Tolman, 1948*; *O'Keefe and Nadel, 1978*; *Epstein et al., 2017*; *Schlichting et al., 2015*; *Tompary and Davachi, 2017*). For example, the hippocampus quickly extracts the temporal structure of an environment as defined by the transition probabilities between items in sequences (*Schapiro et al., 2012*; *Schapiro et al., 2013*), and it represents higher order structure when there is no variance in transition probabilities between items but an underlying community structure in a sequence (*Schapiro et al., 2016*). Moreover, it is essential for inferring indirect relationships between items and events (*Schapiro et al., 2014*; *Zeithamova et al., 2012*; *Vaidya et al., 2021*).

How do overlapping features from separate events lead to integration and generalization? Prior work suggests that retrieval-based learning, recurrent mechanisms within the episodic memory network, and memory replay enable representational integration and allow for the encoding of relationships that have never been directly experienced (*Schapiro et al., 2016*; *Kumaran and McClelland, 2012*; *Kumaran et al., 2016*; *Schapiro et al., 2017*; *Gupta et al., 2010*; *Wu and Foster, 2014*). For example, integrative encoding is thought to occur when a new experience (BC) triggers the recall of a related past episode (AB), and the concurrent activation of the remembered and presently experienced episodes results in the formation of an integrated representation (ABC) that can support future inference and generalization (*Shohamy and Wagner, 2008*). Integrative encoding may relate to other models that rely on recurrent connections between entorhinal cortex (EC) and the hippocampus to recirculate hippocampal output back into the system as new input, allowing for the discovery of higher order structure at the time of knowledge expression (*Kumaran and McClelland, 2012*; *Kumaran et al., 2016*; *Schapiro et al., 2017*). To the extent that the discovered structure is encoded, then across-event relationships are captured in an integrated representation. Moreover, replay within the episodic memory network is also thought to play an important role in updating mnemonic representations (*Ólafsdóttir et al., 2018*). Online replay during periods of awake rest has been shown to integrate multiple events, rather than just replaying a single event (*Gupta et al., 2010*). Offline replay during sharp wave-ripples is associated with memory consolidation and updates to neocortical knowledge structures (*Yu et al., 2018*; *Jadhav et al., 2016*; *Lewis and Durrant, 2011*; *Clewett et al., 2022*).

Despite a large and rapidly expanding theoretical and empirical literature, our understanding of how differentiation and integration occur across discrete experiences and contribute to the building of coherent, multi-level structured representations is still far from comprehensive. Additional insights

may come from examining these processes during the building of structured spatial knowledge, or maps of the external environment. For example, if one moves to a new city, one often first learns individual routes between navigational goals (i.e., from home to work, home to grocery store). Over time, the episodic memory network links individual routes, building a spatial map that enables planning of novel routes, detours, and shortcuts. Spatial knowledge acquired across learning consists of both local and global representations that are used to different extents based on navigational demands (*Chrastil, 2013*), and the hippocampus and EC are thought to be crucial for the acquisition of such representations. Spatial cells in these regions encode position, head-direction, speed, and other environmental features (*O'Keefe and Dostrovsky, 1971*; *McNaughton et al., 2006*; *Hafting et al., 2005*; *Kropff et al., 2015*), supporting Tolman's proposal that navigation is guided by internal cognitive maps and offering insight into how these maps are neurally implemented.

Spatial learning is rich with structure and thus well-suited for exploring the role of differentiation and integration in the formation of local and global knowledge across experiences. Indeed, a previous study in which participants viewed first-person trajectories to real-world destinations found that representations of routes with overlapping sections were differentiated in the hippocampus following learning (*Chanales et al., 2017*). Another study observed differentiation of hippocampal patterns for landmarks that were common to multiple virtual cities (*Zheng et al., 2021*). By contrast, other studies found evidence of integration for items that share spatial and temporal context (*Deuker et al., 2016*) and for objects located in geometrically similar positions across subspaces of segmented environments (*Peer et al., 2021*). More work is needed to characterize experience-driven changes in population-level neural representations of local and global spaces.

Towards this goal, we developed an experimental paradigm that leverages immersive virtual navigation and fMRI to investigate how neural representations of items encountered during goal-directed navigation evolve across learning. Specifically, pattern similarity analyses and linear mixed-effects models characterized experience-driven changes in EC and hippocampus following 'local' and 'global' navigation. We hypothesized that we would find evidence for both integration and differentiation emerging at the same time points across learning, as participants build local and global representations of the virtual environment. Moreover, we predicted that early evidence of global map learning during local navigation would depend on integration and predict participants' ability to subsequently navigate across the environment.

## Results

### Knowledge of local environments

Participants (n = 23) completed two behavioral navigation tasks interspersed between 3 consecutive days of fMRI (*Figure 1a*). Participants first completed a Local Navigation Task, in which they learned to navigate to five goal locations within each of three distinct local environments – that is, three oval tracks experienced in a virtual 3D environment from a first-person perspective (*Figure 1b*). For each track, participants completed four learning runs and six test runs (10 trials per run). At the start of each run, participants were placed on the track and rotated to orient themselves. Each trial began with a fractal cue indicating the navigational goal, followed by the selection of a heading direction and navigation (see Methods; *Appendix 1—figure 1*). We assessed goal-location learning by measuring the proportion of test trials on which participants correctly navigated to the cued location. Navigation was considered accurate if the participant landed within 8 arbitrary units of the goal (36% of the average distance between goals).

Across Day 1 test trials, participants navigated to the correct location on 96% of trials, with performance close to ceiling from the beginning of testing (*Figure 2a*, top). The proportion of correct trials was significantly above chance (0.25, or 1 of 4 possible goals; mean ± SEM: 0.961 ± 0.009; $t_{22}$ = 82.831, one-sample t-test; $p < 2.2e^{-16}$, d = 17.27) and did not differ across Day 1 test runs ($\beta$ = 0.003 ± 0.002; t = 1.701; p = 0.089). To assess whether knowledge of goal locations within local environments was retained overnight, participants completed two additional test runs (Runs 13–14; preceded by two runs of 'top-up' learning trials, Runs 11–12). Accuracy on Day 2 (mean ± SEM: 0.985 ± 0.004) did not differ from that on the final Day 1 test run ($t_{22}$ = 1.418; p = 0.17, paired t-test).

Participants were instructed to take the shortest path to a goal. To assess learning of spatial relationships between goals in local environments, we computed a path inefficiency metric that represents

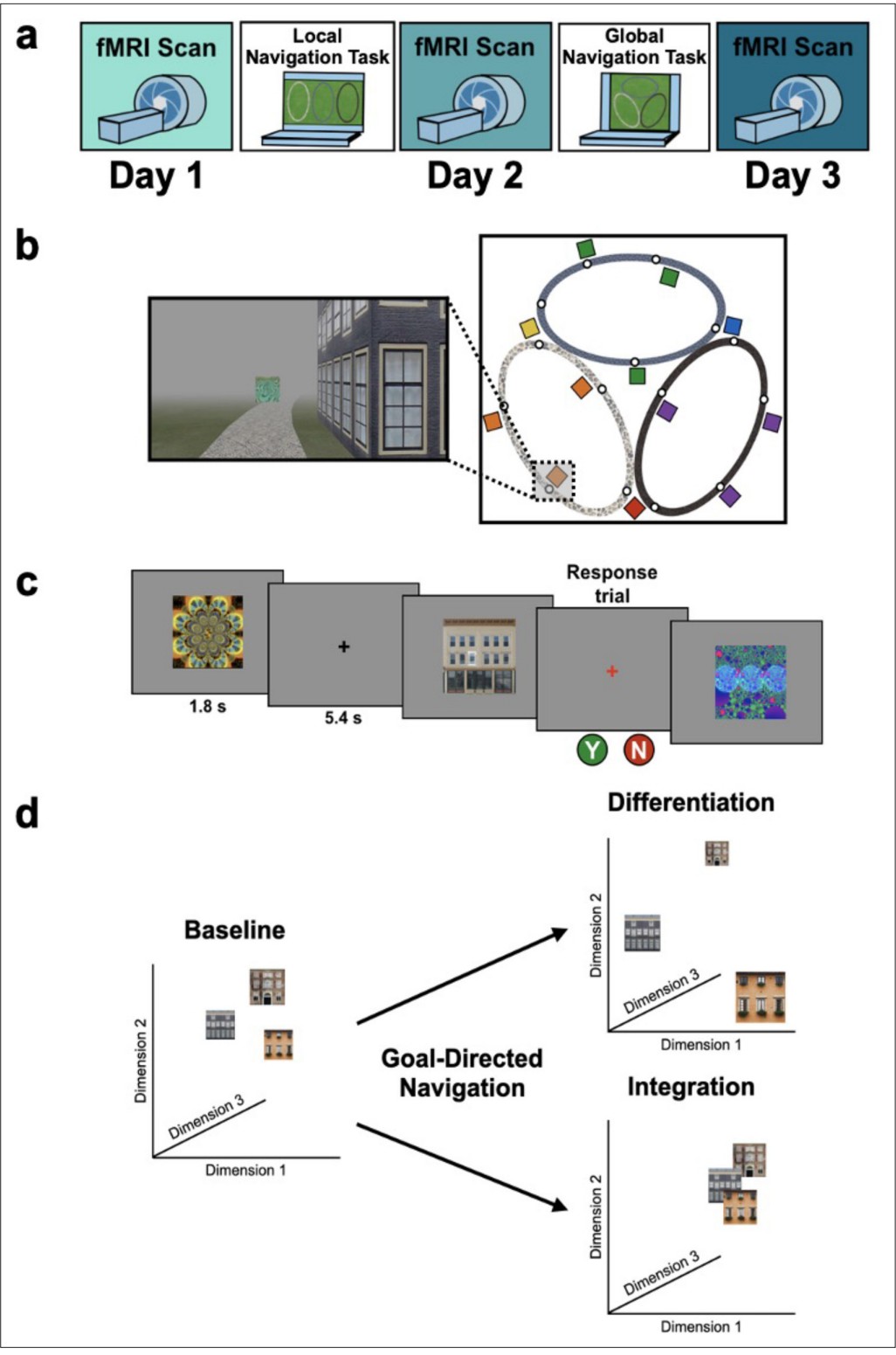

**Figure 1.** Study design. (**A**) Overview of the 3-day experimental paradigm. (**B**) First-person view from an example training trial (left); virtual fog limited the distance viewed, ensuring that no two landmark buildings could be seen at any one time. Overhead view of the virtual environment (right). The environment consisted of three oval tracks. Colored boxes indicate the approximate locations of landmarks, circles approximate goal locations for an individual study participant. (**C**) fMRI paradigm. Participants viewed images of the 12 landmarks and 15 fractals that were used in their unique virtual environment, while performing a perceptual decision-making task. On each

*Figure 1 continued on next page*

*Figure 1 continued*

trial, the stimulus appeared on a gray background for 1.8 s, followed by a fixation cross for 5.4 s; participants were instructed to attend to the stimuli and to determine whether a feature of the stimulus was 'bleached out'; on 'catch' trials (8% of trials), a red fixation cross appeared after image offset and participants indicated a response. (**D**) Schematic illustration of potential representational changes driven by learning. Following goal-directed navigation, the distance between landmarks in neural state-space could have increased (differentiation) or decreased (integration).

a participant's path length relative to the length of the shortest possible path (see Methods). During Local Navigation (*Figure 2a*, bottom), path inefficiency improved across the four learning runs (effect of run, $\beta$ = –8.99 ± 1.262; t = –7.125; p < 1.34e$^{-12}$; d = 1.49), was lower on the first test run relative to the first learning run (t$_{22}$ = 4.272; p = 0.0003, paired t-test; d = 0.89), and was consistently low across Day 1 test trials (mean ± SEM: 9.542 ± 1.773%; no effect of run, $\beta$ = –0.178 ± 0.378; t = –0.471; p = 0.638). As with accuracy, path inefficiency did not differ between Day 2 (mean ± SEM: 6.405 ± 1.681%) and the final Day 1 test run (t$_{22}$ = –0.619; p = 0.542, paired t-test). Together, these data demonstrate that participants quickly acquired precise knowledge of goal locations on the three tracks and the relations between locations within a track, as evidenced by their ability to successfully navigate between goals using nearly the shortest possible path.

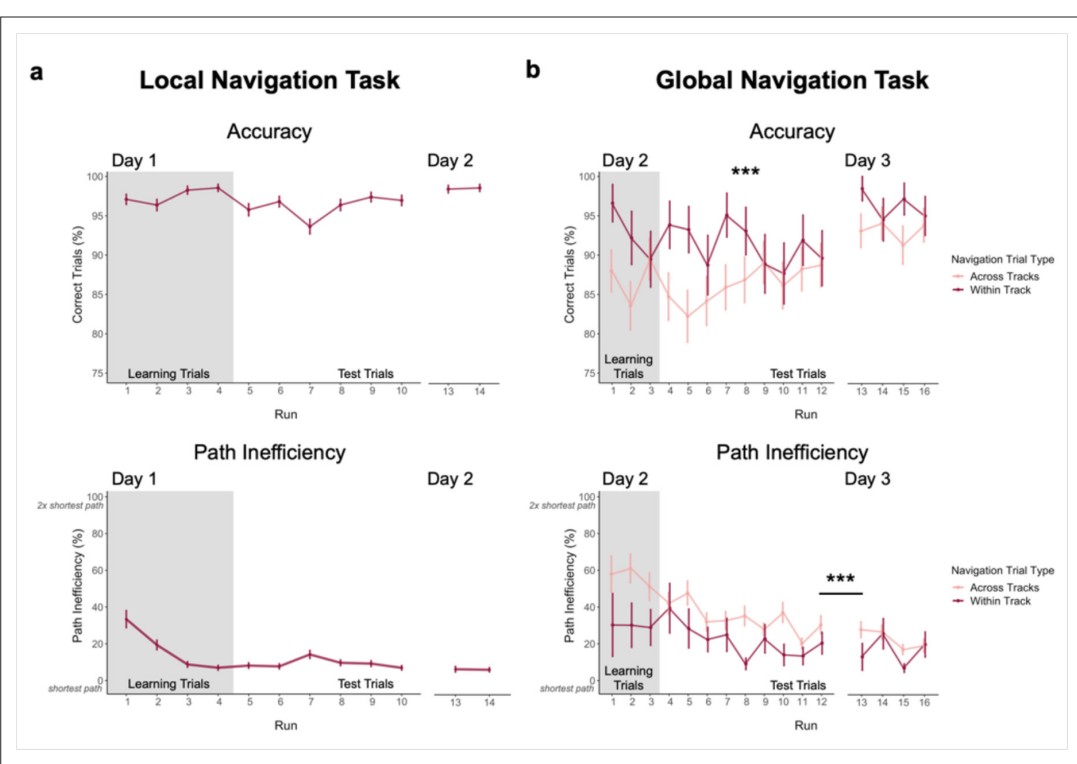

**Figure 2.** Navigation performance on the Local and Global Navigation Tasks. (**A**) Local Navigation accuracy on test trials was near ceiling across runs on Day 1 and Day 2 (top). Participants' navigational efficiency improved over learning trials, and they navigated efficiently across test trials (bottom). (**B**) During the Global Navigation Task, participants navigated more accurately (top) and efficiently (bottom) on within- vs. across-track trials. Participants were more accurate for within-track trials during learning and during early test runs, but improved on across-track trials over the course of test runs on Day 2 (top). Accuracy improved for both trial types on Day 3, such that performance did not significantly differ between within-track and across-track trials. Participants were significantly more efficient on both trial types on Day 3 relative to Day 2. (*** p < 0.001, paired t-test. Error bars denote SEM. Local Navigation Task, n = 23; Global Navigation Task, n = 21. Learning trials = trials on which the fractal marking the goal location was visible on the track; Test trials = trials on which participants had to rely on memory for the goal location, as the fractals were removed from the track).

## Knowledge of the global environment

Next, 21 of 23 participants completed a Global Navigation Task where the separately learned tracks were connected and participants were required to navigate to goals both within and across tracks. While within-track accuracy on Day 2 was lower in the Global Task (mean ± SEM: 0.913 ± 0.031; $t_{20}$ = –3.396; p = 0.003, paired t-test; d = 0.74, *Figure 2b*, top) than the preceding Local Task (mean ± SEM: 0.985 ± 0.004, *Figure 2a*, top), performance on the Global Task was above chance for both within-track (mean ± SEM: 0.913 ± 0.031) and across-track (mean ± SEM: 0.862 ± 0.034) trials (chance = 0.07, or 1 of 14 possible goals; within-track trials: $t_{20}$ = 27.029, one-sample t-test; p < $2.2e^{-16}$, d = 5.90; across-track trials: $t_{20}$ = 23.600, one-sample t-test; p < $2.2e^{-16}$, d = 5.15). During Global Navigation, within-track accuracy was higher than across-track accuracy ($t_{20}$ = 4.073; p = 0.0006, paired t-test; d = 0.89). However, across-track accuracy showed a trend towards increasing across test runs (effect of run, $\beta$ = 0.007 ± 0.004; t = 1.792; p = 0.073; no effect of run on within-track trials, $\beta$ = –0.005 ± 0.004; t = –1.103; p = 0.27), such that accuracy did not differ between trial types on the final Day 2 test run ($t_{20}$ = –0.029; p = 0.977, paired t-test). Before fMRI on Day 3, participants completed four additional test runs of Global Navigation. Accuracy significantly improved overnight for both trial types (within-track mean ± SEM: 0.963 ± 0.013, $t_{20}$ = 2.425, p = 0.025, paired t-test, d = 0.53; across-track mean ± SEM: 0.929 ± 0.025, $t_{20}$ = 2.841, p = 0.01, paired t-test, d = 0.62), and did not differ between them on Day 3 ($t_{20}$ = 1.886, p = 0.074, paired t-test; d = 0.41).

During Day 2, navigation was less efficient on the Global Task (*Figure 2b*, bottom) than the preceding Local Task (*Figure 2a*, bottom). Efficient navigation on across-track trials required a global representation of the virtual environment. During Day 2 learning runs, path inefficiency tended to be greater on across-track (mean ± SEM: 59.028 ± 7.124%) than within-track trials (mean ± SEM: 31.825 ± 13.392%; $t_{20}$ = 1.947, paired t-test; p = 0.066). Similarly, on Day 2 test runs, navigation was significantly more inefficient on across-track (mean ± SEM: 37.86 ± 6.389%) vs. within-track trials (mean ± SEM: 23.479 ± 5.764%; $t_{20}$ = 3.215, paired t-test; p = 0.004; d = 0.70). However, performance on both trial types improved over the course of Day 2 test runs (within-track trials, effect of run: $\beta$ = –2.27 ± 1.012; t = –2.244; p = 0.025; d = 0.49; across-track trials, effect of run: $\beta$ = –2.045 ± 0.647; t = –3.163; p = 0.002; d = 0.69), such that efficiency did not significantly differ between them during the final Day 2 test run ($t_{20}$ = 1.480, paired t-test; p = 0.155). As with accuracy, efficiency tended to improve overnight for both trial types (Day 3 within-track trials mean ± SEM: 17.176 ± 4.222%, $t_{20}$ = –3.043, paired t-test, p = 0.006, d = 0.66; Day 3 across-track trials mean ± SEM: 24.717 ± 4.741%, $t_{20}$ = –2.006, paired t-test, p = 0.059).

We also visually examined participants' routes on each across-track Global Navigation Task trial at the start (first four test runs) and end (last two test runs) of the task on Day 2, and at the end (last two test runs) of Day 3, noting whether participants (a) initially switched to the correct track on the trial and (b) navigated in the correct direction after switching. At the start of the Global Task on Day 2, participants switched to the correct track on 65.78% (SD = 21.35%) of across-track trials. Of those trials, participants navigated in the correct direction after switching 77.75% (SD = 22.52%) of the time. By the end of Day 2, those numbers increased to 75.09% (SD = 20.33%) and 85.64% (SD = 16.91%) respectively. Performance continued to improve on Day 3, with participants switching to the correct track on 78.36% (SD = 21.77%) of across-track trials during the last two runs of the Global Navigation Task and navigating in the correct direction 90.48% (SD = 10.63%) of the time after switching tracks.

Taken together, these data show that participants learned both the local environments (as evidenced by within-track navigation) and the global environment (as evidenced by across-track navigation), achieving high accuracy and efficiency on within- and across-track trials prior to the final scan on Day 3.

## Individual differences in global knowledge

Examination of performance at the participant level revealed striking individual differences in path inefficiency when navigating across tracks in the Global Navigation Task on Day 2 (range = 4.219–107.237%, Q1 = 17.914%, Q3 = 54.625%; Figure 6a). Some participants were highly efficient on across-track trials from the beginning, suggesting they had acquired much of the requisite global knowledge during performance of the preceding Local Task. By contrast, others became more efficient at across-track navigation over the course of the Global Task (*Appendix 1—figure 2*). This variance in when knowledge of the global environment was first evident allowed us to perform an individual differences analysis linking brain activity to behavior that depends on having built global environmental knowledge during Local

Navigation (see **Hippocampal representations predict later navigation performance** and Figure 6).

## fMRI assays of learning-driven representational change

Our primary objectives were to characterize representational changes in the human memory network driven by local and global environmental learning. Participants underwent fMRI at three timepoints across the study—Pre-Learning, Post Local Navigation, and Post Global Navigation (*Figure 1a*). During fMRI, participants viewed images of the landmark buildings and fractals used in their unique virtual environment, performing a low-level perceptual decision-making task. Of interest was how the neural patterns associated with each landmark and fractal changed as a function of learning (*Figure 1d*). Accordingly, we extracted voxel-level estimates of neural activity for each stimulus from EC and hippocampus regions-of-interest (ROIs; see Methods) and used pattern similarity analysis to probe whether learning resulted in representational differentiation or integration as a function of (a) the experienced relations between stimuli in the environment (e.g. same vs. different track, see **Hippocampus and entorhinal cortex learn to separate the three tracks**; distance within and across tracks, see **The hippocampus represents both local and global distance**) and (b) behavioral differences in when knowledge of the global environment was evident (i.e., evidence of good vs. poor across-track navigation at the outset of the Global Navigation Task). To test neural hypotheses, we fit linear mixed-effects models against pattern similarity, and included fixed effects of interest, nuisance regressors corresponding to the average univariate activation for each stimulus as well as estimates of perceptual similarity, and a standard set of random effects. Complete details of modeling procedures can be found in the Methods. In the following sections, we report results from both planned and exploratory analyses. For planned analyses, we interpret *a priori* effects when significant at p < 0.05 uncorrected, but for completeness we note whether all reported effects survive FDR correction (see Methods). Given theoretical interest in ventromedial prefrontal cortex (vmPFC) and the representation of structured knowledge, we report exploratory analyses on data from this region in the Appendix.

## Hippocampus and entorhinal cortex learn to separate the three tracks

EC and hippocampus encode a spatial map that may underlie the formation of a cognitive map (*O'Keefe and Nadel, 1978*; *O'Keefe and Dostrovsky, 1971*; *McNaughton et al., 2006*; *Hafting et al., 2005*). To examine how these regions build structured knowledge, we first asked whether they come to represent the three tracks following learning. We hypothesized that there would be a change in hippocampal and entorhinal pattern similarity for items located on the same track vs. items located on different tracks. An increase in pattern similarity would suggest that within-track item representations are integrated, while a decrease would suggest these representations are differentiated following learning. To test this hypothesis, we ran models predicting pattern similarity within each region, with scan (Pre-Learning/Day 1, Post Local Navigation/Day 2, and Post Global Navigation/Day 3), stimulus type (landmarks and fractals), and context (same track and different tracks) as predictors. We excluded shared landmarks from these models as they are common to multiple tracks, and restricted pattern similarity comparisons to be within stimulus-type (i.e., landmarks to other landmarks, fractals to other fractals).

In the hippocampus, an initial model that included hemisphere revealed a main effect of hemisphere ($\beta$ = 0.006 ± 0.002; t = 2.756; p = 0.006, survived FDR correction; d = 0.60; *Appendix 1—table 1*), and significant interactions between hemisphere and scan (Day 2 > Day 1 × hemisphere, $\beta$ = −0.011 ± 0.003, t = −3.477, p = 0.0006, survived FDR correction; d = 0.76; Day 3 > Day 1 × hemisphere, $\beta$ = 0.012 ± 0.003, t = 3.795, p = 0.0001, survived FDR correction; d = 0.83). The model also demonstrated a main effect of stimulus type ($\beta$ = 0.007 ± 0.003; t = 2.422; p = 0.015, did not survive FDR correction; d = 0.53) and an interaction between stimulus type and scan (Day 3 > Day 1 × stimulus type, $\beta$ = −0.011 ± 0.004; t = −2.604; p = 0.009, survived FDR correction; d = 0.57). The latter interaction reflected a difference in similarity between landmarks and fractals Pre-Learning (Day 1 fractals > Day 1 landmarks, $\beta$ = −0.009 ± 0.002; z-ratio = −4.269; p = 0.0003 adjusted) which was no longer present following Global Navigation (Day 3 fractals > Day 3 landmarks, $\beta$ = 0.003 ± 0.002; z-ratio = 1.248; p = 0.813 adjusted).

Given these interactions, we ran models for each hemisphere and stimulus type separately. As *a priori* predicted, we found evidence in right hippocampus that learning differentiates landmarks

located on the same track (Day 3 > Day 1 × context, $\beta$ = –0.013 ± 0.007; t = –2.014; p = 0.044, did not survive FDR correction; d = 0.44; *Figure 3a*; *Appendix 1—Tables 2–3*). In right hippocampus, Pre-Learning pattern similarity was comparable for within- vs. across-track landmarks. Post Global Navigation, patterns for landmarks on the same track were less similar (differentiated) than those for landmarks on different tracks. No significant interactions between scan and context were observed for fractals (*Figure 3b*; *Appendix 1—Tables 4–5*).

Turning to EC, an initial model with left and right EC revealed a main effect of hemisphere ($\beta$ = –0.022 ± 0.004; t = –6.06; p < 1.38e$^{-9}$, survived FDR correction; d = 1.43; *Appendix 1—table 6*), and interactions between hemisphere and scan (Day 2 > Day 1 × hemisphere, $\beta$ = 0.022 ± 0.005; t = 4.192; p < 2.78e$^{-5}$, survived FDR correction; d = 0.99; Day 3 > Day 1 × hemisphere, $\beta$ = 0.056 ± 0.005; t = 10.552; p < 2e$^{-16}$, survived FDR correction; d = 2.49). This model also revealed an interaction between context and scan (Day 2 > Day 1 × context, $\beta$ = –0.014 ± 0.006; t = –2.05; p = 0.041, did not survive FDR correction; d = 0.48), such that there was no difference in similarity between within-track and across-track items Pre-Learning (Day 1 same track > Day 1 different track, $\beta$ = 0.004 ± 0.004; z-ratio = 1.162; p = 0.855 adjusted), nor were within-track items less similar to each other (i.e. differentiated) following Local Navigation (Day 2 same track > Day 2 different track, $\beta$ = –0.009 ± 0.004; z-ratio = –2.447; p = 0.140 adjusted). No main effect or interactions were observed with stimulus type.

Given the interaction with hemisphere, we ran separate models to examine left and right EC representations of the three tracks as a function of learning. As predicted, we observed an interaction between context and scan in left EC (Day 2 > Day 1 × context, $\beta$ = –0.014 ± 0.006; t = –2.339; p = 0.019, did not survive FDR correction; d = 0.52; *Figure 3c*; *Appendix 1—table 7*), such that similarity for same-track vs. across-track items did not differ Pre-Learning, but same-track items were less similar (differentiated) following Local Navigation. Following Global Navigation, pattern similarity remained lower for within-track items, but the interaction between context and scan did not reach statistical significance (Day 3 > Day 1 × region, $\beta$ = –0.009 ± 0.006; t = –1.457; p = 0.145). A similar pattern of findings was observed in right EC, but did not reach statistical significance (*Figure 3c*; *Appendix 1—table 8*).

To determine whether the effects of learning on representations of the three tracks differed between EC and hippocampus, we included region in a complete model, testing whether pattern similarity in EC for within- vs. across-track items differed from that in the hippocampus. We found a main effect of region ($\beta$ = –0.015 ± 0.002; t = –7.36; p < 1.86e$^{-13}$, survived FDR correction; d = 1.74; *Appendix 1—table 9*), as well as interactions between region and scan (Day 2 > Day 1 × region, $\beta$ = –0.017 ± 0.003; t = –5.757; p < 8.59e$^{-9}$, survived FDR correction; d = 1.36; Day 3 > Day 1 × region, $\beta$ = 0.014 ± 0.003; t = 4.592; p < 4.41e$^{-6}$, survived FDR correction; d = 1.08). Importantly, there was no interaction between region, scan, and context (*Appendix 1—table 9*), which suggests statistical support for a functional differentiation between these two regions is absent.

## The hippocampus represents both local and global distance

In studies of episodic memory and spatial navigation, the hippocampus is thought to play a crucial role in disambiguating memories for overlapping events (*Yassa and Stark, 2011*; *LaRocque et al., 2013*; *Favila et al., 2016*; *Chanales et al., 2017*). Moreover, prior work indicates that spatial distance is reflected in both hippocampal pattern similarity (*Deuker et al., 2016*) and univariate BOLD activity (*Nyberg et al., 2022*; *Howard et al., 2014*; *Patai et al., 2019*). We hypothesized that for participants to navigate successfully, they would need to form distinct representations of local environmental landmarks used to aid navigation. We tested whether learning of local environments led to differentiated hippocampal activity patterns by examining pattern similarity for landmarks located on the same track that were nearest neighbors (link distance 1) vs. slightly further away (link distance 2; *Figure 4a*). Here, we predicted pattern similarity would be lower for nearest neighbors following Local and Global Navigation. We ran a model predicting pattern similarity, with scan session and link distance as predictors.

An initial model including hemisphere revealed no main effect or interactions with hemisphere (*Appendix 1—table 10*). As *a priori* predicted, we found significant interactions between distance and scan (Day 2 > Day 1 × distance, $\beta$ = 0.04 ± 0.015; t = 2.70; p = 0.007, fell near FDR-corrected threshold (p = 0.005), d = 0.56; Day 3 > Day 1 × distance, $\beta$ = 0.034 ± 0.015; t = 2.259; p = 0.024; did not survive FDR correction; d = 0.49; *Appendix 1—table 11*), such that following the Local and Global Tasks, hippocampal patterns were less similar (differentiated) for nearby landmarks vs. landmarks

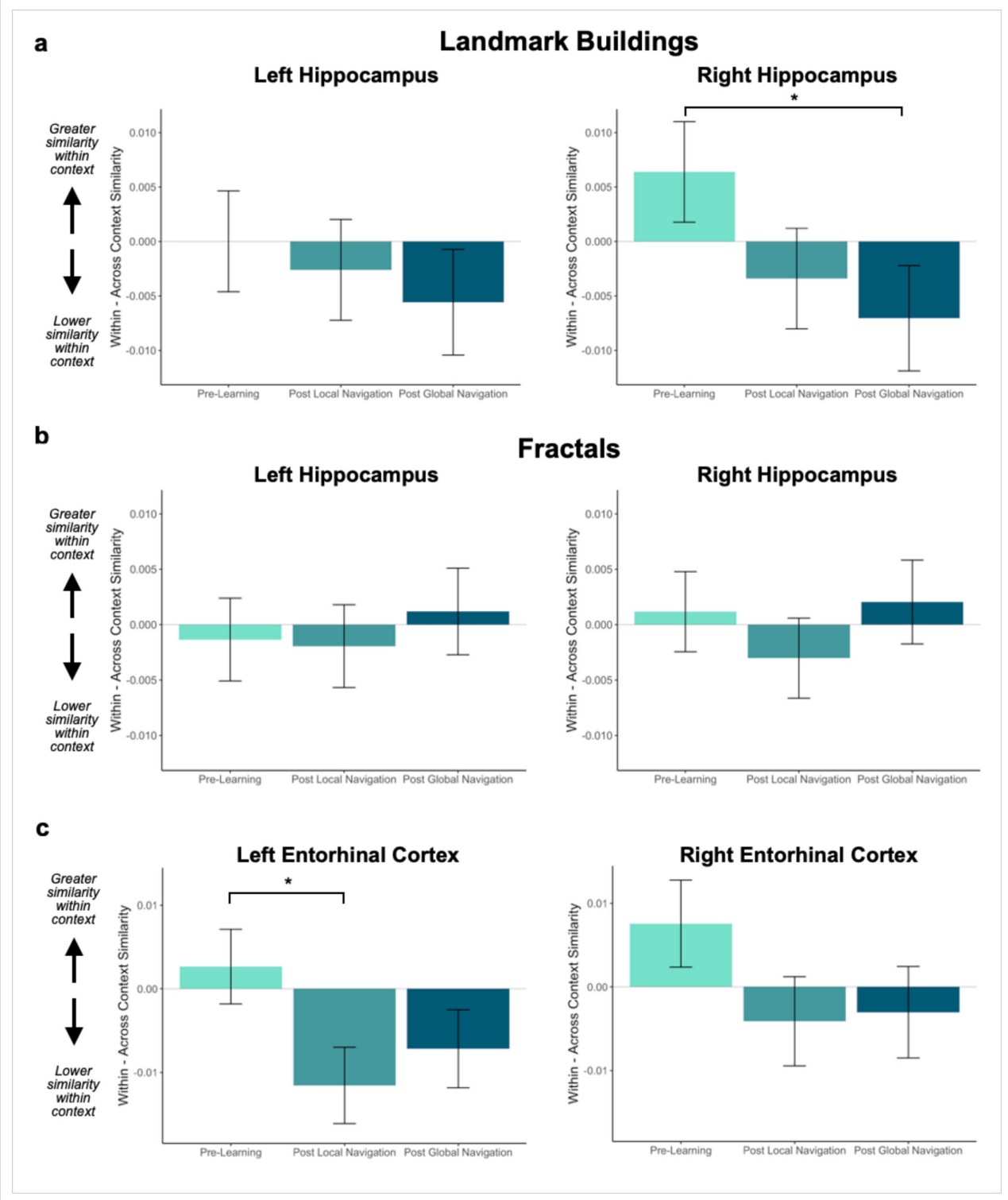

**Figure 3.** Context representations in hippocampus and entorhinal cortex (EC). (**A**) Contrast estimates for models predicting landmark similarity in left and right hippocampus. Right hippocampus differentiates landmarks located on the same track following Global Navigation, such that those experienced within the same track became less similar. A similar pattern of findings was observed in left hippocampus but did not reach statistical significance. (**B**) Contrast estimates for models predicting fractal similarity in left and right hippocampus. Interactions between scan session and context were not significant. (**C**) Contrast estimates for models in left and right EC that include both landmark and fractal stimuli. Left EC differentiates items located on the same track following Local Navigation, such that items experienced within the same track became less similar. Following Global Navigation, pattern similarity remained lower for within-track items, but the interaction between context and scan session did not reach statistical

*Figure 3 continued on next page*

*Figure 3 continued*

significance. A similar pattern of findings was observed in right EC, but interactions between context and scan session did not reach statistical significance (a *priori* predicted effects: * p < 0.05, uncorrected. Error bars denote SE of the estimates. Hippocampus: Day 2 > Day 1, n = 23; Day 3 > Day 1, n = 21; Left EC: n = 20; Right EC: n = 18).

located further apart (*Figure 4b*). There were no significant interactions between distance and scan when a similar model was run for fractal stimuli (*Appendix 1—table 12*).

In addition to encoding local (within-track) spatial representations, navigation may result in the acquisition of a global map of the virtual environment. Although participants performed the Local Task separately on each track, it is possible that the landmarks shared across tracks trigger integration

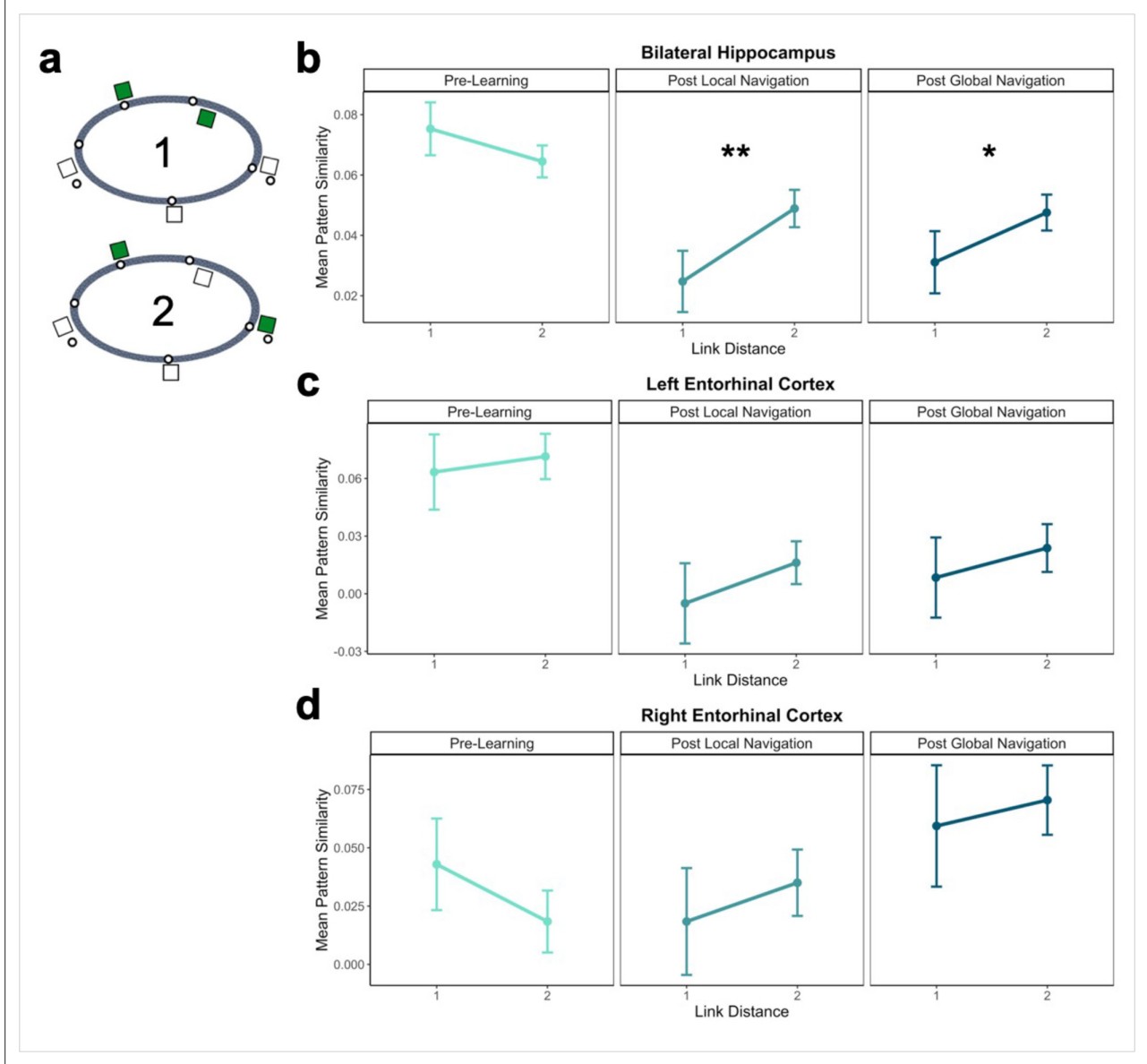

**Figure 4.** Hippocampal pattern similarity reflects distance in local environments. (**A**) Examples of landmarks at link distances 1 and 2 on the same track. (**B**) Hippocampal pattern similarity for within-track landmarks Pre-Learning (left), after the Local Navigation Task (center), and after the Global Navigation Task (right). Interactions between distance and scan session were significant from Pre-Learning to Post Local and Global Navigation. (**C–D**) Pattern similarity for within-track landmarks in the left (**C**) and right (**D**) EC. Interactions between distance and scan session were not significant. (a *priori* predicted effects: ** p < 0.01, * p < 0.05, uncorrected. Error bars denote SE of the estimates. Hippocampus: Day 2 > Day 1, n = 23; Day 3 > Day 1, n = 21; Left EC: n = 20; Right EC: n = 18).

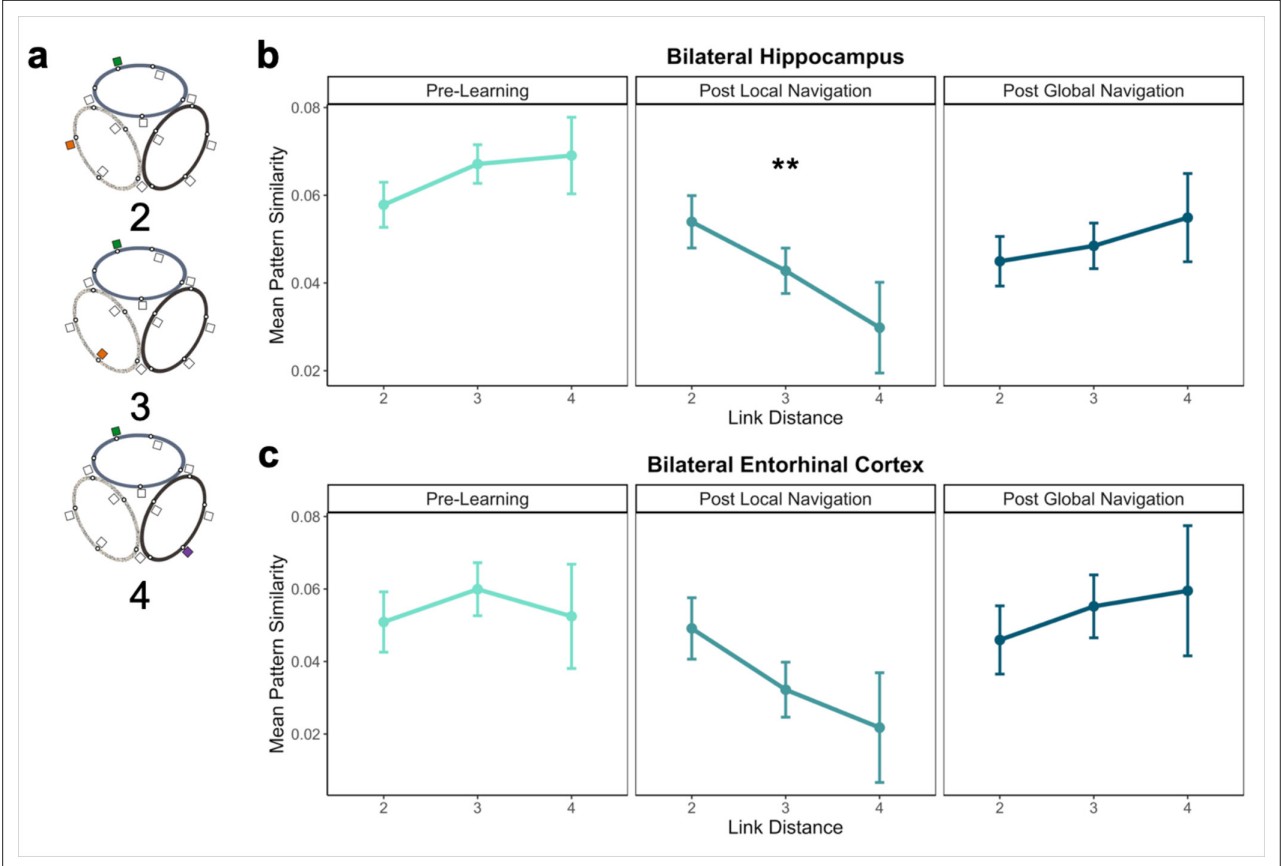

**Figure 5.** Distance representations in the Global environment. (**A**) Examples of landmarks at different link distances on different tracks. (**B**) Hippocampal pattern similarity for landmarks on different tracks Pre-Learning (left), after the Local Navigation Task (center), and after the Global Navigation Task (right). The interaction between distance and scan session was significant from Pre-Learning to Post Local Navigation, but not to Post Global Navigation. (**C**) A similar pattern of findings was observed in Entorhinal Cortex, but the interaction between distance and scan session was not significant from Pre-Learning to Post Local Navigation (a *priori* predicted effects: ** p < 0.01, uncorrected. Error bars denote SE of the estimates. Hippocampus: Day 2 > Day 1, n = 23; Day 3 > Day 1, n = 21; EC: n = 18).

of the tracks, contributing to the encoding of (at least some dimensions of) a global environmental map. We were particularly interested in whether there would be evidence of global knowledge of the environment (i.e., knowledge of relations that span tracks) after the Local Task, prior to any experience navigating across the connected tracks.

To this end, we examined pattern similarity for landmarks located on different tracks at increasing distances, predicting it would scale with distance such that pattern similarity would be greatest for proximal vs. distal landmarks (*Figure 5a*). To test our prediction, we ran a model predicting pattern similarity between landmarks on different tracks, with scan (Pre-Learning/Day 1, Post Local Navigation/Day 2, and Post Global Navigation/Day 3) and link distance (2, 3, and 4) as predictors. We excluded shared landmarks from this model as they are common to multiple tracks; however, the results do not differ if these landmarks are included in the analysis. An initial model that included hemisphere revealed no main effect or interactions with hemisphere (*Appendix 1—table 13*).

Consistent with our prediction, we found an interaction between link distance and scan following the Local Task (Day 2 > Day 1 × distance, $\beta$ = –0.02 ± 0.007; t = –2.891; p = 0.004, survived FDR correction; d = 0.60), but not the Global Task (Day 3 > Day 1 × distance, $\beta$ = –0.003 ± 0.007; t = –0.413; p = 0.679; *Figure 5b*; *Appendix 1—table 14*). In contrast to the distance-related differentiation observed between spatial locations within a track, this interaction reflected that hippocampal pattern similarity for locations across tracks did not vary as a function of distance Pre-Learning, but was higher for closer (vs. farther) locations across tracks Post Local Navigation. This finding suggests that some global map knowledge is evident in the hippocampus even though participants had only engaged in within-track

navigation. Following Global Navigation and in contrast to our predictions, similarity was comparable at all across-track distances. There were no significant interactions between distance and scan when a similar model was run for fractal stimuli (*Appendix 1—table 15*).

We tested for within- and across-track distance effects within EC as well, given the extensive literature characterizing spatial coding in the region. As with the hippocampus, we tested whether pattern similarity for landmark buildings in EC scaled with local (within-track) and global (across-track) distance. An initial model of local distance that included hemisphere (left and right EC) and link distance (1 and 2) revealed interactions between scan session and hemisphere (Day 2 > Day 1 × hemisphere, $\beta$ = 0.050 ± 0.023, t = 2.093, p = 0.036, did not survive FDR correction; d = 0.47; Day 3 > Day 1 × hemisphere, $\beta$ = 0.069 ± 0.024, t = 2.835, p = 0.005, survived FDR correction; d = 0.67; *Appendix 1—table 16*). However, in contrast to hippocampus, we found no interactions between distance and scan session (*Figure 4c–d*, *Appendix 1—Tables 17–18*) when we fit models to data from each hemisphere individually. An initial model of global distance that included hemisphere (left and right EC) and link distance (2, 3, and 4) revealed no main effect of hemisphere or interactions with hemisphere (*Appendix 1—table 17*). Unlike the hippocampus, we found no interactions between distance and scan session in the global model (*Figure 5c*, *Appendix 1—table 20*). No significant interactions between distance and scan were observed when similar models were run for fractal stimuli (*Appendix 1—Tables 21–23*). Our findings in EC were not surprising, as the extent to which EC itself can support structured representations is unclear. Spatial properties of EC neurons are known to be important for path integration and the building of structured knowledge in the hippocampus and neocortex (*McNaughton et al., 2006*; *Hafting et al., 2005*). Yet, hippocampal conjunctive coding and interactions between the hippocampus, EC, and neocortex are necessary for cross event-generalization and retrieval-mediated learning (*Zeithamova et al., 2012*; *Kumaran and McClelland, 2012*; *Kumaran et al., 2016*; *Schapiro et al., 2017*; *Preston and Eichenbaum, 2013*).

## Hippocampal representations predict later navigation performance

The observed negative relationship between hippocampal pattern similarity and the distance between across-track landmarks that emerged after Local Navigation (*Figure 5b*), but before participants experienced the connected tracks, was quite variable across participants.

The variability in the observed across-track hippocampal distance effect may reflect that some participants encoded global map knowledge during Local Navigation, whereas others did not (or did so less fully; *Figure 6a*). To the extent that this is the case, this would predict that the distance-related hippocampal pattern similarity effect Post Local Navigation should relate to navigational efficiency at the outset of performing the Global Task. Specifically, we predicted that more efficient navigators would have a negative distance function, such that pattern similarity would be greatest for the most proximal across-track landmarks and decrease with distance. To test this hypothesis, we first ran a mixed-effects model predicting neural pattern similarity Post Local Navigation (Day 2), with path inefficiency (median path inefficiency for across-track trials in the first four test runs of Global Navigation on Day 2) and link distance as predictors. Indeed, we observed a significant interaction between path inefficiency and link distance ($\beta$ = –0.001 ± 0.001; t = –1.983; p = 0.048, did not survive FDR correction; d = 0.43; *Appendix 1—table 24*), but the direction of the effect was unexpected. Participants who did well from the beginning of the Global Task showed no effect of distance in hippocampal pattern similarity, whereas less efficient navigators showed a negative slope (*Figure 6b*). A similar interaction between path inefficiency and link distance was not observed when the model was fit to data from Day 1 ($\beta$ = 0.001 ± 0.001; t = 1.523; p = 0.128; *Appendix 1—figure 3a*; *Appendix 1—table 25*) or to data from Day 3 ($\beta$ = 0.001 ± 0.001; t = 1.379; p = 0.168; *Appendix 1—figure 3b*; *Appendix 1—table 26*).

Next, we examined single-trial navigation data in relation to pairwise neural similarity in the hippocampus (for each given pair of landmarks and fractals), using mixed-effects models to predict path inefficiency for each trial across the first four test runs of the Global Navigation Task (Day 2). The models included (a) neural similarity (Post Local Navigation) for a given pair of fractals or the nearby landmarks and (b) length of the optimal path for each trial as predictors, and a regressor indicating whether the trial was a within-track or across-track trial. Models were run for landmarks and fractals separately. Here, we observed trend-level evidence that hippocampal pattern similarity (Post Local Navigation) for landmark pairs predicted trial-level subsequent Global Navigation performance ($\beta$ = –41.245 ± 24.162; t = –1.707; p = 0.088; *Appendix 1—table 27*), such that greater hippocampal

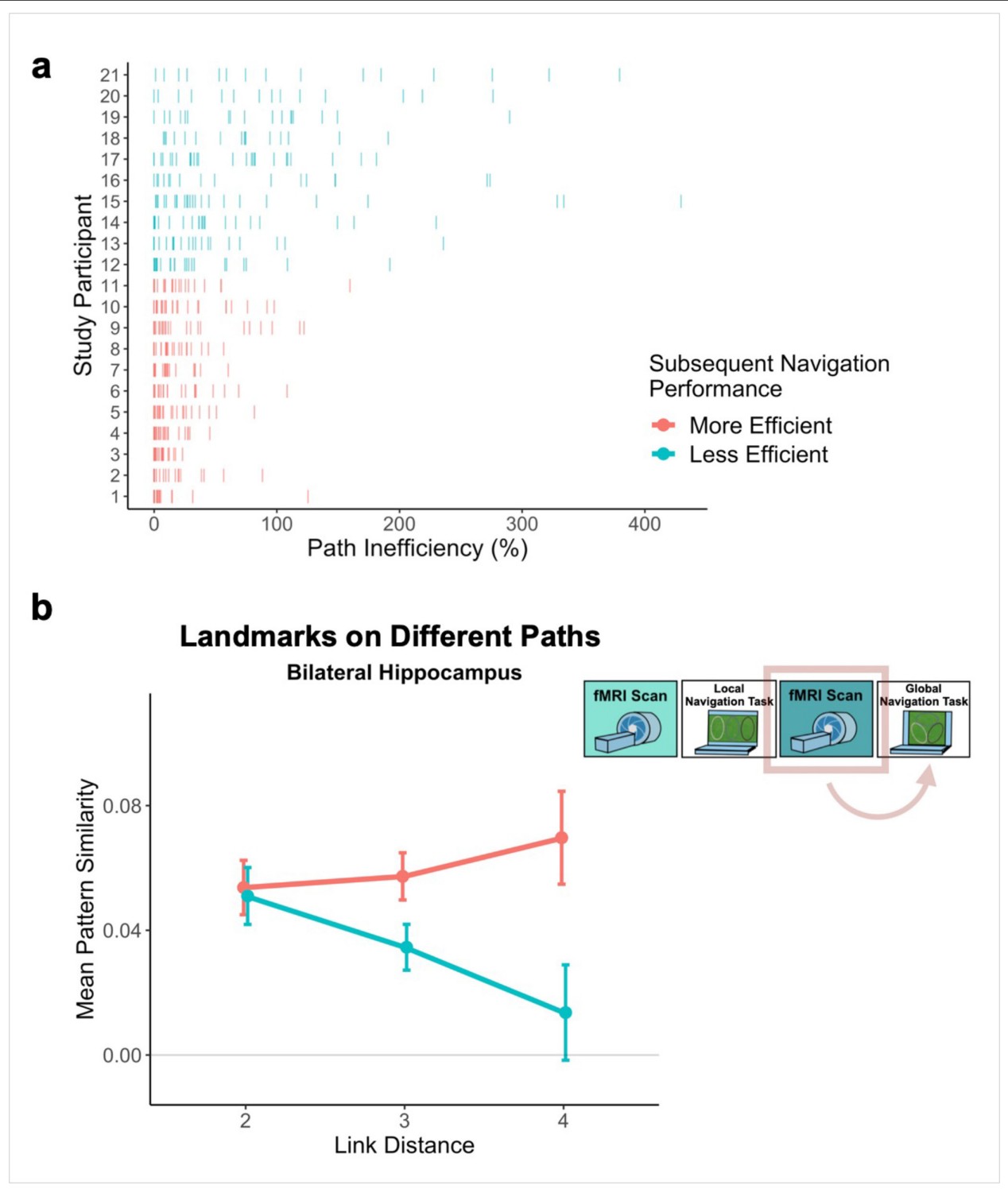

**Figure 6.** Path inefficiency on the Global Navigation Task varies with across-track distance-related hippocampal pattern similarity after the Local Navigation Task. To qualitatively visualize the relationship between pattern similarity, link distance, and path inefficiency, we split participants into two groups – More Efficient and Less Efficient – based on their median path inefficiency on across-track trials in the first four test runs of the Global Task on Day 2. (**A**) Path inefficiency (%) for each across-track trial during the first four test runs of the Global Navigation Task, plotted for each participant and colored by performance group. (**B**) We used a linear mixed-effects model to formally test this relationship (see main text for details). The linear model revealed a significant interaction between path inefficiency and link distance, with the direction of the effect being unexpected. To qualitatively depict this effect, we plot hippocampal pattern similarity for landmarks on different tracks prior to the Global Navigation Task for the Less Efficient and More

*Figure 6 continued on next page*

*Figure 6 continued*

Efficient median-split data. Data are split by participants' subsequent navigation performance as shown in (**A**). (Error bars denote SE of the estimates. More Efficient, n = 11; Less Efficient, n = 10).

pattern similarity was predictive of a more efficient path (*Appendix 1—figure 4*). The length of the optimal path ($\beta$ = –0.663 ± 0.183; t = –3.631; p = 0.0003; d = 0.79) and trial type (within-track vs. across-track; $\beta$ = 20.551 ± 6.245; t = 3.291; p = 0.001; d = 0.71) significantly predicted navigation performance in the landmark model. There was no interaction between hippocampal pattern similarity for landmark pairs and trial type ($\beta$ = 54.618 ± 43.489; t = 1.256; p = 0.210).

The length of the optimal path ($\beta$ = –0.646 ± 0.184; t = –3.519; p = 0.0005; d = 0.77) and trial type (within-track vs. across-track; $\beta$ = 27.289 ± 6.196; t = 4.404; p < 1.24e$^{-5}$, survived FDR correction; d = 0.96; *Appendix 1—table 28*) were also significant predictors of navigation performance in the model relating pattern similarity for fractal pairs to subsequent performance on the Global Task. However, hippocampal pattern similarity for fractal pairs did not significantly predict subsequent performance ($\beta$ = 3.026 ± 27.687; t = 0.109; p = 0.913).

## Discussion

The episodic memory network builds structured knowledge across experiences to form internal models of the world that enable planning, decision-making, and goal-directed behavior. In this study, we investigated how humans build structured knowledge in immersive, goal-directed tasks. Results revealed that (a) learning restructures representations in the hippocampus and EC, reflecting the structure of the virtual environment; (b) the hippocampus begins to build a representational structure extending beyond directly experienced transitions; and (c) changes in the similarity structure of hippocampal representations relate to subsequent navigation performance.

We first characterized learning on two behavioral navigation tasks. In the Local Navigation Task, participants quickly learned goal locations along individual tracks and navigated between goals using the most efficient paths (*Figure 2*). Prior to the Global Navigation Task, participants were unaware that the tracks would be connected and that they would be required to navigate across tracks to reach cued goals. Nonetheless, some were able to navigate efficiently from the outset of Global Navigation, suggesting they had already formed a global representation of the environment; by contrast, others required extensive experience to do so (*Figure 6a*). Our findings are consistent with behavioral research examining individual differences in route integration and cognitive map formation. Prior work characterized navigators into groups of 'integrators', 'nonintegrators', and 'imprecise navigators', suggesting that while some individuals build accurate internal maps, others may rely on fragmented maps or route-based strategies during navigation (*Weisberg et al., 2014*; *Weisberg and Newcombe, 2016*; *Weisberg and Newcombe, 2018*). Importantly, individual differences in navigation inefficiency enabled us to probe variability in neural representations and relate neural pattern similarity to behavior. Our work provides an important extension of prior studies in that it identifies potential neural substrates of individual differences in cognitive map formation.

At the neural level, we investigated whether mnemonic regions come to represent the three local tracks following navigation by comparing pattern similarity for items located on the same track vs. different tracks. Motivated by recent studies finding lower hippocampal pattern similarity for events with overlapping vs. distinct features (*Schlichting et al., 2015*; *LaRocque et al., 2013*; *Favila et al., 2016*; *Chanales et al., 2017*; *Hulbert and Norman, 2015*), we expected to observe changes in hippocampal pattern similarity for within-track items following local learning. Indeed, we found evidence for differentiation in EC: items on the same track elicited less similar activity patterns when compared to patterns for items on different tracks following local learning (*Figure 3c*). We observed a quantitatively similar pattern of effects for landmarks in the hippocampus (*Figure 3a*).

Several factors may explain why we did not find stronger hippocampal differentiation effects. The demands of our navigation tasks differ substantially from those of prior tasks eliciting strong differentiation effects (*Schlichting et al., 2015*; *Kyle et al., 2015*; *Favila et al., 2016*; *Chanales et al., 2017*; *Brown et al., 2010*). In prior studies, participants explicitly learned overlapping associations that had to be distinguished at a later test; for instance, the same item was paired with two different associates, or similar items or routes led to distinct outcomes. In the present study, overlapping features were

incidental to task demand; thus individuals may have adopted different navigational strategies with varying consequences for within-track item pattern similarity (for instance, some participants may have learned the sequence of fractals for each track, while others may have formed associations between fractals and nearby landmarks). Tracks had a number of features in common, including shared visual features, a similar spatial layout, and common landmarks. Further, our findings from across-track analyses examining individual differences in navigation suggest that some participants began to build an integrated, global representation of the environment prior to the Global Task, with associated effects in the hippocampus. Given these dynamics, additional variance inherent in our paradigm may explain the nature of the observed effects in hippocampus and EC.

While we did not observe particularly strong differentiation of hippocampal patterns for all items located on the same track, we hypothesized that participants would need to form distinct representations of local environmental landmarks to perform well. Accordingly, we examined local spatial representations by comparing the similarity between nearest neighboring landmarks to second nearest neighbors on the same track, and found that hippocampal patterns for nearest neighboring landmarks were differentiated following both Local and Global Navigation (*Figure 4b*). This finding is consistent with the idea that mnemonic representations – and in particular, representations of large-scale spaces – are hierarchical (*McKenzie et al., 2014*; *Milivojevic and Doeller, 2013*; *Chrastil and Warren, 2014*). That is, we found differentiated representations of local space, whereas representations of track context or of the spatial relationships between the tracks were perhaps more coarse.

We next investigated global representations of map knowledge within the virtual environment. Here, we examined hippocampal pattern similarity for landmarks encountered on different tracks at increasing distances in the virtual environment, expecting to see the emergence of a relationship between hippocampal pattern similarity and distance. We were particularly interested in whether this relationship would be observed after the Local Task, which would suggest that participants were starting to build a global map of the environment prior to direct experience navigating the connected tracks.

Our results provide support for this prediction – we found a significant interaction between distance and scan (Pre-Learning vs. Post Local Navigation; *Figure 5b*, center). Notably, we also observed significant variability in the slope of this function across individuals (*Figure 6b*). We hypothesized that this variability in the neural data might relate to the behavioral variability we observed on the subsequent Global Navigation Task, and explored this relationship in two ways. First, we ran a mixed effects model predicting hippocampal pattern similarity Post Local Navigation, with across-track path inefficiency at the start of Global Navigation and link distance as predictors. Here, we observed a significant interaction between path inefficiency and link distance, but the effect was not in the direction we expected. Our results indicate that the interaction between distance and scan (the negative slope in the center panel of *Figure 5b*) was driven by participants who were *less* efficient at the outset of Global Navigation, whereas we predicted a negative slope would be more apparent in highly efficient navigators. We initially hypothesized a negative distance-related similarity function would reflect global map learning, and thus expected to observe such a function in efficient navigators at the start of the Global Task on Day 2 and across all participants on Day 3. Such a function was absent following Global Navigation (the right panel of *Figure 5b*).

Why do we observe a flat distance-related similarity function after global map learning? It is possible that individual differences in navigational strategy or the particular learning processes utilized by efficient navigators in the present task organized the map in such a way. It is also possible our findings reflect a change in event boundaries. Evidence from a study examining temporal context found that hippocampal pattern similarity did not differ between items located nearby vs. further apart within a temporal event (*Ezzyat and Davachi, 2014*). Hippocampal pattern similarity differed, however, between items that crossed an event boundary, such that nearby items had increased pattern similarity compared to items located further apart in time. While this study examined temporal events, spatial event boundaries may function similarly in that during retrieval, representations of other within-event items are reinstated. The flat slope observed on Day 3 could thus be a signature of an integrated map.

Our second approach to relating variability in the neural data to behavioral variability used a model to predict path inefficiency for each trial across the first four test runs of the Global Navigation Task (Day 2). While the present study was not designed to optimize power to detect trial-level effects, here we found trend-level evidence that hippocampal pattern similarity (Post Local Navigation) for

landmark pairs predicted trial-level subsequent Global Navigation performance (*Appendix 1—figure 4*), providing additional suggestive evidence for a relationship between hippocampal pattern similarity and subsequent behavior.

Replay and recurrence within the hippocampal network is thought to enable the hippocampus to learn relationships between items that have never been directly experienced together (*Zeithamova et al., 2012*; *Kumaran and McClelland, 2012*; *Kumaran et al., 2016*; *Gupta et al., 2010*; *Wu and Foster, 2014*). The present study provides complementary evidence that the hippocampus begins to build a cognitive map extending beyond directly experienced transitions in a goal-directed task. Overlapping features within the virtual environment (the landmarks shared between tracks) provide the hippocampus with the links it needs to begin building a global map, despite participants not yet experiencing the connected environment. Crucially, not all participants demonstrated integration early in the Global Task, suggesting that cognitive map formation may not be a 'default' that incidentally emerges during performance of immersive, goal-directed local navigation. An open question is: How does this variability in whether and how individuals build global knowledge of the environment during local navigation relate to individual differences in spatial and mnemonic processing, in navigational strategy, and/or in other cognitive and contextual factors (*Weisberg and Newcombe, 2016*; *Weisberg and Newcombe, 2018*; *Wolbers and Hegarty, 2010*; *Brunec et al., 2022*)? The current study was limited in its investigation of individual differences due to sample size. Future work is needed to characterize the differences between performance groups and examine how representations of individual tracks and particular spatial positions within tracks covary with performance on subsequent global navigation. Moreover, some of our findings do not survive correction for multiple comparisons, and thus should be interpreted with caution.

The present findings appear to diverge from work finding hippocampal pattern similarity scaled with perceived spatiotemporal distance in a virtual environment, such that patterns for nearby items were integrated (*Deuker et al., 2016*). In that study, participants were scanned before and after a task with a duration of ~80 min; thus, the findings may reflect hippocampal representations at an earlier stage of learning. In the present paradigm, fMRI took place 21.5–31 hr after extensive navigational practice, raising the possibility that hippocampal differentiation requires extensive learning, potentially including learning that takes place via replay and consolidation processes occurring during sleep. Future studies are necessary to pinpoint when integration occurs across the broader global map.

While we expected similar results for both fractal and landmark stimuli throughout our analyses, the null findings we observed across ROIs when local and global distance models were run with fractal stimuli were not completely surprising. Fractal and landmark stimuli differ in several key ways, which we believe explain the observed pattern of findings. For example, fractals are only visible in the environment for a minority of trials. During the majority of navigation, participants must rely on the landmark buildings to guide them. Fractals serve as pointers to the goal location to which a participant must navigate on a particular trial, but fractal information may not be used during route planning. Further, fractals are not necessary for participants to learn the layout of the environment; local and global maps can be built from landmarks alone.

Mnemonic integration depends in part on retrieval and interactions between the hippocampus and vmPFC (*Zheng et al., 2021*; *Preston and Eichenbaum, 2013*; *Kuhl et al., 2012*; *Zeithamova and Preston, 2010*). Our *a priori* analyses focused on the hippocampus and EC, but we conducted exploratory analyses in to determine whether and how vmPFC would come to represent the virtual environment, given its proposed role as a storage site for integrated memory representations (*Frankland and Bontempi, 2005*; *Takashima et al., 2006*; *Takehara-Nishiuchi and McNaughton, 2008*; see Appendix). In the present work, we did not observe robust effects in vmPFC, but it remains possible that this region may represent features of the environment that were not explicit to our modeling approach.

An emergent body of work is advancing understanding of fundamental mechanisms in the medial temporal lobe and connected cortical structures that build knowledge across events. Part of this work documents pattern separation and pattern differentiation in the hippocampus, which serves to minimize confusion between related memories. At the same time, other theoretical and empirical findings are elucidating the role of these structures in mnemonic integration and cross-event generalization. It has been unclear whether and how these two seemingly opposing mechanisms—differentiation and integration—work together to facilitate the emergence of structured knowledge. Here, we

provide complementary evidence that both differentiation and integration occur concurrently within the hippocampus and can emerge at the same points in learning. Our data suggest that hippocampal differentiation provides a structure necessary for participants to distinguish between nearby locations, while integration processes serve to build a global map of the environment. Moreover, integration relates to individual differences in the ability to efficiently navigate between locations in the global map that have not been previously traversed. Future research promises to reveal whether individual differences in global knowledge building are strategic, relate to differences in other neural systems, and/or are differentially sensitive to dysfunction or disease.

## Methods

### Participants

Thirty-three participants gave informed written consent, in accordance with procedures approved by the Stanford University Institutional Review Board. Nine dropped out before completing the 3-day study (two felt motion sick, two experienced unrelated illness, and five did not want to continue past Day 1), and data from one participant were excluded for failing to respond during the fMRI task. The final sample consisted of 23 right-handed participants (mean age = 22.91 years, SD = 5.03, range 18–35; 14 females) with normal or corrected-to-normal vision and no self-reported history of psychiatric or neurological disorders. Day 3 data were excluded for two of these participants (one due to scanner-related issues that prevented data acquisition, and one who reported falling asleep during scanning). Thus, Global Navigation Task/Day 3 analyses were conducted with 21 participants. Participants received compensation for their participation ($20/hr).

### Paradigm overview

Participants completed an experimental paradigm where they learned to navigate to goal locations within a virtual environment. The 3-day study included two behavioral navigation tasks—the Local Navigation Task and Global Navigation Task—and three fMRI scan sessions (*Figure 1a*) that assayed changes in hippocampal and neocortical representations across learning. The virtual environment contained three oval-shaped tracks of distinct texture and color (*Figure 1b*). Individual tracks contained five goals, initially marked by unique fractal images. Tracks also contained unique landmark buildings that participants could use to guide their navigation, as there were no distal cues in the virtual environment. Landmarks were located near (but not at) goals, with the distance between them varying for each location and participant. Importantly, each track had one landmark in common with each of the other two tracks.

Participants first completed the Local Navigation Task, where they learned to navigate to goals on each of the three tracks separately, with fog serving to hide others from view (at any given point on a track, only one landmark and goal was visible in the field of view). The next day, participants completed the Global Navigation Task, where the separately learned tracks were connected and participants were required to navigate to goals both within and across tracks (*Figure 1b*, right). Participants were not informed and not aware that the tracks would be connected prior to the start of the Global Task, despite the shared landmarks. Efficient navigation on the Global Task required knowledge of the spatial relationships between locations in the global environment both within and across tracks.

### Stimuli

Nineteen building facades were selected from https://www.lughertexture.com and used to create landmarks in the virtual environment. Facades were selected to have a variety of features and be of different architectural styles and colors. After resizing, removing signs and reflections, and adding roofs in Adobe Photoshop (Adobe, Inc), buildings were rendered onto 3D models of equal shape and size. Twelve buildings were randomly selected to be incorporated into each participant's unique environment. Fractal images marked goal locations within the virtual environment. Twenty-three fractals were drawn from the Dryad Digital Repository (*Wilming et al., 2017*; *Wilming, 2017*) and from the stimulus set used in *Brown et al., 2016*. Images were resized and rendered onto 3D models of equal shape and size. Fifteen fractals were randomly selected to be incorporated into each participant's unique environment. Each participant's unique set of building and fractal stimuli were used in both

behavioral navigation tasks and all fMRI sessions. Buildings and fractals were matched for low-level visual features by using the SHINE toolbox in Matlab (version R2015b) to equate their luminance histograms (*Willenbockel et al., 2010*).

All 3D models for the virtual environment were created in Maya (Autodesk, Inc), a 3D graphics program. Textures were acquired via a Google search and https://www.lughertexture.com. While the specific stimuli used for landmarks varied between participants, the positions of the landmarks were fixed in the environment. However, goal locations were selected to fall within a 3 arbitrary unit (a.u.) radius of each landmark (1 a.u. = ~3 m/10 ft); thus, goal locations differed slightly across participants.

## Virtual navigation tasks

The navigation tasks were developed with Panda3D (1.9.4), a python-based open-source gaming engine, and the PandaEPL library (*Solway et al., 2013*). For all navigation tasks, participants were instructed to navigate to goals using the shortest possible path. Each behavioral run contained 10 navigation trials. At the start of each run, participants were placed at a location on the path and slowly rotated 360 degrees to orient themselves (6 s). Trials then proceeded as follows: (1) a fractal cue appeared onscreen (1 s) indicating the goal to which the participant should navigate; (2) participants chose their heading direction and (3) navigated to the cued goal, pressing the spacebar at arrival; (4) feedback appeared onscreen revealing whether the participant was at the correct location and had navigated via the shortest path (2 s); and (5) the camera panned down and a fixation cross appeared (1 s) before the next trial began (*Appendix 1—figure 1*). Participants navigated in the environment by pressing the forward arrow key and adjusted their heading direction using the left and right arrow keys. Movement was fixed along the centerline of the paths and movement speed was held constant. Traveling around one individual path required approximately 30 s.

## Local Navigation Task

During the Local Navigation Task, participants learned to navigate to goals on the individual tracks separately, with the other tracks hidden from view. After fMRI scanning on Day 1 (see fMRI task below), the Local Task began with four runs of learning trials on track 1. On learning trials, fractals were visible at goals to allow for learning of their locations within the track environment. The participants then completed two runs of test trials, where fractals were no longer visible and participants had to rely on memory to navigate successfully. This procedure was repeated for track 2 and track 3. After completing two runs of test trials on each track, participants began additional interleaved blocks of test trials, switching between tracks until six test runs (60 trials) were completed for each of the individual tracks. In total, participants navigated from every goal on a track to every other goal on that track, several times. Track order was randomized and counterbalanced across participants. After Day 1, participants returned 24 hr (range = 21–31.5 hr) later for Day 2, which started with two additional runs of Local Navigation Task test trials on each track prior to fMRI on Day 2. These two runs provided an assay of whether knowledge of the individual tracks had been retained overnight.

## Global Navigation Task

After fMRI on Day 2, participants began the Global Navigation Task. Here, the three tracks were connected and participants were required to navigate to goals both within and across the different tracks. In this task, common landmarks served as linking points within the larger environment (*Figure 1b*). Participants first completed three runs of learning trials, where fractals were again visible at goals. While the goal locations did not change between the Local and Global Navigation Tasks, learning trials allowed participants to become accustomed to moving across tracks and gave them an opportunity to orient themselves to the larger environment. Participants then completed nine runs of test trials, where fractals were no longer visible in the environment. For 30 of 90 test trials, participants were cued to navigate to a goal along the track on which they were already located. For the other 60 test trials, participants were required to navigate to a different track. After Day 2, participants returned 24 hr (range = 15.5–28.5 hr) later for Day 3, which started with four additional runs of Global Navigation test trials prior to fMRI on Day 3. These four runs provided an assay of whether spatial knowledge had been retained overnight.

## Behavioral data analysis

To assay whether participants successfully learned goal locations, we computed percent correct as the ratio of the number of trials where participants ended navigation within 8 a.u. of the virtual goal location (defined as a correct response) over the number of trials attempted. To assay whether participants had learned spatial relationships between the locations in the environment, we computed a path inefficiency metric by dividing path length by the length of the shortest possible path to the goal, subtracting 1, and multiplying by 100 to express as a percent. Thus, a path inefficiency of 0% would indicate that the participant took the shortest path possible, and a path inefficiency of 100% would indicate that their path was twice the length of the shortest possible path to the goal.

## fMRI task

Participants underwent fMRI scanning prior to any learning (Day 1), after the Local Navigation Task (Day 2), and after the Global Navigation Task (Day 3). During each fMRI session, participants viewed images of the 12 landmarks and 15 fractals that were used in their unique virtual environment. Stimuli were presented 12 times per scan session (3 repetitions/run x 4 runs). Within each scan, the stimuli appeared in mini-blocks such that participants saw all 27 images prior to seeing any repeated, with third repeats appearing only after all stimuli had appeared twice. Images were pseudo-randomized within each mini-block such that the same image could not appear within three steps of itself at the block transitions. Moreover, no first order stimulus-to-stimulus transitions were repeated within a scan session.

On each trial, the stimulus appeared on a gray background for 1.8 s, followed by a fixation cross of 5.4 s (intertrial interval; *Figure 1c*). To ensure that participants paid close attention to the images, they performed a visual anomaly detection task in which they were asked to report whether a feature of an image was 'bleached out' on trials when a red fixation cross appeared after image offset (*Clarke et al., 2016*). Only a small proportion of stimulus presentations were response trials (~8%). The probability of a response trial having a bleached feature was 0.5. Response trials were excluded from all further analyses.

## MR data acquisition

Whole-brain imaging data were acquired on a 3T GE Discovery MR750 MRI scanner (GE Healthcare) using a 32-channel radiofrequency receive-only head coil (Nova Medical). Functional data were acquired using a three-band echo planar imaging (EPI) sequence (acceleration factor = 2) consisting of 63 oblique axial slices parallel to the long axis of the hippocampus (TR = 1.8 s, TE = 30ms, flip angle = 75°, FOV = 220mm × 220mm, voxel size = 2 × 2 × 2 mm$^3$). To correct for distortions of the B0 field that may occur with EPI, two B0 field maps were acquired before every functional run, one in each phase encoding direction, with the same slice prescription as the functional runs. Structural images were acquired using a T1-weighted (T1w) spoiled gradient recalled echo structural sequence (186 sagittal slices, TR = 7.26ms, FoV = 230mm × 230mm, voxel size = 0.9 × 0.9 × 0.9 mm$^3$). The MR data collection techniques closely mirrored procedures in *Jiang et al., 2020*.

## Anatomical data preprocessing

fMRI data preprocessing was performed with fMRIPrep 1.5.3rc2 (RRID:SCR_016216) (*Esteban et al., 2019*; *Esteban, 2022*), which is based on *Nipype* 1.3.1 (RRID:SCR_002502) (*Gorgolewski et al., 2011*; *Esteban et al., 2022*). The T1-weighted (T1w) structural images were corrected for intensity non-uniformity (INU) with N4BiasFieldCorrection (*Tustison et al., 2010*), distributed with ANTs 2.2.0 (RRID:SCR_004757) (*Avants et al., 2008*), and used as the T1w-reference throughout the workflow. The T1w-reference was then skull-stripped with a *Nipype* implementation of the ants-BrainExtraction.sh workflow (from ANTs), using OASIS30ANTs as the target template. Brain tissue segmentation of cerebrospinal fluid (CSF), white-matter (WM) and gray-matter (GM) was performed on the brain-extracted T1w-reference using FAST (FSL 5.0.9, RRID:SCR_002823; *Zhang et al., 2001*). Brain surfaces were reconstructed using recon-all (FreeSurfer 6.0.1, RRID:SCR_001847; *Dale et al., 1999*), and the brain mask estimated previously was refined with a custom variation of the method to reconcile ANTs-derived and FreeSurfer-derived segmentations of the cortical gray-matter of Mindboggle (RRID:SCR_002438; *Klein et al., 2017*). Volume-based spatial normalization to standardized space (MNI152NLin2009cAsym) was performed through nonlinear registration with antsRegistration

(ANTs 2.2.0), using brain-extracted versions of both the T1w-reference and T1w-template. The ICBM 152 Nonlinear Asymmetrical template version 2009 (RRID:SCR_008796; TemplateFlow ID:MNI152N-Lin2009cAsym; *Fonov et al., 2009*) was selected for spatial normalization.

## Functional data preprocessing

For each of the 12 functional runs per participant (4 per scan session), the following preprocessing was performed. First, a reference volume and its skull-stripped version were generated using a custom methodology of fMRIPrep. A B0-nonuniformity map (or fieldmap) was estimated based on the two EPI references with opposing phase-encoding directions, with 3dQwarp (AFNI) (*Cox and Hyde, 1997*). Based on the estimated susceptibility distortion, a corrected EPI reference was calculated for a more accurate co-registration with the anatomical reference. The BOLD reference was then co-registered to the T1w-reference using bbregister (FreeSurfer) which implements boundary-based registration (*Greve and Fischl, 2009*). Co-registration was configured with six degrees of freedom. Head-motion parameters with respect to the BOLD reference (transformation matrices and six corresponding rotation and translation parameters) were estimated before any spatiotemporal filtering using MCFLIRT (FSL 5.0.9) (*Jenkinson et al., 2002*). The BOLD time-series were resampled onto their original, native space by applying a single, composite transform to correct for head-motion and susceptibility distortions. These resampled BOLD time-series will be referred to as preprocessed BOLD in original space, or just preprocessed BOLD. The BOLD time-series were resampled into standard space, generating a preprocessed BOLD run in MNI space. Several confounding time-series were calculated based on the preprocessed BOLD: framewise displacement (FD), DVARS and a set of low-frequency regressors for temporal high-pass filtering. FD and DVARS were calculated for each functional run, both using their implementations in Nipype (following the definitions by *Power et al., 2014*). The head-motion estimates calculated in the correction step were also placed within the corresponding confounds file. All resamplings were performed with a single interpolation step by composing all the pertinent transformations (i.e., head-motion transform matrices, susceptibility distortion correction when available, and co-registrations to anatomical and output spaces). Gridded (volumetric) resamplings were performed using antsApplyTransforms (ANTs), configured with Lanczos interpolation to minimize the smoothing effects of other kernels (*Lanczos, 1964*). The preprocessed fMRI data were smoothed by a 2 mm full-width-half-maximum Gaussian kernel.

## fMRI analysis

Prior to fMRI analyses, we removed the first 6 TRs in each run. For each scan session, we built general linear models (GLMs) for even and odd runs, which were regressed against preprocessed fMRI data at the voxel level. To obtain stimulus-level beta estimates for brain activity, each stimulus (i.e., landmark or fractal) was represented by a single regressor, time-locked to when it appeared onscreen in odd or in even runs. Each event was modeled as an epoch lasting for the duration of stimulus presentation (1.8 s) and convolved with a canonical hemodynamic response function. A separate regressor was included for response trials to exclude them from further analysis. Response trials were modeled as epochs lasting for the duration of the stimulus presentation and subsequent response period (7.2 s). GLMs also included nuisance regressors marking outlier TRs (DVARS > 5 or FD > 0.9 mm from previous TR), a run regressor, and regressors generated by fMRIPREP representing TR-level six-dimensional head movement estimates, framewise displacement (mm), and low frequency components for temporal high-pass filtering. GLMs yielded estimates of voxel-level brain activity for each stimulus during odd and even runs of each scan session. Modeling was performed with SPM12 (Wellcome Trust Centre for Neuroimaging) and custom Matlab (vR2017b) routines.

Primary analyses were performed using an *a priori* region-of-interest (ROI) approach targeting the hippocampus and EC. Bilateral hippocampal, entorhinal, parahippocampal, and perirhinal cortical ROIs were manually delineated on each participant's high-resolution T1-weighted structural image using established procedures (*Olsen et al., 2009*). Due to low TSNR (<28), selected scans from 3 participants were excluded from analyses in right EC, and 1 scan from 1 participant was excluded from analyses in left EC. For exploratory analyses in vmPFC, we obtained a vmPFC mask from a Neurosynth parcellation (*Chang et al., 2021*), and transformed the mask into native space for each participant. We also created a smaller 8 mm spherical vmPFC mask using the peak voxel reported in *Takashima et al., 2006*. ROIs were resampled, masked to exclude voxels outside the brain, and aligned with functional

volumes. Finally, we used a visual ROI defined by FreeSurfer's automated segmentation procedure as a control for our analyses. The mask for this ROI was defined as a conjunction of FreeSurfer's pericalcarine and calcarine sulcus regions in both hemispheres.

The activity pattern for each stimulus was quantified as a vector of multi-voxel normalized betas by dividing the original betas by the square root of the covariance matrix of the error terms from the GLM estimation (*Walther et al., 2016*). Separately for each participant, ROI, and scan session, we computed pattern similarity by correlating activity patterns between even and odd runs. Pattern similarity analyses were conducted in Python (v2.7).

## Statistical analyses

No explicit power analysis was conducted to predetermine sample size, but we aimed to collect 30 participants for the multi-day study. The present sample size (n = 23) is comparable to that of prior fMRI studies examining mnemonic representations that did not scan across three consecutive days (e.g. *Chanales et al., 2017*; *Deuker et al., 2016*; *Tompary and Davachi, 2017*; *Zheng et al., 2021*).

Statistical analyses were implemented in R (v3.6.3). For neural analyses described above, we implemented linear mixed-effects models using the lme4 and lmerTest statistical packages (*Bates et al., 2015*; *Kuznetsova et al., 2017*). Correlations, our dependent variable, were Fisher transformed to follow a normal distribution. Models included fixed effects of interest (i.e., scan session and link distance), fixed effects corresponding to the average univariate activation for each of the stimuli, fixed effects estimating perceptual similarity in V1 and IT cortex, and a standard set of random effects, including a random intercept modeling the mean subject-specific outcome value, as well as random slope terms modeling subject-specific effects of independent variables of interest (i.e., scan session and average univariate activation). We included average univariate activation as a regressor to limit the possibility that overall activation differences within an ROI would impact pattern similarity (*Ritchey et al., 2013*).

To control for perceptual similarity between task stimuli, we input the stimuli into vNet, a deep neural network model. vNet was trained on ecoset, a large-scale image set containing images from 565 basic level categories (*Mehrer et al., 2021*). We extracted hidden layer representations putatively corresponding to V1 and IT cortex (layers 1 and 10, respectively), computed similarity matrices, and included these estimates as regressors in linear mixed effects models. Open-source code is available at https://codeocean.com/capsule/9570390/tree/v1.

For all mixed-effects models, the standard set of random effects was chosen by taking a maximal model with all random effects indicated by the experiment setup, then incrementally removing effects and testing the nested model fits using the likelihood ratio test. This procedure was performed for all ROIs. The final standard set of random effects was composed of the minimum necessary effects to achieve the best fit in all regions, and included a random intercept modeling the mean subject-specific outcome value, as well as random slope terms modeling subject-specific effects of independent variables of interest (i.e., scan session and average univariate activation). Scan session was dummy-coded with Pre-Learning (Day 1) as the baseline. Stimulus type and context were sum-coded with the contrasts of landmark > fractal and same track > different tracks, respectively. For the trial-level model relating performance to hippocampal pattern similarity and distance, separate models were run for landmarks and fractals. Models predicted path inefficiency for each trial in the first four test runs of the Global Navigation Task (Day 2). Fixed effects of interest included the length of the optimal path for that trial, the estimated hippocampal pattern similarity for the pair of fractal goals or nearby landmarks associated with the start and end locations the trial, and a regressor indicating whether the trial was a within- or across-track trial. Models were estimated using a restricted maximum likelihood (REML) approach. Model convergence issues were resolved by changing from the default optimizer to 'bobyqa' and not having the model calculate derivatives.

To correct for multiple comparisons, we first computed *m*, a number representing the number of tests conducted for a given hypothesis multiplied by the number of ROIs we examined (including the visual control). For example, to examine learning-driven changes in pattern similarity for items located at different distances on the same track, we tested for two interactions: distance and scan (Day 2 > Day 1), and distance and scan (Day 3 > Day 1). ROIs tested were hippocampus, left and right EC, vmPFC, and the visual control region (Calcarine). *m* thus represented 2 interactions * 5 ROIs = 10. We then controlled for the false discovery rate (FDR) by ordering the p-values for each hypothesis test

from smallest to largest (P(min)...P(max)), and checking if the following was satisfied for each ordered p-value (*Benjamini and Hochberg, 1995*):

$$P(i) \leq \alpha \times \frac{i}{m}, \; where \; \alpha = 0.05$$

Unless otherwise specified, all p-values reported were uncorrected and we interpret *a priori* predicted effects at this level. For completeness, we also note whether the reported effects survived FDR correction throughout the text and in the Appendix tables. Cohen's *d* effect sizes were computed for t-values as $d = t/\sqrt{N}$ .

## Data availability

Raw fMRI data are available on OpenNeuro. Analytical code and behavioral data for reproducing analyses, results, and figures shown in the paper are available at https://github.com/coreyfernandez/RID, (copy archived at swh:1:rev:f4a2d4915f1922c8a74f1f1a86469cf13789abb5; *Fernandez, 2023*). Further information and requests for resources should be directed to and will be fulfilled by the Lead Contact, Corey Fernandez (coreyf@stanford.edu).

## Acknowledgements

This project was supported by The Marcus and Amalia Wallenberg Foundation and The Stanford Center for Cognitive and Neurobiological Imaging.

## Additional information

### Funding

| Funder | Grant reference number | Author |
| --- | --- | --- |
| The Marcus and Amalia Wallenberg Foundation | MAW 2015.0043 | Anthony D Wagner |
| The Stanford Center for Cognitive and Neurobiological Imaging | | Anthony D Wagner |

The funders had no role in study design, data collection and interpretation, or the decision to submit the work for publication.

### Author contributions

Corey Fernandez, Conceptualization, Data curation, Software, Formal analysis, Investigation, Visualization, Methodology, Writing - original draft, Writing - review and editing; Jiefeng Jiang, Conceptualization, Software, Formal analysis, Methodology, Writing - review and editing; Shao-Fang Wang, Formal analysis, Writing - review and editing; Hannah Lee Choi, Data curation, Investigation; Anthony D Wagner, Conceptualization, Resources, Supervision, Funding acquisition, Writing - review and editing

### Author ORCIDs

Corey Fernandez (ID) http://orcid.org/0000-0003-3901-5552
Hannah Lee Choi (ID) http://orcid.org/0000-0003-0556-8351
Anthony D Wagner (ID) http://orcid.org/0000-0003-0624-4543

### Ethics

Human subjects: All participants provided written informed consent in accordance with a protocol approved by the Stanford Institutional Review Board (IRB #13032).

### Decision letter and Author response

Decision letter https://doi.org/10.7554/eLife.80281.sa1
Author response https://doi.org/10.7554/eLife.80281.sa2

## Additional files

### Supplementary files
• MDAR checklist

### Data availability
Raw fMRI data is available at https://doi.org/10.18112/openneuro.ds004406.v1.0.0. Analytical code and behavioral data for reproducing analyses, results, and figures shown in the paper are available at https://github.com/coreyfernandez/RID, (copy archived at swh:1:rev:f4a2d4915f1922c8a74f-1f1a86469cf13789abb5). Further information and requests for resources should be directed to and will be fulfilled by the Lead Contact, Corey Fernandez (coreyf@stanford.edu).

The following previously published dataset was used:

| Author(s) | Year | Dataset title | Dataset URL | Database and Identifier |
|-----------|------|---------------|-------------|-------------------------|
| Fernandez C, Jiang J, Wang S-F, Choi HL, Wagner AD | 2023 | Representational integration and differentiation in the human hippocampus following goal-directed navigation | https://doi.org/10.18112/openneuro.ds004406.v1.0.0 | OpenNeuro, 10.18112/openneuro.ds004406.v1.0.0 |

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

## Appendix 1

### Hippocampal-vmPFC interactions

The building of structured knowledge is not solely the province of the hippocampus. Retrieval-mediated changes in mnemonic representations involve interactions between the hippocampus and ventromedial prefrontal cortex (vmPFC; *Tompary and Davachi, 2017*; *Zeithamova et al., 2012*; *Kuhl et al., 2012*). Functional coupling between these regions during learning supports the initial formation of structured knowledge and predicts subsequent memory (*Preston and Eichenbaum, 2013*; *Tse et al., 2007*; *Tse et al., 2011*; *Ranganath et al., 2005*; *Siapas et al., 2005*). Over time, vmPFC abstracts and represents commonalities across episodes (*Tompary and Davachi, 2017*; *Zeithamova and Preston, 2010*; *Kumaran et al., 2009*), leading some to propose that structured representations are stored in this region (*Zheng et al., 2021*; *Frankland and Bontempi, 2005*; *Takashima et al., 2006*; *Takehara-Nishiuchi and McNaughton, 2008*). Given the learning-related changes observed in the hippocampus, we conducted an exploratory analysis in vmPFC, hypothesizing that vmPFC would demonstrate a similar representational similarity structure. To test this, we first asked whether vmPFC pattern similarity for stimuli located on the same vs. different tracks significantly differed from that in the hippocampus by running a complete model that included data from both regions. Here, we found no interactions between region, scan, and context (*Appendix 1—table 29*), indicating that the similarity structure was not significantly different between regions. By contrast, pattern similarity in both the hippocampus and vmPFC differed from that of a visual control region (see Control Analyses, *Appendix 1—table 30*). However, models predicting pattern similarity for stimuli located on the same vs. different tracks in vmPFC revealed no interactions between context and scan (*Appendix 1—figure 5*; *Appendix 1—table 31*).

Next, we asked whether vmPFC pattern similarity for landmarks at different distances significantly differed from the hippocampus by running a complete model that included data from both regions. Here, we found no effect of region or interactions between region, scan, and distance (*Appendix 1—table 32*), indicating that the similarity structure was not significantly different between regions. By contrast, pattern similarity in both the hippocampus and vmPFC differed from that of a visual control region (see Control Analyses; *Appendix 1—table 33*). However, in contrast to hippocampus, models predicting pattern similarity for landmarks at different link distances in vmPFC revealed no interactions between link distance and scan for within-track (*Appendix 1—figure 6a*; *Appendix 1—table 34*) or across-track landmarks (*Appendix 1—figure 6b*; *Appendix 1—table 35*). We observed no significant interactions when similar models were run for fractal stimuli (*Appendix 1—Tables 36 and 37*).

While context- and distance-related effects on vmPFC and hippocampal pattern similarity did not significantly differ, such effects were significant in hippocampus but not in vmPFC. To further test whether distance-related similarity structures were similar between the regions, we modeled the relationship between vmPFC pattern similarity and hippocampal pattern similarity. Specifically, we ran a linear mixed-effects model predicting pairwise similarity values in vmPFC, with scan session, pairwise similarity values in the hippocampus (averaged across hemispheres) and pairwise similarity values in a visual control region as predictors. Here, we observed significant interactions between scan session and pairwise similarity values in the hippocampus Post Local (Day 2 > Day 1 × hippocampal pattern similarity, $\beta = -0.094 \pm 0.017$; $t = -5.713$; $p < 1.12e^{-8}$, survived FDR correction; $d = 1.25$) and Global Navigation (Day 3 > Day 1 × hippocampal pattern similarity, $\beta = -0.076 \pm 0.017$; $t = -4.355$; $p < 1.35e^{-5}$, survived FDR correction; $d = 0.95$), but not in the visual control region (Day 2 > Day 1 × calcarine pattern similarity, $\beta = 0.019 \pm 0.013$; $t = 1.427$; $p = 0.154$; Day 3 > Day 1 × calcarine pattern similarity, $\beta = -0.020 \pm 0.014$; $t = -1.433$; $p = 0.152$; *Appendix 1—table 38*). This pattern of findings suggests that functional connectivity between the hippocampus and vmPFC weakens over time.

### Control analyses

All linear mixed-effects models were fit to data from a visual control region defined as a conjunction of FreeSurfer's pericalcarine and calcarine sulcus regions in both hemispheres.

We first tested whether context effects differed between the visual control region, the hippocampus, EC, and vmPFC by running a complete model predicting neural pattern similarity, with scan session (Pre-Learning/Day 1, Post Local Navigation/Day 2, and Post Global Navigation/Day 3), context (same path and different paths), and region (calcarine, hippocampus, EC, and vmPFC) as predictors. Region was dummy-coded with the visual control region (calcarine) serving as the

baseline. Here we found main effects (hippocampus: $\beta$ = –0.50 ± 0.003, t = –176.735, p < 2e$^{-16}$, survived FDR correction, d = 38.51; EC: $\beta$ = –0.513 ± 0.003, t = –181.929, p < 2e$^{-16}$, survived FDR correction, d = 39.70; vmPFC: $\beta$ = –0.480 ± 0.003, t = –144.553, p < 2e$^{-16}$, survived FDR correction, d = 31.54) and region × scan session interactions for all regions (hippocampus × Day 2 > Day 1, $\beta$ = 0.027 ± 0.004, t = 7.233, p < 4.75e$^{-13}$, survived FDR correction, d = 1.51; hippocampus × Day 3 > Day 1, $\beta$ = 0.030 ± 0.004, t = 7.673, p < 1.69e$^{-14}$, survived FDR correction, d = 1.67; EC × Day 2 > Day 1, $\beta$ = 0.011 ± 0.004, t = 3.024, p = 0.003, survived FDR correction, d = 0.63; EC × Day 3 > Day 1, $\beta$ = 0.039 ± 0.004, t = 9.957, p < 2e$^{-16}$, survived FDR correction, d = 2.17; vmPFC × Day 2 > Day 1, $\beta$ = 0.030 ± 0.004, t = 7.029, p < 2.10e$^{-12}$, survived FDR correction, d = 1.47; vmPFC × Day 3 > Day 1, $\beta$ = 0.012 ± 0.004, t = 2.633, p = 0.008, d = 0.57; *Appendix 1—table 30*).

We then fit a linear mixed-effects model predicting neural pattern similarity to data from the visual control region. Scan session (Pre-Learning/Day 1, Post Local Navigation/Day 2, and Post Global Navigation/Day 3), stimulus type (landmarks and fractals), and context (same track and different tracks) were included as predictors. When the context model was fit to data from the control region, we found no interactions between context and scan session (Day 2 > Day 1 × context, $\beta$ = –0.003 ± 0.005; t = –0.717; p = 0.474; Day 3 > Day 1 × context, $\beta$ = –0.002 ± 0.005; t = –0.416; p = 0.678) or context, stimulus type, and scan session (Day 2 > Day 1 × stimulus type × context, $\beta$ = –0.004 ± 0.009; t = –0.377; p = 0.706; Day 3 > Day 1 × stimulus type × context, $\beta$ = 0.014 ± 0.01; t = 1.509; p = 0.131; *Appendix 1—figure 7*; *Appendix 1—table 39*).

We next tested whether distance effects differed between the visual control region, the hippocampus, and vmPFC by running a complete model predicting neural pattern similarity for landmarks, with scan session (Pre-Learning/Day 1, Post Local Navigation/Day 2, and Post Global Navigation/Day 3), link distance, and region (calcarine, hippocampus, and vmPFC) as predictors. Region was dummy-coded with the visual control region (calcarine) serving as the baseline. Here we found main effects (hippocampus: $\beta$ = –0.506 ± 0.011, t = –46.889, p < 2e$^{-16}$, survived FDR correction, d = 9.78; vmPFC: $\beta$ = –0.483 ± 0.013, t = –38.294, p < 2e$^{-16}$, survived FDR correction, d = 7.98) and region × scan session interactions for both regions (hippocampus × Day 2 > Day 1, $\beta$ = 0.077 ± 0.015, t = 5.102, p < 3.38e$^{-7}$, survived FDR correction, d = 1.06, hippocampus × Day 3 > Day 1, $\beta$ = 0.047 ± 0.015, t = 3.079, p = 0.002, survived FDR correction, d = 0.67; vmPFC × Day 2 > Day 1, $\beta$ = 0.089 ± 0.017, t = 5.129, p < 2.92e$^{-7}$, survived FDR correction, d = 1.07; *Appendix 1—table 33*).

We fit a linear mixed-effects model predicting neural pattern similarity between landmarks on the same track to data from the visual control region. Scan session (Pre-Learning/Day 1, Post Local Navigation/Day 2, and Post Global Navigation/Day 3) and link distance (1 and 2) were included as predictors. When the local distance model was fit to data from the control region, we found no interactions between link distance and scan session (Day 2 > Day 1 × distance, $\beta$ = 0.007 ± 0.022; t = 0.304; p = 0.761; Day 3 > Day 1 × distance, $\beta$ = –0.035 ± 0.022; t = –1.57; p = 0.117, *Appendix 1—figure 8a*; *Appendix 1—table 40*). Finally, we fit a linear mixed-effects model predicting neural pattern similarity between landmarks on different tracks to data from the visual control region. Scan session (Pre-Learning/Day 1, Post Local Navigation/Day 2, and Post Global Navigation/Day 3) and link distance (2, 3, and 4) were included as predictors. When this distance model was fit to data from the control region, we found no interactions between link distance and scan session (Day 2 > Day 1 × distance, $\beta$ = –0.003 ± 0.01; t = –0.261; p > 0.794; Day 3 > Day 1 × distance, $\beta$ = 0.007 ± 0.01; t = 0.727; p > 0.468; *Appendix 1—figure 8b*; *Appendix 1—table 41*). These control analyses indicate that the effects observed in the hippocampus were not observed throughout the brain.

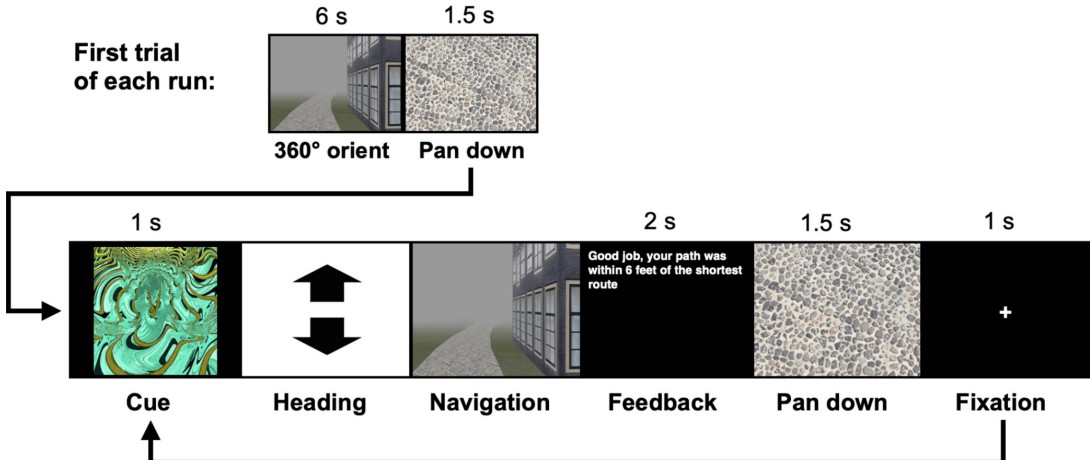

**Appendix 1—figure 1.** Trial structure for all navigation tasks. Each behavioral run contained 10 navigation trials. At the start of each run, participants were placed at a location on the track and rotated 360 degrees (6 s). Trials then proceeded as follows: (1) a fractal cue appeared onscreen (1 s) indicating the goal to which the participant should navigate; (2) participants chose their heading direction and (3) navigated to the cued goal location, pressing the spacebar when they arrived; (4) feedback appeared onscreen revealing whether the participant was at the correct location and whether they had navigated via the shortest path (2 s); and (5) the camera panned down and a fixation cross appeared (1 s) before the next trial began. On learning trials, goal locations were marked by fractal images appearing on the track. On test trials, fractals were not visible on the track and participants had to rely on memory to navigate.

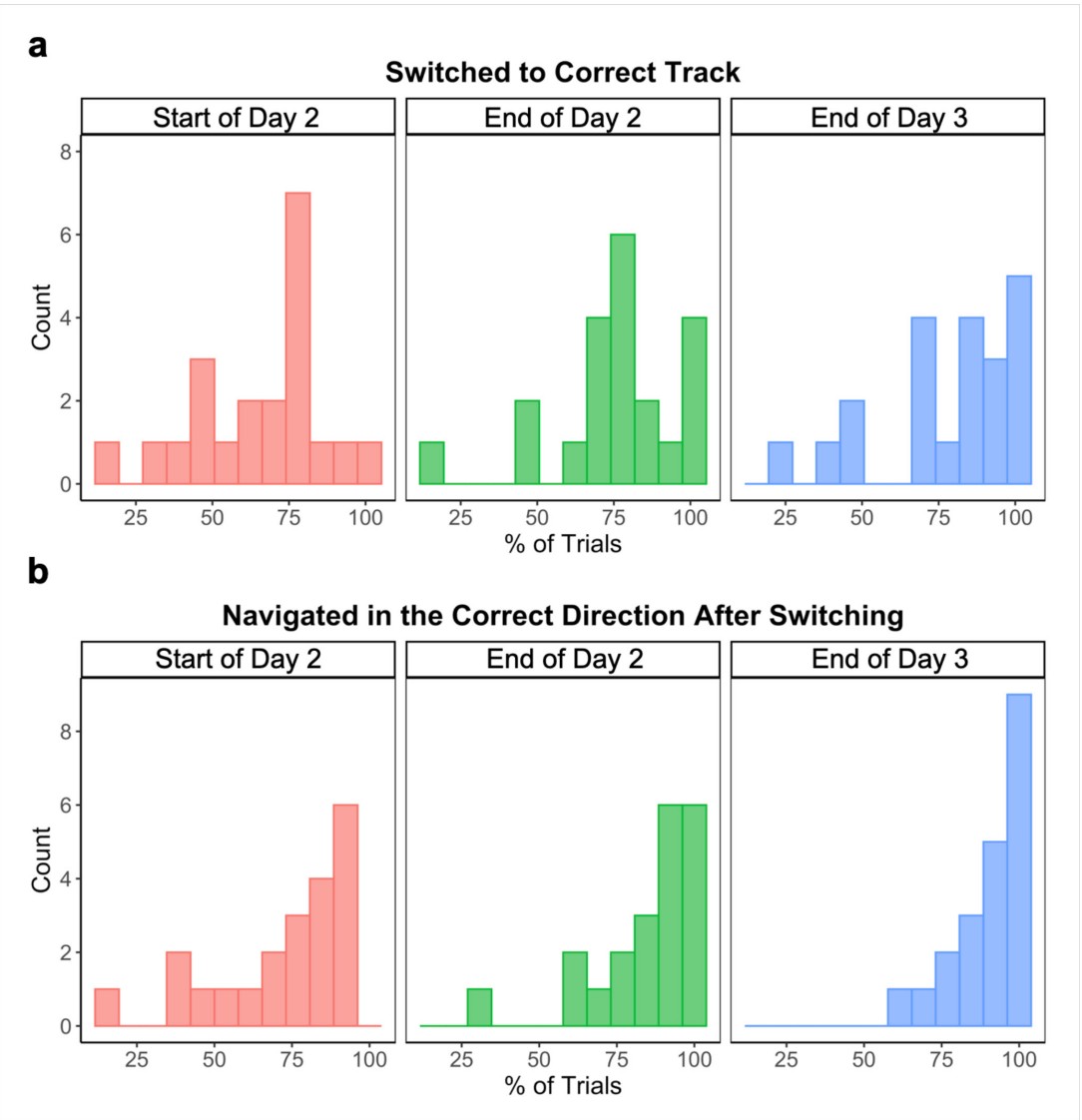

**Appendix 1—figure 2.** Performance improves across the Global Navigation Task. Individual participants' performance on across-track navigation trials at the start (first four test runs; left) and end (last two test runs; center) of the task on Day 2, and at the end (last two test runs; right) of Day 3. (**A**) Percent of across-track trials where participants switched to the correct track. (**B**) Percent of across-track trials where participants navigated in the correct direction once switching (n = 21).

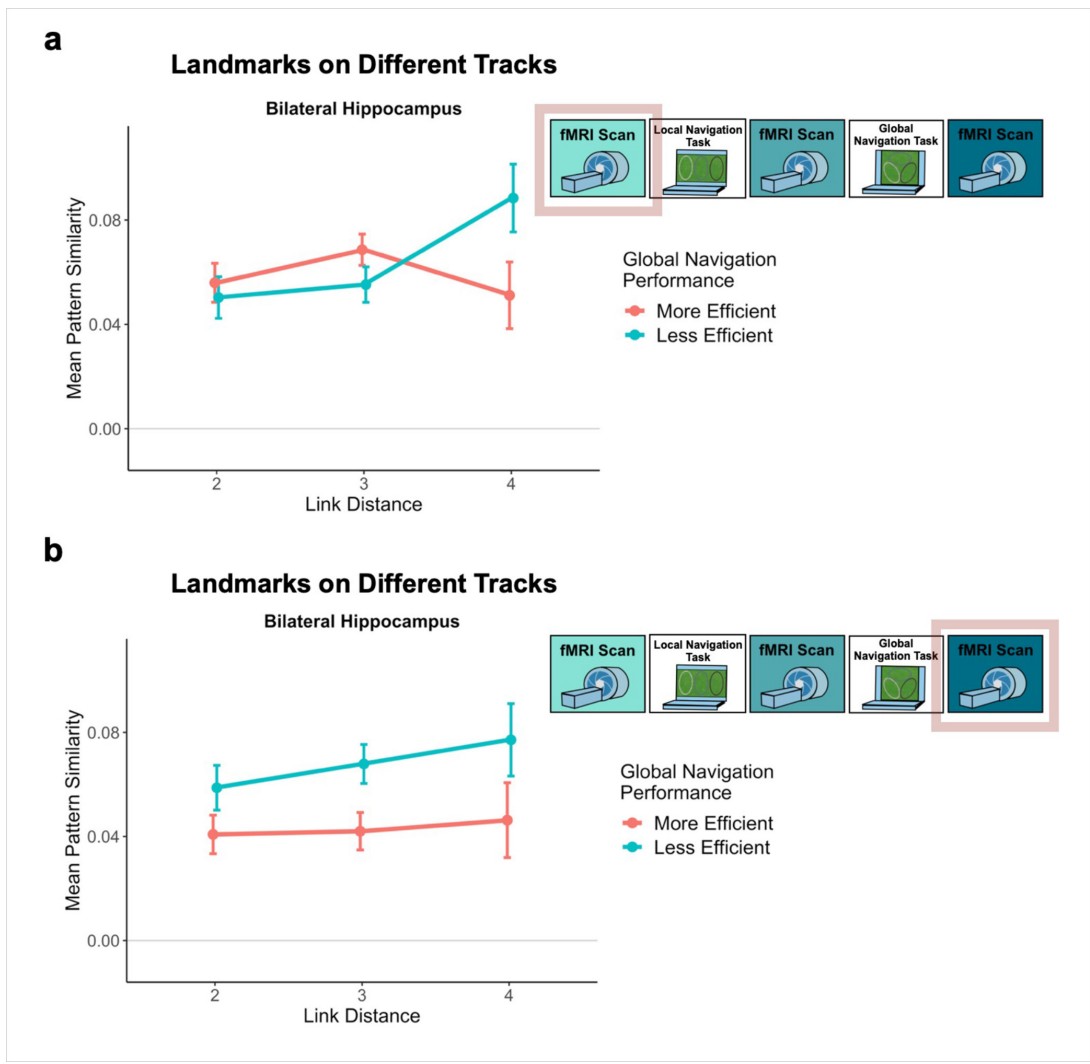

**Appendix 1—figure 3.** Hippocampal pattern similarity for landmark buildings on different tracks. To visualize the relationship between pattern similarity, link distance, and path inefficiency, we split participants into two groups – More Efficient and Less Efficient – based on their median path inefficiency on across-track trials in the first four test runs of the Global Task on Day 2. Pattern similarity relationships did not differ between participants who are more or less efficient on the Global Navigation Task, (**A**) Pre-Learning (Day 1) and (**B**) Post Global Navigation (Day 3). (Error bars denote SE of the estimates. More Efficient, n = 11; Less Efficient, n = 10).

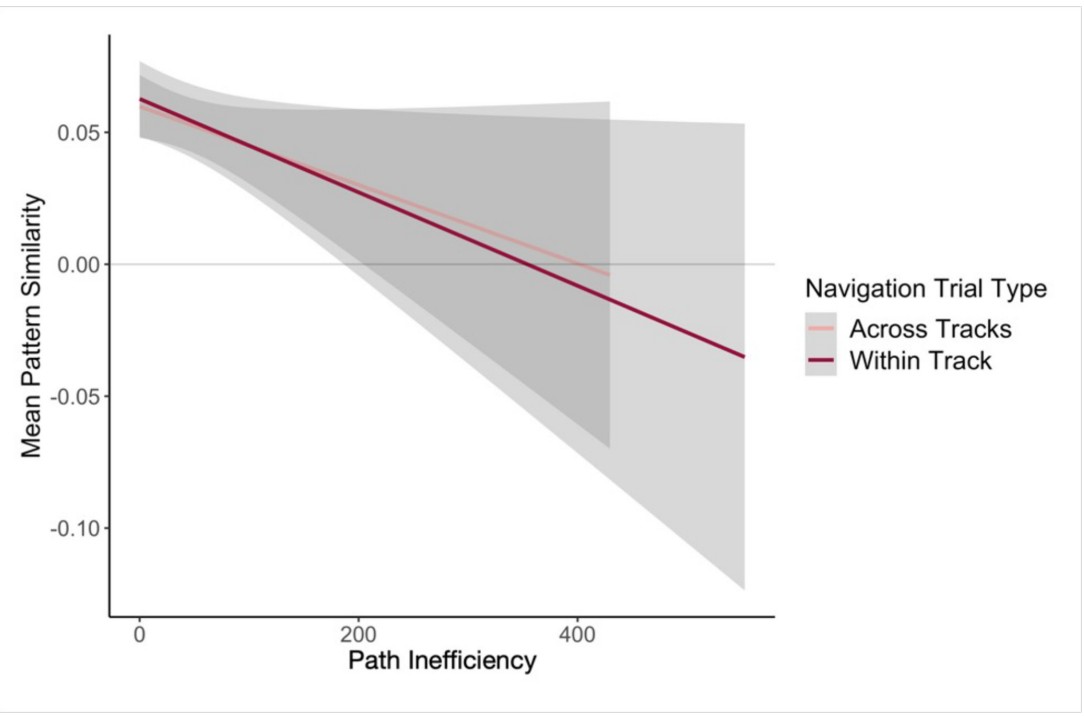

**Appendix 1—figure 4.** Hippocampal pattern similarity (Post Local Navigation) relates to trial-level performance on the subsequent Global Navigation Task. We observed trend-level evidence that greater hippocampal pattern similarity predicted more efficient paths at the start of Global Navigation for both trial types. (Solid lines = estimated linear fit to the data, gray = 95% CI).

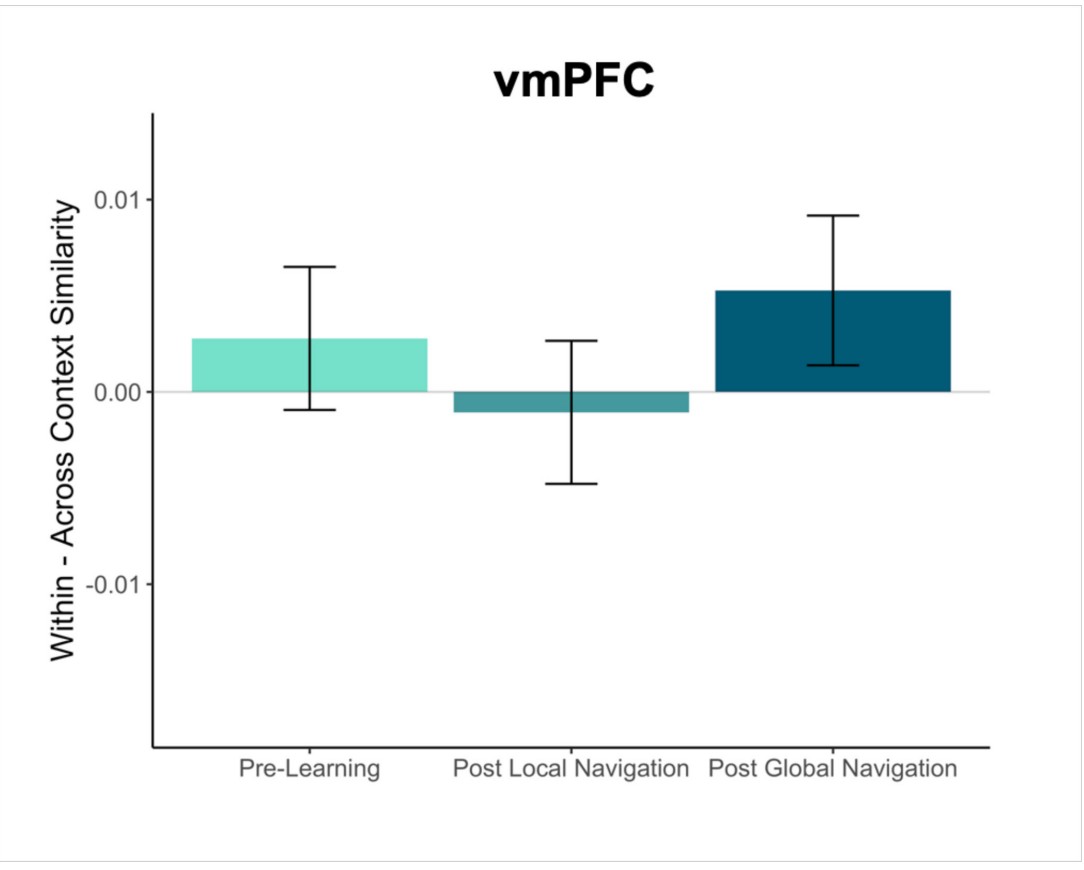

**Appendix 1—figure 5.** Contrast estimates from a model in vmPFC that includes both landmark and fractal stimuli. Interactions between scan session and context were not significant. (Error bars denote SE of the estimates. Day 2 > Day 1, n = 23; Day 3 > Day 1, n = 21).

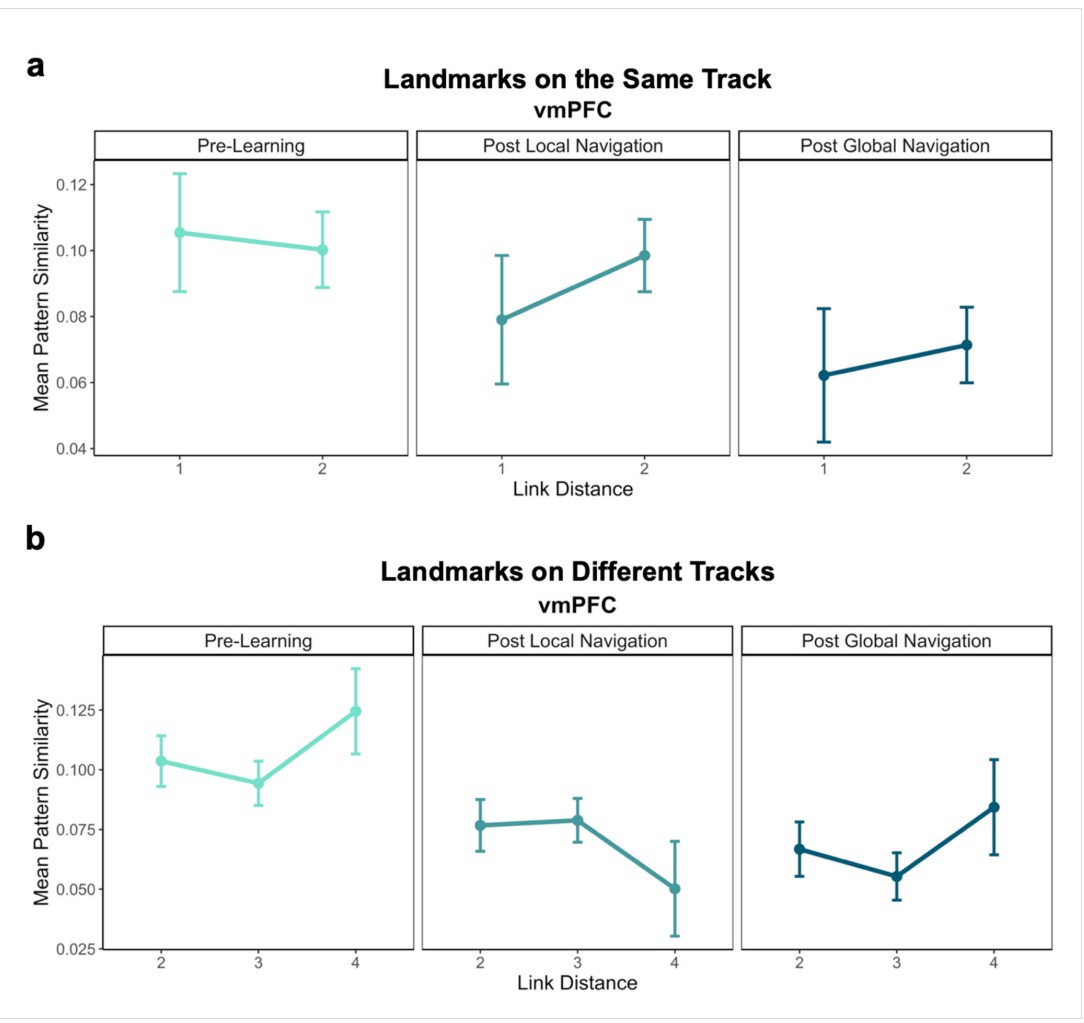

**Appendix 1—figure 6.** Pattern similarity for landmark buildings at different link distances in vmPFC. (**A**) Pattern similarity for landmarks on the same track Pre-Learning (left), after the Local Navigation Task (center), and after the Global Navigation Task (right). Interactions between link distance and scan session were not significant. (**B**) Pattern similarity for landmarks on different tracks. Interactions between link distance and scan session were not significant. (Error bars denote SE of the estimates. Day 2 > Day 1, n = 23; Day 3 > Day 1, n = 21).

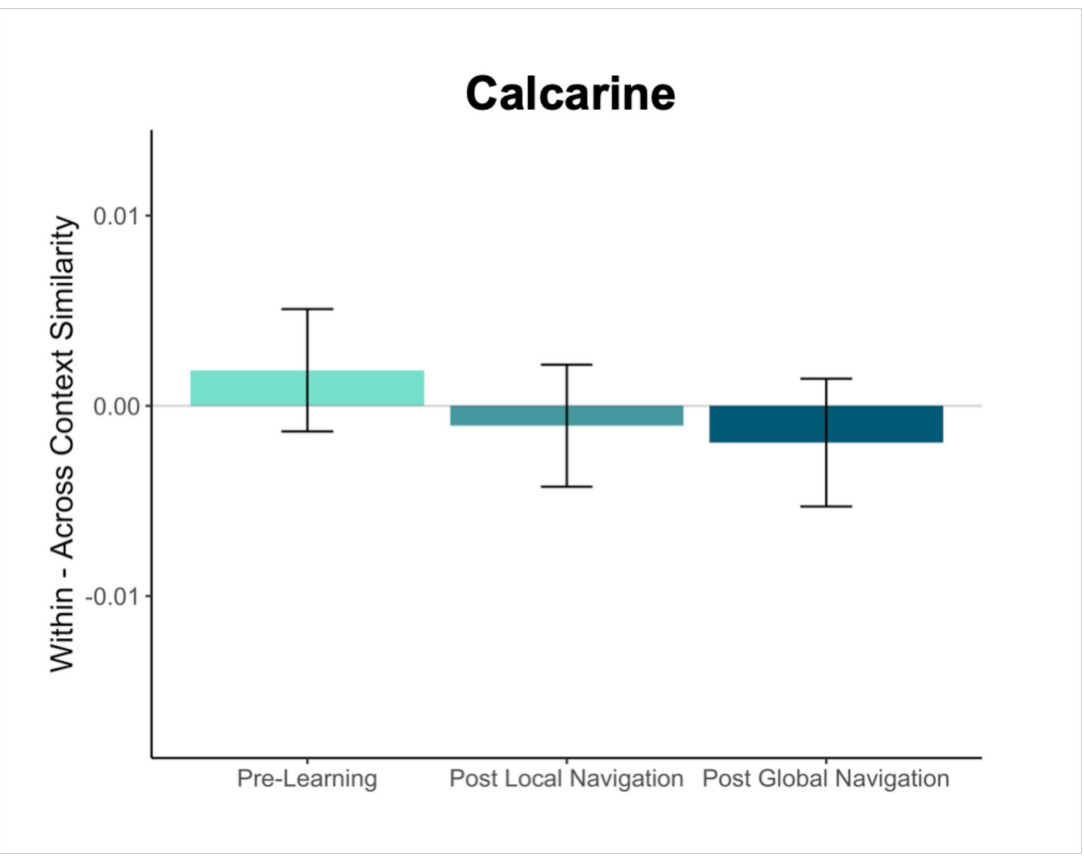

**Appendix 1—figure 7.** Contrast estimates for context models fit to data in a visual region serving as a control (calcarine). Within - across context similarity for landmark buildings. Interactions between scan session and context were not significant. (Error bars denote SE of the estimates. Day 2 > Day 1, n = 23; Day 3 > Day 1, n = 21).

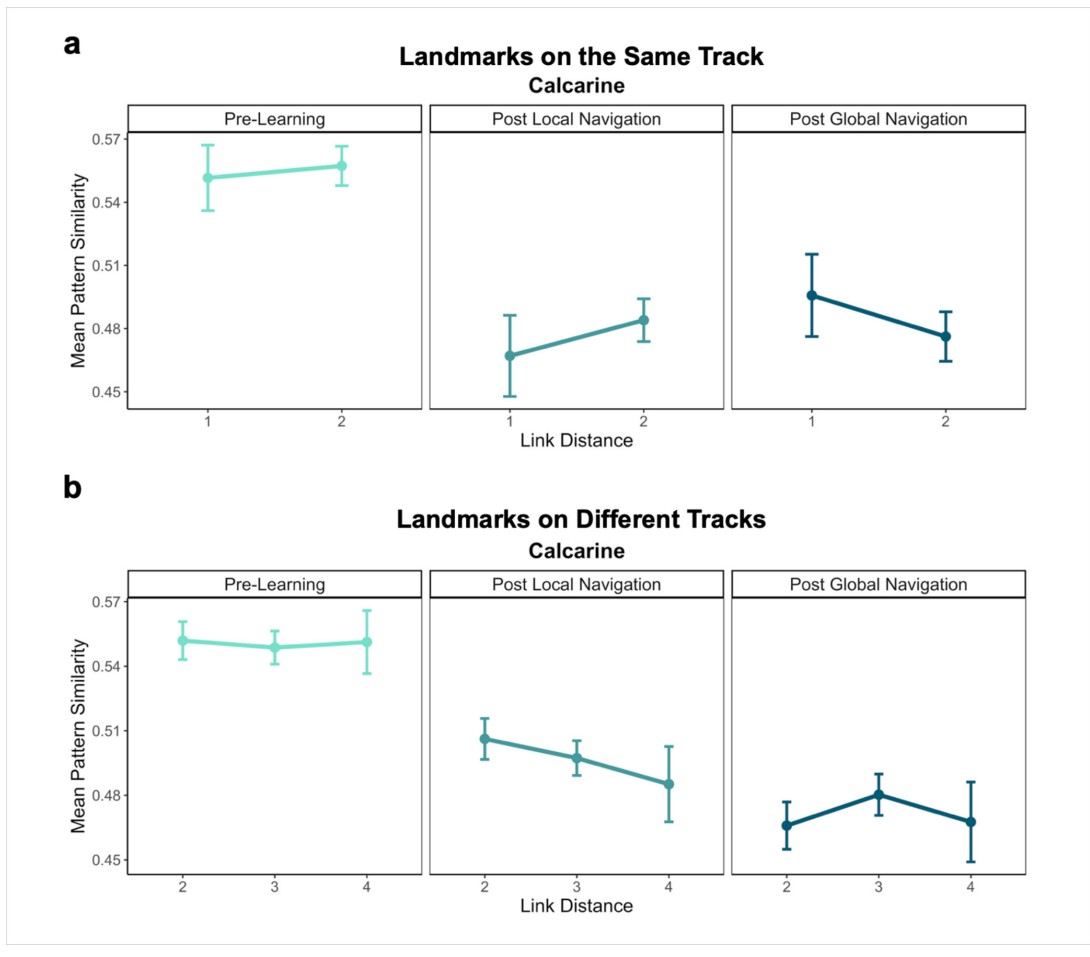

**Appendix 1—figure 8.** Pattern similarity for landmark buildings at different link distances in a visual region serving as a control (calcarine). (**A**) Pattern similarity for landmarks on the same track Pre-Learning (left), after the Local Navigation Task (center), and after the Global Navigation Task (right). Interactions between link distance and scan session were not significant. (**B**) Pattern similarity for landmarks on different tracks. Interactions between link distance and scan session were not significant (Error bars denote SE of the estimates. Day 2 > Day 1, n = 23; Day 3 > Day 1, n = 21).

**Appendix 1—table 1.** Linear mixed-effects model results for an omnibus context model predicting neural pattern similarity, fit to data in the hippocampus.

SE = standard error. To correct for multiple comparisons, we adjusted the α-value for 18 tests. An asterisk (*) denotes findings that survived FDR correction.

| Variable | $\beta$ | SE | $t$ | p |
|---|---|---|---|---|
| (Intercept) | 0.035 | 0.009 | 3.858 | **$8.11e^{-4}$** |
| Day 2 | 0.002 | 0.011 | 0.136 | 0.892 |
| Day 3 | −0.006 | 0.012 | −0.542 | 0.588 |
| stimulus type (landmark > fractal) | 0.007 | 0.003 | 2.422 | **0.015** |
| context (same track > different tracks) | −0.001 | 0.003 | −0.254 | 0.800 |
| hemisphere | 0.006 | 0.002 | 2.756 | **0.006*** |
| Day 2 × stimulus type | −0.003 | 0.004 | −0.715 | 0.475 |
| Day 3 × stimulus type | −0.011 | 0.004 | −2.604 | **0.009*** |
| Day 2 × context | −0.002 | 0.004 | −0.365 | 0.715 |

*Appendix 1—table 1 Continued on next page*

*Appendix 1—table 1 Continued*

| Variable | β | SE | t | p |
|---|---|---|---|---|
| Day 3 × context | –0.001 | 0.004 | –0.334 | 0.739 |
| stimulus type × context | 0.001 | 0.006 | 0.246 | 0.806 |
| Day 2 × hemisphere | –0.011 | 0.003 | –3.477 | **5.08e⁻⁴\*** |
| Day 3 × hemisphere | 0.012 | 0.003 | 3.795 | **1.48e⁻⁴\*** |
| stimulus type × hemisphere | 0.004 | 0.004 | 0.866 | 0.387 |
| context × hemisphere | 0.005 | 0.004 | 1.066 | 0.287 |
| Day 2 × stimulus type × context | –0.002 | 0.009 | –0.216 | 0.829 |
| Day 3 × stimulus type × context | –0.008 | 0.009 | –0.920 | 0.357 |
| Day 2 × stimulus type × hemisphere | –0.002 | 0.006 | –0.320 | 0.749 |
| Day 3 × stimulus type × hemisphere | –0.001 | 0.006 | –0.192 | 0.848 |
| Day 2 × context × hemisphere | –0.005 | 0.006 | –0.899 | 0.369 |
| Day 3 × context × hemisphere | –0.005 | 0.006 | –0.767 | 0.443 |
| stimulus type × context × hemisphere | 0.004 | 0.009 | 0.497 | 0.619 |
| Day 2×stimulus type×context × hemisphere | –0.004 | 0.012 | –0.322 | 0.747 |
| Day 3 × stimulus type × context × hemisphere | –0.006 | 0.012 | –0.511 | 0.610 |

**Appendix 1—table 2.** Linear mixed-effects model results for a context model predicting neural pattern similarity for landmark buildings, fit to data in left hippocampus.
SE = standard error. To correct for multiple comparisons, we adjusted the α-value for 1<u>6</u> tests.

| Variable | β | SE | t | p |
|---|---|---|---|---|
| (Intercept) | 0.051 | 0.049 | 1.043 | 0.297 |
| Day 2 | 0.001 | 0.014 | 0.058 | 0.954 |
| Day 3 | –0.011 | 0.016 | –0.686 | 0.493 |
| context (same track > different tracks) | 0.000 | 0.005 | 0.003 | 0.998 |
| Day 2 × context | –0.003 | 0.007 | –0.400 | 0.689 |
| Day 3 × context | –0.006 | 0.007 | –0.834 | 0.404 |

**Appendix 1—table 3.** Linear mixed-effects model results for a context model predicting neural pattern similarity for landmark buildings, fit to data in right hippocampus.
SE = standard error. No correction for multiple comparisons was applied. To correct for multiple comparisons, we adjusted the α-value for 16 tests. No findings survived FDR correction.

| Variable | β | SE | t | p |
|---|---|---|---|---|
| (Intercept) | 0.062 | 0.049 | 1.260 | 0.208 |
| Day 2 | –0.011 | 0.015 | –0.747 | 0.455 |
| Day 3 | –0.002 | 0.017 | –0.099 | 0.921 |
| context (same track > different tracks) | 0.006 | 0.005 | 1.385 | 0.166 |
| Day 2 × context | –0.010 | 0.007 | –1.502 | 0.133 |
| Day 3 × context | –0.013 | 0.007 | –2.014 | **0.044** |

**Appendix 1—table 4.** Linear mixed-effects model results for a context model predicting neural pattern similarity for fractals, fit to data in left hippocampus.
SE = standard error. To correct for multiple comparisons, we adjusted the α-value for 1<u>6</u> tests.

| Variable | $\beta$ | SE | t | p |
|---|---|---|---|---|
| (Intercept) | 0.029 | 0.030 | 0.945 | 0.345 |
| Day 2 | 0.003 | 0.012 | 0.262 | 0.794 |
| Day 3 | −0.002 | 0.011 | −0.150 | 0.881 |
| context (same track > different tracks) | −0.001 | 0.004 | −0.363 | 0.717 |
| Day 2 × context | −0.001 | 0.005 | −0.111 | 0.911 |
| Day 3 × context | 0.003 | 0.005 | 0.472 | 0.637 |

**Appendix 1—table 5.** Linear mixed-effects model results for a context model predicting neural pattern similarity for fractals, fit to data in right hippocampus.
SE = standard error. To correct for multiple comparisons, we adjusted the α-value for 16 tests.

| Variable | $\beta$ | SE | t | p |
|---|---|---|---|---|
| (Intercept) | 0.100 | 0.029 | 3.398 | **$6.89e^{-4}$** |
| Day 2 | −0.003 | 0.012 | −0.268 | 0.789 |
| Day 3 | 0.011 | 0.014 | 0.817 | 0.414 |
| context (same track > different tracks) | 0.001 | 0.004 | 0.326 | 0.745 |
| Day 2 × context | −0.004 | 0.005 | −0.823 | 0.411 |
| Day 3 × context | 0.001 | 0.005 | 0.166 | 0.868 |

**Appendix 1—table 6.** Linear mixed-effects model results for an omnibus context model predicting neural pattern similarity, fit to data in EC.
SE = standard error. To correct for multiple comparisons, we adjusted the α-value for 18 tests. An asterisk (*) denotes findings that survived FDR correction.

| Variable | $\beta$ | SE | t | p |
|---|---|---|---|---|
| (Intercept) | 0.049 | 0.017 | 2.912 | **0.004** |
| Day 2 | −0.037 | 0.020 | −1.874 | 0.061 |
| Day 3 | −0.023 | 0.020 | −1.178 | 0.239 |
| stimulus type (landmark > fractal) | 0.008 | 0.005 | 1.607 | 0.108 |
| context (same track > different tracks) | 0.002 | 0.005 | 0.461 | 0.645 |
| hemisphere | −0.022 | 0.004 | −6.060 | **$1.37e^{-9}$\*** |
| Day 2 × stimulus type | 0.010 | 0.007 | 1.457 | 0.145 |
| Day 3 × stimulus type | −0.003 | 0.007 | −0.358 | 0.720 |
| Day 2 × context | −0.014 | 0.007 | −2.046 | **0.041** |
| Day 3 × context | −0.010 | 0.007 | −1.394 | 0.163 |
| stimulus type × context | −0.001 | 0.010 | −0.139 | 0.889 |
| Day 2 × hemisphere | 0.022 | 0.005 | 4.192 | **$2.77e^{-5}$\*** |
| Day 3 × hemisphere | 0.056 | 0.005 | 10.552 | **$<2e^{-16}$\*** |
| stimulus type × hemisphere | 0.000 | 0.007 | 0.036 | 0.971 |
| context × hemisphere | 0.004 | 0.007 | 0.524 | 0.600 |
| Day 2 × stimulus type × context | 0.002 | 0.014 | 0.132 | 0.895 |
| Day 3 × stimulus type × context | −0.005 | 0.014 | −0.387 | 0.699 |
| Day 2 × stimulus type × hemisphere | −0.011 | 0.010 | −1.132 | 0.258 |
| Day 3 × stimulus type × hemisphere | −0.003 | 0.010 | −0.319 | 0.749 |

*Appendix 1—table 6 Continued on next page*

*Appendix 1—table 6 Continued*

| Variable | β | SE | t | p |
|---|---|---|---|---|
| Day 2 × context × hemisphere | 0.003 | 0.010 | 0.263 | 0.793 |
| Day 3 × context × hemisphere | 0.001 | 0.010 | 0.117 | 0.907 |
| stimulus type × context × hemisphere | –0.009 | 0.014 | –0.633 | 0.527 |
| Day 2 × stimulus type × context × hemisphere | –0.001 | 0.020 | –0.050 | 0.960 |
| Day 3 × stimulus type×context × hemisphere | 0.022 | 0.020 | 1.057 | 0.291 |

**Appendix 1—table 7.** Linear mixed-effects model results for a context model predicting neural pattern similarity, fit to data in left EC.
SE = standard error. To correct for multiple comparisons, we adjusted the α-value for 16 tests. No findings survived FDR correction.

| Variable | β | SE | t | p |
|---|---|---|---|---|
| (Intercept) | 0.082 | 0.040 | 2.025 | **0.043** |
| Day 2 | –0.036 | 0.024 | –1.490 | 0.136 |
| Day 3 | –0.026 | 0.021 | –1.223 | 0.221 |
| context (same track > different tracks) | 0.003 | 0.004 | 0.656 | 0.512 |
| Day 2 × context | –0.014 | 0.006 | –2.339 | **0.019** |
| Day 3 × context | –0.009 | 0.006 | –1.457 | 0.145 |

**Appendix 1—table 8.** Linear mixed-effects model results for a context model predicting neural pattern similarity, fit to data in right EC.
SE = standard error. To correct for multiple comparisons, we adjusted the α-value for 16 tests.

| Variable | β | SE | t | p |
|---|---|---|---|---|
| (Intercept) | 0.067 | 0.049 | 1.362 | 0.173 |
| Day 2 | 0.002 | 0.022 | 0.071 | 0.943 |
| Day 3 | 0.044 | 0.024 | 1.836 | 0.066 |
| context (same track > different tracks) | 0.008 | 0.005 | 1.455 | 0.146 |
| Day 2 × context | –0.012 | 0.007 | –1.568 | 0.117 |
| Day 3 × context | –0.011 | 0.008 | –1.404 | 0.160 |

**Appendix 1—table 9.** Linear mixed-effects model results for a context model predicting neural pattern similarity that included EC and hippocampus.
SE = standard error. To correct for multiple comparisons, we adjusted the α-value for 5 tests. An asterisk (*) denotes findings that survived FDR correction.

| Variable | β | SE | t | p |
|---|---|---|---|---|
| (Intercept) | 0.074 | 0.021 | 3.623 | **$3.66e^{-4}$** |
| Day 2 | –0.012 | 0.013 | –0.883 | 0.377 |
| Day 3 | –0.010 | 0.013 | –0.780 | 0.436 |
| context (same track > different tracks) | 0.001 | 0.003 | 0.473 | 0.636 |
| region | –0.015 | 0.002 | –7.360 | **$1.85e^{-13}$*** |
| Day 2 × context | –0.004 | 0.004 | –0.956 | 0.339 |
| Day 3 × context | –0.002 | 0.004 | –0.594 | 0.553 |
| Day 2 × region | –0.017 | 0.003 | –5.757 | **$8.58e^{-9}$*** |

*Appendix 1—table 9 Continued on next page*

*Appendix 1—table 9 Continued*

| Variable | β | SE | t | p |
|---|---|---|---|---|
| Day 3 × region | 0.014 | 0.003 | 4.592 | 4.40e$^{-6}$* |
| context × region | 0.004 | 0.004 | 0.907 | 0.364 |
| Day 2 × context × region | −0.009 | 0.006 | −1.635 | 0.102 |
| Day 3 × context × region | −0.007 | 0.006 | −1.278 | 0.201 |

**Appendix 1—table 10.** Linear mixed-effects model results for an omnibus model of local (within-track) distance, predicting neural pattern similarity for landmark buildings in the hippocampus. SE = standard error. To correct for multiple comparisons, we adjusted the α-value for 5 tests.

| Variable | β | SE | t | p |
|---|---|---|---|---|
| (Intercept) | 0.044 | 0.012 | 3.621 | 9.12e$^{-4}$ |
| Day 2 | −0.016 | 0.015 | −1.041 | 0.298 |
| Day 3 | −0.020 | 0.016 | −1.236 | 0.217 |
| distance (link distance 2 > link distance 1) | −0.014 | 0.015 | −0.974 | 0.330 |
| hemisphere | 0.006 | 0.011 | 0.528 | 0.598 |
| Day 2 × distance | 0.036 | 0.021 | 1.721 | 0.085 |
| Day 3 × distance | 0.033 | 0.021 | 1.520 | 0.129 |
| Day 2 × hemisphere | −0.011 | 0.015 | −0.723 | 0.470 |
| Day 3 × hemisphere | 0.004 | 0.015 | 0.288 | 0.774 |
| distance × hemisphere | −0.003 | 0.021 | −0.124 | 0.901 |
| Day 2 × distance × hemisphere | 0.008 | 0.030 | 0.280 | 0.780 |
| Day 3 × distance × hemisphere | 0.004 | 0.030 | 0.121 | 0.904 |

**Appendix 1—table 11.** Linear mixed-effects model results in a model of local (within-track) distance, predicting neural pattern similarity for landmarks buildings in the hippocampus. SE = standard error. To correct for multiple comparisons, we adjusted the α-value for 10 tests. No results survived FDR correction.

| Variable | β | SE | t | p |
|---|---|---|---|---|
| (Intercept) | −0.037 | 0.088 | −0.419 | 0.675 |
| Day 2 | −0.021 | 0.013 | −1.597 | 0.110 |
| Day 3 | −0.018 | 0.014 | −1.247 | 0.213 |
| distance (link distance 2 > link distance 1) | −0.016 | 0.010 | −1.510 | 0.131 |
| Day 2 × distance | 0.040 | 0.015 | 2.700 | **0.007** |
| Day 3 × distance | 0.034 | 0.015 | 2.259 | **0.024** |

**Appendix 1—table 12.** Linear mixed-effects model results in a model of local (within-track) distance, predicting neural pattern similarity for fractals in the hippocampus. SE = standard error. To correct for multiple comparisons, we adjusted the α-value for 10 tests.

| Variable | β | SE | t | p |
|---|---|---|---|---|
| (Intercept) | 0.038 | 0.047 | 0.802 | 0.422 |
| Day 2 | 0.001 | 0.010 | 0.068 | 0.946 |
| Day 3 | 0.005 | 0.012 | 0.456 | 0.653 |
| distance (link distance 2 > link distance 1) | 0.003 | 0.006 | 0.488 | 0.625 |

*Appendix 1—table 12 Continued on next page*

*Appendix 1—table 12 Continued*

| Variable | β | SE | t | p |
|---|---|---|---|---|
| Day 2 × distance | –0.003 | 0.008 | –0.368 | 0.713 |
| Day 3 × distance | –0.003 | 0.008 | –0.338 | 0.735 |

**Appendix 1—table 13.** Linear mixed-effects model results in an omnibus model of global (across-track) distance, predicting neural pattern similarity for landmark buildings in the hippocampus.
SE = standard error. To correct for multiple comparisons, we adjusted the α-value for 5 tests.

| Variable | β | SE | t | p |
|---|---|---|---|---|
| (Intercept) | 0.033 | 0.016 | 2.023 | **0.043** |
| Day 2 | 0.018 | 0.023 | 0.779 | 0.436 |
| Day 3 | –0.010 | 0.023 | –0.445 | 0.657 |
| distance | 0.002 | 0.007 | 0.369 | 0.712 |
| hemisphere | –0.012 | 0.018 | –0.659 | 0.510 |
| Day 2 × distance | –0.012 | 0.010 | –1.297 | 0.195 |
| Day 3 × distance | 0.000 | 0.010 | 0.044 | 0.965 |
| Day 2 × hemisphere | 0.019 | 0.025 | 0.763 | 0.445 |
| Day 3 × hemisphere | 0.016 | 0.026 | 0.632 | 0.527 |
| distance × hemisphere | 0.011 | 0.010 | 1.159 | 0.246 |
| Day 2 × distance × hemisphere | –0.015 | 0.013 | –1.082 | 0.279 |
| Day 3 × distance × hemisphere | –0.007 | 0.014 | –0.485 | 0.628 |

**Appendix 1—table 14.** Linear mixed-effects model results in a model of global (across-track) distance, predicting neural pattern similarity for landmark buildings in the hippocampus.
SE = standard error. To correct for multiple comparisons, we adjusted the α-value for 8 tests. An asterisk (*) denotes findings that survived FDR correction.

| Variable | β | SE | t | p |
|---|---|---|---|---|
| (Intercept) | 0.076 | 0.064 | 1.181 | 0.238 |
| Day 2 | 0.057 | 0.027 | 2.126 | **0.034** |
| Day 3 | 0.002 | 0.027 | 0.077 | 0.939 |
| distance | 0.007 | 0.005 | 1.567 | 0.117 |
| Day 2 × distance | –0.020 | 0.007 | –2.891 | **0.004*** |
| Day 3 × distance | –0.003 | 0.007 | –0.413 | 0.679 |

**Appendix 1—table 15.** Linear mixed-effects model results in a model of global (across-track) distance, predicting neural pattern similarity for fractals in the hippocampus.
SE = standard error. To correct for multiple comparisons, we adjusted the α-value for 8 tests.

| Variable | β | SE | t | p |
|---|---|---|---|---|
| (Intercept) | 0.014 | 0.032 | 0.432 | 0.666 |
| Day 2 | –0.005 | 0.013 | –0.412 | 0.682 |
| Day 3 | –0.003 | 0.014 | –0.192 | 0.849 |
| distance (link distance 2 > link distance 1) | –0.002 | 0.002 | –0.944 | 0.345 |
| Day 2 × distance | 0.004 | 0.003 | 1.471 | 0.141 |
| Day 3 × distance | 0.003 | 0.003 | 1.081 | 0.280 |

**Appendix 1—table 16.** Linear mixed-effects model results in an omnibus model of local (within-track) distance, predicting neural pattern similarity for landmark buildings in EC.

SE = standard error. To correct for multiple comparisons, we adjusted the α-value for 5 tests. An asterisk (*) denotes findings that survived FDR correction.

| Variable | β | SE | t | p |
| --- | --- | --- | --- | --- |
| (Intercept) | 0.053 | 0.021 | 2.521 | **0.012** |
| Day 2 | –0.057 | 0.023 | –2.503 | **0.012** |
| Day 3 | –0.040 | 0.024 | –1.662 | 0.097 |
| distance (link distance 2 > link distance 1) | 0.011 | 0.024 | 0.475 | 0.635 |
| hemisphere | –0.030 | 0.017 | –1.775 | 0.076 |
| Day 2 × distance | 0.006 | 0.033 | 0.191 | 0.849 |
| Day 3 × distance | 0.008 | 0.034 | 0.226 | 0.821 |
| Day 2 × hemisphere | 0.050 | 0.024 | 2.093 | **0.036** |
| Day 3 × hemisphere | 0.069 | 0.024 | 2.835 | **0.005*** |
| distance × hemisphere | –0.034 | 0.033 | –1.031 | 0.303 |
| Day 2 × distance × hemisphere | 0.029 | 0.047 | 0.620 | 0.535 |
| Day 3 × distance × hemisphere | 0.026 | 0.048 | 0.545 | 0.586 |

**Appendix 1—table 17.** Linear mixed-effects model results in a model of local (within-track) distance, predicting neural pattern similarity for landmark buildings in left EC.

SE = standard error. To correct for multiple comparisons, we adjusted the α-value for 10 tests.

| Variable | β | SE | t | p |
| --- | --- | --- | --- | --- |
| (Intercept) | 0.109 | 0.180 | 0.604 | 0.546 |
| Day 2 | –0.057 | 0.022 | –2.529 | **0.012** |
| Day 3 | –0.043 | 0.023 | –1.920 | 0.055 |
| distance (link distance 2 > link distance 1) | 0.011 | 0.021 | 0.508 | 0.611 |
| Day 2 × distance | 0.007 | 0.030 | 0.238 | 0.812 |
| Day 3 × distance | 0.007 | 0.031 | 0.232 | 0.816 |

**Appendix 1—table 18.** Linear mixed-effects model results in a model of local (within-track) distance, predicting neural pattern similarity for landmark buildings in right EC.

SE = standard error. To correct for multiple comparisons, we adjusted the α-value for 10 tests.

| Variable | β | SE | t | p |
| --- | --- | --- | --- | --- |
| (Intercept) | –0.410 | 0.213 | –1.921 | 0.055 |
| Day 2 | –0.001 | 0.024 | –0.060 | 0.952 |
| Day 3 | 0.029 | 0.028 | 1.056 | 0.291 |
| distance (link distance 2 > link distance 1) | –0.025 | 0.025 | –0.977 | 0.329 |
| Day 2 × distance | 0.034 | 0.036 | 0.937 | 0.349 |
| Day 3 × distance | 0.038 | 0.037 | 1.043 | 0.297 |

**Appendix 1—table 19.** Linear mixed-effects model results in an omnibus model of global (across-track) distance, predicting neural pattern similarity for landmark buildings in EC.

SE = standard error. To correct for multiple comparisons, we adjusted the α-value for 5 tests.

| Variable | β | SE | t | p |
| --- | --- | --- | --- | --- |
| (Intercept) | 0.062 | 0.026 | 2.393 | **0.017** |

*Appendix 1—table 19 Continued on next page*

*Appendix 1—table 19 Continued*

| Variable | β | SE | t | p |
|---|---|---|---|---|
| Day 2 | 0.003 | 0.036 | 0.091 | 0.928 |
| Day 3 | −0.036 | 0.035 | −1.042 | 0.297 |
| distance | 0.002 | 0.011 | 0.194 | 0.846 |
| hemisphere | −0.031 | 0.028 | −1.096 | 0.273 |
| Day 2 × distance | −0.024 | 0.015 | −1.555 | 0.120 |
| Day 3 × distance | 0.001 | 0.016 | 0.084 | 0.933 |
| Day 2 × hemisphere | 0.008 | 0.040 | 0.208 | 0.835 |
| Day 3 × hemisphere | 0.055 | 0.041 | 1.336 | 0.182 |
| distance × hemisphere | 0.001 | 0.015 | 0.069 | 0.945 |
| Day 2 × distance × hemisphere | 0.013 | 0.021 | 0.614 | 0.540 |
| Day 3 × distance × hemisphere | 0.003 | 0.022 | 0.139 | 0.889 |

**Appendix 1—table 20.** Linear mixed-effects model results in a model of global (across-track) distance, predicting neural pattern similarity for landmark buildings in EC.
SE = standard error. To correct for multiple comparisons, we adjusted the α-value for 8 tests.

| Variable | β | SE | t | p |
|---|---|---|---|---|
| (Intercept) | 0.139 | 0.102 | 1.361 | 0.174 |
| Day 2 | 0.034 | 0.042 | 0.812 | 0.417 |
| Day 3 | −0.014 | 0.041 | −0.328 | 0.743 |
| distance | 0.002 | 0.008 | 0.201 | 0.841 |
| Day 2 × distance | −0.017 | 0.011 | −1.607 | 0.108 |
| Day 3 × distance | 0.003 | 0.011 | 0.260 | 0.795 |

**Appendix 1—table 21.** Linear mixed-effects model results in a model of local (within-track) distance, predicting neural pattern similarity for fractals in left EC.
SE = standard error. To correct for multiple comparisons, we adjusted the α-value for 10 tests.

| Variable | β | SE | t | p |
|---|---|---|---|---|
| (Intercept) | 0.094 | 0.098 | 0.965 | 0.334 |
| Day 2 | −0.041 | 0.024 | −1.664 | 0.110 |
| Day 3 | −0.023 | 0.027 | −0.851 | 0.404 |
| distance | −0.015 | 0.012 | −1.266 | 0.205 |
| Day 2 × distance | −0.007 | 0.017 | −0.422 | 0.673 |
| Day 3 × distance | 0.016 | 0.017 | 0.946 | 0.344 |

**Appendix 1—table 22.** Linear mixed-effects model results in a model of local (within-track) distance, predicting neural pattern similarity for fractals in right EC.
SE = standard error. To correct for multiple comparisons, we adjusted the α-value for 10 tests.

| Variable | β | SE | t | p |
|---|---|---|---|---|
| (Intercept) | 0.116 | 0.114 | 1.021 | 0.307 |
| Day 2 | −0.016 | 0.018 | −0.919 | 0.368 |
| Day 3 | 0.020 | 0.022 | 0.906 | 0.374 |
| distance | −0.004 | 0.014 | −0.253 | 0.800 |

*Appendix 1—table 22 Continued on next page*

*Appendix 1—table 22 Continued*

| Variable | β | SE | t | p |
|---|---|---|---|---|
| Day 2 × distance | –0.015 | 0.020 | –0.769 | 0.442 |
| Day 3 × distance | 0.003 | 0.020 | 0.147 | 0.883 |

**Appendix 1—table 23.** Linear mixed-effects model results in a model of global (across-track) distance, predicting neural pattern similarity for fractals in EC.

SE = standard error. To correct for multiple comparisons, we adjusted the α-value for 8 tests.

| Variable | β | SE | t | p |
|---|---|---|---|---|
| (Intercept) | 0.121 | 0.053 | 2.293 | **0.022** |
| Day 2 | –0.034 | 0.024 | –1.415 | 0.165 |
| Day 3 | 0.007 | 0.026 | 0.284 | 0.778 |
| distance | –0.004 | 0.003 | –1.102 | 0.271 |
| Day 2 × distance | 0.008 | 0.005 | 1.655 | 0.098 |
| Day 3 × distance | 0.004 | 0.005 | 0.778 | 0.436 |

**Appendix 1—table 24.** Linear mixed-effects model results from a model predicting hippocampal pattern similarity Post Local Navigation (Day 2), with performance on subsequent Global Navigation trials (median path inefficiency for across-track trials in the first four test runs of Global Navigation on Day 2) and link distance as predictors.

SE = standard error. To correct for multiple comparisons, we adjusted the α-value for 3 tests. No results survived FDR correction.

| Variable | β | SE | t | p |
|---|---|---|---|---|
| (Intercept) | 0.038 | 0.030 | 1.264 | 0.208 |
| distance | 0.006 | 0.007 | 0.814 | 0.416 |
| path inefficiency | 0.001 | 0.001 | 1.174 | 0.241 |
| distance × path inefficiency | –0.000 | 0.000 | –1.983 | **0.048** |

**Appendix 1—table 25.** Linear mixed-effects model results from a model predicting hippocampal pattern similarity Pre-Learning (Day 1), with performance on subsequent Global Navigation trials (median path inefficiency for across-track trials in the first four test runs of Global Navigation on Day 2) and link distance as predictors.

SE = standard error. To correct for multiple comparisons, we adjusted the α-value for 3 tests.

| Variable | β | SE | t | p |
|---|---|---|---|---|
| (Intercept) | 0.073 | 0.026 | 2.786 | **0.006** |
| distance | –0.000 | 0.007 | –0.052 | 0.959 |
| path inefficiency | –0.001 | 0.001 | –1.825 | 0.070 |
| distance × path inefficiency | 0.000 | 0.000 | 1.523 | 0.128 |

**Appendix 1—table 26.** Linear mixed-effects model results from a model predicting hippocampal pattern similarity Post Global Navigation (Day 3), with performance on subsequent Global Navigation trials (median path inefficiency for across-track trials in the first four test runs of Global Navigation on Day 2) and link distance as predictors.

SE = standard error. To correct for multiple comparisons, we adjusted the α-value for 3 tests.

| Variable | β | SE | t | p |
|---|---|---|---|---|
| (Intercept) | 0.056 | 0.030 | 1.847 | 0.067 |
| distance | –0.003 | 0.007 | –0.347 | 0.729 |

| Variable | β | SE | t | p |
|---|---|---|---|---|
| path inefficiency | −0.001 | 0.001 | −1.068 | 0.288 |
| distance × path inefficiency | 0.000 | 0.000 | 1.379 | 0.168 |

**Appendix 1—table 27.** Linear mixed-effects model results from a model predicting median path inefficiency across first four test runs of Global Navigation on Day 2, with hippocampal pattern similarity Post Local Navigation (Day 2) for landmark pairs and the length of the optimal path as predictors.
SE = standard error.

| Variable | β | SE | t | p |
|---|---|---|---|---|
| (Intercept) | 75.944 | 12.743 | 5.960 | **3.07e-8** |
| hippocampal pattern similarity (LM$_A$, LM$_B$) | −41.245 | 24.163 | −1.707 | 0.088 |
| length of optimal path | 20.551 | 6.245 | 3.291 | **0.001** |
| trial type (within-track > across-track) | −0.663 | 0.183 | −3.631 | **0.0003** |
| hippocampal pattern similarity × length of optimal path | 54.618 | 43.489 | 1.256 | 0.210 |

**Appendix 1—table 28.** Linear mixed-effects model results from a model predicting median path inefficiency across first four test runs of Global Navigation on Day 2, with hippocampal pattern similarity Post Local Navigation (Day 2) for fractal pairs and the length of the optimal path as predictors.
SE = standard error.

| Variable | β | SE | t | p |
|---|---|---|---|---|
| (Intercept) | 71.855 | 12.352 | 5.817 | **3.98e-8** |
| hippocampal pattern similarity (FR$_A$, FR$_B$) | 3.026 | 27.687 | 0.109 | 0.913 |
| length of optimal path | 27.289 | 6.196 | 4.404 | **1.23e-5** |
| trial type (within-track > across-track) | −0.646 | 0.184 | −3.519 | **0.0005** |
| hippocampal pattern similarity × length of optimal path | −68.107 | 48.915 | −1.392 | 0.164 |

**Appendix 1—table 29.** Linear mixed-effects model results for a context model predicting neural pattern similarity that included vmPFC and hippocampus.
SE = standard error. To correct for multiple comparisons, we adjusted the α-value for 6 tests. No results survived FDR correction.

| Variable | β | SE | t | p |
|---|---|---|---|---|
| (Intercept) | 0.054 | 0.018 | 3.017 | **0.003** |
| Day 2 | −0.005 | 0.010 | −0.528 | 0.603 |
| Day 3 | −0.002 | 0.009 | −0.239 | 0.813 |
| context (same track > different tracks) | 0.001 | 0.002 | 0.525 | 0.600 |
| region | −0.005 | 0.002 | −2.195 | **0.028** |
| Day 2 × context | −0.004 | 0.003 | −1.156 | 0.248 |
| Day 3 × context | −0.002 | 0.003 | −0.720 | 0.471 |
| Day 2 × region | 0.005 | 0.003 | 1.761 | 0.078 |
| Day 3 × region | −0.005 | 0.003 | −1.660 | 0.097 |
| context × region | 0.002 | 0.004 | 0.386 | 0.700 |
| Day 2 × context × region | −0.000 | 0.006 | −0.008 | 0.993 |
| Day 3 × context × region | 0.005 | 0.006 | 0.845 | 0.398 |

**Appendix 1—table 30.** Linear mixed-effects model results from a context model predicting neural pattern similarity that included the hippocampus, EC, vmPFC and a visual control region.

The visual control region served as a baseline. SE = standard error. To correct for multiple comparisons, we adjusted the α-value for 18 tests. An asterisk (*) denotes findings that survived FDR correction.

| Variable | β | SE | t | p |
| --- | --- | --- | --- | --- |
| (Intercept) | 0.591 | 0.019 | 30.662 | **<2e-16** |
| Day 2 | –0.042 | 0.014 | –3.053 | **0.005** |
| Day 3 | –0.039 | 0.016 | –2.400 | **0.024** |
| context (same track > different tracks) | 0.002 | 0.004 | 0.425 | 0.670 |
| region (hippocampus) | –0.499 | 0.003 | –176.735 | **<2e-16*** |
| region (EC) | –0.513 | 0.003 | –181.929 | **<2e-16*** |
| region (vmPFC) | –0.480 | 0.003 | –144.553 | **<2e-16*** |
| Day 2 × context | –0.003 | 0.006 | –0.460 | 0.645 |
| Day 3 × context | –0.004 | 0.006 | –0.587 | 0.557 |
| Day 2 × region (hippocampus) | 0.027 | 0.004 | 7.233 | **4.74e-13*** |
| Day 3 × region (hippocampus) | 0.030 | 0.004 | 7.673 | **1.68e-14*** |
| Day 2 × region (EC) | 0.012 | 0.004 | 3.024 | **0.002*** |
| Day 3 × region (EC) | 0.039 | 0.004 | 9.957 | **<2e-16*** |
| Day 2 × region (vmPFC) | 0.031 | 0.004 | 7.029 | **2.09e-12*** |
| Day 3 × region (vmPFC) | 0.012 | 0.004 | 2.633 | **0.008*** |
| context × region (hippocampus) | –0.001 | 0.005 | –0.116 | 0.908 |
| context × region (EC) | 0.003 | 0.005 | 0.567 | 0.570 |
| context × region (vmPFC) | 0.001 | 0.006 | 0.156 | 0.876 |
| Day 2 × context × region (hippocampus) | –0.001 | 0.008 | –0.117 | 0.907 |
| Day 3 × context × region (hippocampus) | 0.001 | 0.008 | 0.182 | 0.855 |
| Day 2 × context × region (EC) | –0.010 | 0.008 | –1.349 | 0.177 |
| Day 3 × context × region (EC) | –0.006 | 0.008 | –0.780 | 0.435 |
| Day 2 × context × region (vmPFC) | –0.001 | 0.009 | –0.109 | 0.913 |
| Day 3 × context × region (vmPFC) | 0.006 | 0.009 | 0.725 | 0.468 |

**Appendix 1—table 31.** Linear mixed-effects model results for a context model predicting neural pattern similarity, fit to data in vmPFC.

SE = standard error. To correct for multiple comparisons, we adjusted the α-value for 16 tests.

| Variable | β | SE | t | p |
| --- | --- | --- | --- | --- |
| (Intercept) | 0.049 | 0.036 | 1.366 | 0.172 |
| Day 2 | –0.004 | 0.015 | –0.244 | 0.809 |
| Day 3 | –0.015 | 0.015 | –0.981 | 0.340 |
| context (same track > different tracks) | 0.003 | 0.004 | 0.748 | 0.455 |
| Day 2 × context | –0.004 | 0.005 | –0.731 | 0.465 |
| Day 3 × context | 0.002 | 0.005 | 0.463 | 0.643 |

**Appendix 1—table 32.** Linear mixed-effects model results from a distance model predicting neural pattern similarity for landmark buildings that included the hippocampus and vmPFC.

SE = standard error. To correct for multiple comparisons, we adjusted the α-value for 6 tests.

| Variable | β | SE | t | p |
|---|---|---|---|---|
| (Intercept) | 0.036 | 0.037 | 0.982 | 0.326 |
| Day 2 | 0.002 | 0.014 | 0.138 | 0.890 |
| Day 3 | −0.003 | 0.012 | −0.205 | 0.837 |
| distance | 0.004 | 0.002 | 1.853 | 0.064 |
| region | −0.005 | 0.010 | −0.544 | 0.587 |
| Day 2 × distance | −0.003 | 0.003 | −1.025 | 0.306 |
| Day 3 × distance | −0.002 | 0.003 | −0.572 | 0.567 |
| Day 2 × region | 0.022 | 0.014 | 1.543 | 0.123 |
| Day 3 × region | −0.002 | 0.014 | −0.126 | 0.900 |
| distance × region | −0.002 | 0.004 | −0.525 | 0.600 |
| Day 2 × distance × region | −0.003 | 0.005 | −0.663 | 0.508 |
| Day 3 × distance × region | 0.001 | 0.005 | 0.156 | 0.876 |

**Appendix 1—table 33.** Linear mixed-effects model results from a distance model predicting neural pattern similarity for landmark buildings that included the hippocampus, vmPFC, and a visual control region.
The visual control region served as a baseline. SE = standard error. To correct for multiple comparisons, we adjusted the α-value for 12 tests. An asterisk (*) denotes findings that survived FDR correction.

| Variable | β | SE | t | p |
|---|---|---|---|---|
| (Intercept) | 0.537 | 0.037 | 14.413 | **<2e-16** |
| Day 2 | −0.087 | 0.020 | −4.329 | **6.55e-5** |
| Day 3 | −0.059 | 0.022 | −2.712 | **0.009** |
| distance | −0.002 | 0.003 | −0.628 | 0.530 |
| region (hippocampus) | −0.506 | 0.011 | −46.889 | **<2e-16*** |
| region (vmPFC) | −0.483 | 0.013 | −38.294 | **<2e-16*** |
| Day 2 × distance | 0.008 | 0.005 | 1.655 | 0.098 |
| Day 3 × distance | −0.001 | 0.005 | −0.233 | 0.816 |
| Day 2 × region (hippocampus) | 0.077 | 0.015 | 5.102 | **3.38e-7*** |
| Day 3 × region (hippocampus) | 0.047 | 0.015 | 3.079 | **0.002*** |
| Day 2 × region (vmPFC) | 0.089 | 0.017 | 5.129 | **2.92e-7*** |
| Day 3 × region (vmPFC) | 0.028 | 0.018 | 1.549 | 0.121 |
| distance × region (hippocampus) | 0.006 | 0.004 | 1.387 | 0.165 |
| distance × region (vmPFC) | 0.005 | 0.005 | 1.073 | 0.283 |
| Day 2 × distance × region (hippocampus) | −0.010 | 0.006 | −1.821 | 0.069 |
| Day 3 × distance × region (hippocampus) | −0.000 | 0.006 | −0.041 | 0.967 |
| Day 2 × distance × region (vmPFC) | −0.015 | 0.007 | −2.296 | **0.022*** |
| Day 3 × distance × region (vmPFC) | −0.001 | 0.007 | −0.086 | 0.931 |

**Appendix 1—table 34.** Linear mixed-effects model results from a model of local (within-track) distance, predicting neural pattern similarity for landmark buildings in vmPFC.
SE = standard error. To correct for multiple comparisons, we adjusted the α-value for 10 tests.

| Variable | β | SE | t | p |
|---|---|---|---|---|
| (Intercept) | 0.144 | 0.167 | 0.866 | 0.387 |
| Day 2 | –0.005 | 0.027 | –0.174 | 0.862 |
| Day 3 | –0.024 | 0.027 | –0.867 | 0.386 |
| distance (link distance 2 > link distance 1) | –0.005 | 0.020 | –0.234 | 0.815 |
| Day 2 × distance | 0.025 | 0.028 | 0.910 | 0.363 |
| Day 3 × distance | 0.021 | 0.029 | 0.745 | 0.456 |

**Appendix 1—table 35.** Linear mixed-effects model results from the model of global (across-track) distance, predicting neural pattern similarity for landmark buildings in vmPFC.

SE = standard error. To correct for multiple comparisons, we adjusted the α-value for 8 tests.

| Variable | β | SE | t | p |
|---|---|---|---|---|
| (Intercept) | 0.055 | 0.119 | 0.462 | 0.644 |
| Day 2 | 0.043 | 0.044 | 0.993 | 0.321 |
| Day 3 | –0.001 | 0.046 | –0.012 | 0.990 |
| distance | 0.008 | 0.009 | 0.955 | 0.340 |
| Day 2 × distance | –0.013 | 0.013 | –1.071 | 0.284 |
| Day 3 × distance | –0.004 | 0.013 | –0.320 | 0.749 |

**Appendix 1—table 36.** Linear mixed-effects model results from the model of local (within-track) distance, predicting neural pattern similarity for fractals in vmPFC.

SE = standard error. To correct for multiple comparisons, we adjusted the α-value for 10 tests.

| Variable | β | SE | t | p |
|---|---|---|---|---|
| (Intercept) | –0.009 | 0.085 | –0.107 | 0.914 |
| Day 2 | 0.004 | 0.021 | 0.172 | 0.864 |
| Day 3 | –0.002 | 0.022 | –0.094 | 0.926 |
| distance | 0.009 | 0.010 | 0.865 | 0.387 |
| Day 2 × distance | –0.005 | 0.015 | –0.347 | 0.729 |
| Day 3 × distance | –0.021 | 0.015 | –1.403 | 0.161 |

**Appendix 1—table 37.** Linear mixed-effects model results from the model of global (across-track) distance, predicting neural pattern similarity for fractals in vmPFC.

SE = standard error. To correct for multiple comparisons, we adjusted the α-value for 8 tests.

| Variable | β | SE | t | p |
|---|---|---|---|---|
| (Intercept) | –0.055 | 0.058 | –0.958 | 0.338 |
| Day 2 | 0.001 | 0.023 | 0.022 | 0.982 |
| Day 3 | –0.046 | 0.024 | –1.974 | 0.053 |
| distance | 0.002 | 0.004 | 0.432 | 0.666 |
| Day 2 × distance | –0.001 | 0.006 | –0.136 | 0.892 |
| Day 3 × distance | 0.009 | 0.006 | 1.632 | 0.103 |

**Appendix 1—table 38.** Linear mixed-effects model results from a model predicting neural pattern similarity between pairs of items in vmPFC, with scan session, hippocampal pattern similarity, and pattern similarity in a visual control region (calcarine) as predictors.

SE = standard error. To correct for multiple comparisons, we adjusted the α-value for 2 tests. An asterisk (*) denotes findings that survived FDR correction.

| Variable | β | SE | t | p |
|---|---|---|---|---|
| (Intercept) | –0.007 | 0.022 | –0.331 | 0.743 |
| Day 2 | –0.006 | 0.019 | –0.298 | 0.768 |
| Day 3 | 0.004 | 0.020 | 0.199 | 0.844 |
| hippocampal pattern similarity (Item$_A$, Item$_B$) | 0.414 | 0.029 | 14.387 | **1.25e-14** |
| calcarine pattern similarity (Item$_A$, Item$_B$) | 0.108 | 0.020 | 5.465 | **4.83e-6** |
| Day 2 × hippocampal pattern similarity (Item$_A$, Item$_B$) | –0.094 | 0.016 | –5.713 | **1.11e-8** |
| Day 3 × hippocampal pattern similarity (Item$_A$, Item$_B$) | –0.076 | 0.017 | –4.355 | **1.34e-5** |
| Day 2 × calcarine pattern similarity (Item$_A$, Item$_B$) | 0.019 | 0.013 | 1.427 | 0.154 |
| Day 3 × calcarine pattern similarity (Item$_A$, Item$_B$) | –0.020 | 0.014 | –1.433 | 0.152 |

**Appendix 1—table 39.** Linear mixed-effects model results from a context model predicting neural pattern similarity, fit to data from a visual control region.
SE = standard error. To correct for multiple comparisons, we adjusted the α-value for 1<u>6</u> tests.

| Variable | β | SE | t | p |
|---|---|---|---|---|
| (Intercept) | 0.554 | 0.043 | 13.030 | **<2e-16** |
| Day 2 | –0.021 | 0.025 | –0.820 | 0.412 |
| Day 3 | –0.039 | 0.041 | –0.944 | 0.345 |
| stimulus type (landmark > fractal) | –0.024 | 0.003 | –7.009 | **2.45e-12** |
| context (same track > different tracks) | 0.001 | 0.003 | 0.245 | 0.806 |
| Day 2 × stimulus type | 0.005 | 0.005 | 1.088 | 0.277 |
| Day 3 × stimulus type | –0.002 | 0.005 | –0.407 | 0.684 |
| Day 2 × context | –0.003 | 0.005 | –0.717 | 0.474 |
| Day 3 × context | –0.002 | 0.005 | –0.416 | 0.678 |
| stimulus type × context | –0.008 | 0.007 | –1.252 | 0.211 |
| Day 2 × stimulus type × context | –0.004 | 0.009 | –0.377 | 0.706 |
| Day 3 × stimulus type × context | 0.014 | 0.010 | 1.509 | 0.131 |

**Appendix 1—table 40.** Linear mixed-effects model results from a model of local (within-track) distance predicting neural pattern similarity, fit to data from a visual control region.
SE = standard error. To correct for multiple comparisons, we adjusted the α-value for 10 tests.

| Variable | β | SE | t | p |
|---|---|---|---|---|
| (Intercept) | 0.317 | 0.135 | 2.343 | **0.019** |
| Day 2 | –0.040 | 0.030 | –1.351 | 0.177 |
| Day 3 | –0.025 | 0.044 | –0.577 | 0.564 |
| distance (link distance 2 > link distance 1) | 0.014 | 0.016 | 0.870 | 0.384 |
| Day 2 × distance | 0.007 | 0.022 | 0.304 | 0.761 |
| Day 3 × distance | –0.035 | 0.022 | –1.570 | 0.117 |

**Appendix 1—table 41.** Linear mixed-effects model results from a model of global (across-track) distance predicting neural pattern similarity, fit to data from a visual control region.
SE = standard error. To correct for multiple comparisons, we adjusted the α-value for 8 tests.

| Variable | β | SE | t | p |
|---|---|---|---|---|
| (Intercept) | 0.418 | 0.095 | 4.379 | **1.31e-5** |

*Appendix 1—table 41 Continued on next page*

*Appendix 1—table 41 Continued*

| Variable | β | SE | t | p |
|---|---|---|---|---|
| Day 2 | −0.003 | 0.040 | −0.082 | 0.935 |
| Day 3 | −0.061 | 0.051 | −1.190 | 0.234 |
| distance | −0.005 | 0.007 | −0.783 | 0.434 |
| Day 2 × distance | −0.003 | 0.010 | −0.261 | 0.794 |
| Day 3 × distance | 0.007 | 0.010 | 0.727 | 0.468 |

