## [Editor Report]

This is a carefully designed and analysed fMRI study investigating how neural representations in the hippocampus, entorhinal cortex, and ventromedial prefrontal cortex change as a function of spatial learning. These important and compelling results provide new insight into how local and global knowledge about our environment is represented. It will be of great interest to researchers studying the differentiation and integration of memories and the formation of cognitive maps.

---

## [Decision Letter]

**Decision letter after peer review:**

Thank you for submitting your article "Representational integration and differentiation in the human hippocampus following goal-directed navigation" for consideration by *eLife*. Your article has been reviewed by 3 peer reviewers, and the evaluation has been overseen by a Reviewing Editor and Chris Baker as the Senior Editor. The following individuals involved in the review of your submission have agreed to reveal their identity: Kenneth A Norman (Reviewer #1); Jacob L S Bellmund (Reviewer #3).

As you will see from the detailed comments below, the reviewers all think this study addresses an interesting and timely question with careful design and analysis all leading to a high-quality manuscript.

However, the reviewers also highlight areas where the manuscript could be improved, including additional analyses. Below, I summarize the essential revisions you will need to make, although we would like you to respond in detail to all of the comments and recommendations of the reviewers.

Essential revisions:

1) Given the exploratory nature of many of the analyses, you should provide greater clarity on the specific predictions, highlight what alternative predictions/outcomes would mean, and provide a clearer link between the conceptual prediction and analytic outcomes. In this context, you should also provide greater clarity on how multiple comparisons were applied for the different analyses.

2) Please present the results in a clear and consistent way throughout the manuscript (in particular, see comments from Reviewers 1 and 2 for details).

3) Consider an alternate analytic analysis to the median split applied to the performance data. We would recommend a mixed effects model (see Reviewer 2).

4) The reviewers were skeptical of the explanation of the results linking behavior to neural similarity (where you interpret a "flatter" pattern of similarity values across distances as reflecting the integration of landmarks into a global map). The additional behavioral and neural analyses suggested in the reviews might help you arrive at a better-substantiated account of these results.

5) Fractals and landmarks should be analyzed in a more consistent way (if you did not have an a priori reason to favor one over the other, you should always present results from both).

6) Present additional behavioral measures to more precisely examine navigational performance (see comments from Reviewer 3).

7) Reconsider the hippocampus-vmPFC "connectivity" analysis (see comments from Reviewer 2).

*Reviewer #1 (Recommendations for the authors):*

In addition to addressing the recommendations noted in the public review, here are some other points for the authors to consider:

1. In the Introduction, the authors say, "we predicted that early evidence of global map learning during local navigation would depend on the integration and predict participants' ability to subsequently navigate across the environment". In the paper, it is not clear what the basis for this prediction was: Was this an a priori prediction arising from existing results and/or theories, or did the prediction arise after the authors observed the slope across the distance 2, 3, 4 conditions post-local-learning? It is also unclear whether the authors predicted the specific direction of this relationship (I actually find it somewhat counterintuitive: I would have thought that participants who show more sensitivity to location – i.e., a sharper slope across the distance 2, 3, and 4 conditions – would show better navigation performance). Lastly, it is unclear whether the authors originally planned to operationalize "global map learning" using the slope across the 2, 3, and 4 conditions, and whether they also considered other ways of relating neural data to navigation behavior. Specifying the answers more clearly in the paper will help readers to evaluate the results.

2. Issues with results presentation: The way that the results are presented makes it very challenging to do apples-to-apples comparisons of different brain regions. For example, Figure 3 shows the "within – across context similarity" measure for EC and Figure 4 shows the "link distance 1 – link distance 2 similarity" measure for the hippocampus. Featuring different measures for different areas makes it hard to compare them. I know that the corresponding apples-to-apples measures are in various supplements but readers should not have to dig for them. If the authors had a priori reasons to feature these different dependent measures for different regions, that might justify the results presentation strategy used here, but – as things stand – it is unclear why some results were featured in the main text and others were relegated to the supplement. In the absence of an a priori justification for the present way of displaying results, I think it would be useful if the authors showed corresponding results for hippocampus and EC right next to each other in the main paper, regardless of significance.

3. Additional analyses to connect to other findings in the literature: I think it would be useful for the authors to consider whether there are additional exploratory analyses they could run to connect better to other findings in the literature. For example, many of the extant studies on MTL integration and differentiation separately investigate hippocampal subfields. If there are reasons to expect different results in different subfields, it might be worth doing this (but if the scans are not well suited for this purpose, or the authors don't think that there are good reasons to do so, that is fine). Also, the "hippocampal-vmPFC interaction" analysis is unconventional. Usually, this involves some kind of functional connectivity but here it was a second-order similarity analysis. Is there a reason that the authors took this approach? Do the authors think that their paradigm lends itself to interesting predictions relating to the more standard functional connectivity approach? Lastly, there are other ROIs that are known to be involved in spatial learning, e.g., RSC. Why did the authors focus on vmPFC instead of RSC in their analyses?

4. I think that it might be useful to show the similarity values for the 1, 2 within-track conditions alongside the 2, 3, and 4 (across-track) conditions. This may not be enough to resolve the question about whether integration is taking place, but looking at these results together (instead of presenting them in separate figures/analyses) might provide a more clear picture of what the regions are doing.

*Reviewer #2 (Recommendations for the authors):*

I have a few more suggestions to help with readability/interpretation.

1. It was difficult to keep track of the results as they related to the different ROIs, within vs. across tracks, distances, and time (related to my comment #5 above). It would be helpful to display all relevant plots in the same way throughout the manuscript, especially as the same mixed-effects models were applied (e.g., the same panels for different regions in the same order). Even if the findings are not significant, it is helpful to see the results in the same figure rather than comparing them with supplementary figures.

For example, Figure 4B shows data for same-track landmarks collapsed across link distance, and Figure 4C shows data for different-track landmarks split by link distance. This makes it difficult to directly compare these values, even though they were entered into the same model.

2. More broadly, the reporting of the results was somewhat difficult to follow. While I appreciate the importance of accounting for all factors in the same model, the order in which different factors were reported sometimes varied across models, making it difficult to keep in mind how the different models compared.

3. As I was reading the results, I made a note to suggest a distance analysis only to realize it was carried out later on (p. 16) – it would be helpful to include more signposting to motivate and help the reader anticipate this analysis.

4. Figure 5B was difficult to follow. The solid lines represent similarity prior to the global navigation task for both groups, but the dashed line only represents similarity post-global navigation for the less efficient navigators.

*Reviewer #3 (Recommendations for the authors):*

1. The path inefficiency metric conflates the effects of local and global knowledge of the environment. For example, when participants switch tracks correctly in the global condition but fail to respond precisely on the goal track, this could result in a relatively high error despite efficient global navigation performance. While the comparison to the local condition is helpful of course, the report of additional behavioral measures would be desirable in my view. Relevant measures could quantify how frequently participants (a) manage to switch tracks correctly and (b) how frequently they choose to navigate in the correct direction after switching tracks.

2. It is unclear to me why the distance-based analyses (Figure 4) are based on the landmark images only. I would expect a similar effect to be present for the fractal stimuli as participants were asked to memorize the fractal locations. Thus, I think demonstrating a similar effect for the fractals as for the landmarks would substantially strengthen the finding of a hippocampal representation of local and global distances.

3. Again related to the analysis of distance representations in the hippocampus, I am wondering why the authors chose the link distance as their distance measure. I think the paper would benefit from a justification for why this measure was chosen.

4. With respect to the formation of cognitive maps of space there is also evidence for the representation of (remembered) Euclidean distances in the hippocampus (e.g. Howard et al., J Neurosci, 2011; Deuker et al., *eLife*, 2016). I would be curious to see if the authors can detect similar representations of Euclidean distances in their data in an additional exploratory analysis.

5. A surprising and somewhat puzzling aspect of the reported results is that the global distance effect can only be detected after the local, but not after the global navigation task. Further, this effect seems to be driven by participants who take less efficient paths. I cannot quite follow the authors' interpretation in the Results section that the "acquisition of a more fully integrated global map" would lead to this pattern of results. Further, the authors state in the Discussion section that a "negative distance-related similarity function reflects restricted learning of the global map, hindering performance on the Global Task". In my view, a negative relationship between distance and representational similarity could rather point towards the formation of an integrated, map-like representation of the environment, where nearby landmarks share more similar representations than those that are separated by larger distances. I would appreciate some clarification on why the authors think such an effect would hinder task performance and why it might disappear with the "building of a more global cognitive map".

6. I am wondering about why the authors chose a median split to analyze interindividual differences. As dichotomizing a continuous variable such as navigation performance is often problematic (as discussed in detail e.g. in MacCallum et al., Psychological Methods, 2000), I would like to know whether the effect also holds when using an approach based on correlation/regression.

7. I tried to access the analysis code, but the GitHub repository referred to in the Data availability statement is currently empty.

---

## [Author Response]

Essential Revisions (for the authors):1) Given the exploratory nature of many of the analyses, you should provide greater clarity on the specific predictions, highlight what alternative predictions/outcomes would mean, and provide a clearer link between the conceptual prediction and analytic outcomes. In this context, you should also provide greater clarity on how multiple comparisons were applied for the different analyses.

We thank the Editors and Reviewers for these helpful directions. Throughout the revised manuscript, we added text that we believe more clearly links predictions with outcomes. For example:

“We hypothesized that there would be a change in hippocampal and entorhinal pattern similarity for items located on the same track vs. items located on different tracks. An increase in pattern similarity would suggest that within-track item representations are integrated, while a decrease would suggest these representations are differentiated following learning.”

Other edits made to address this concern are marked in the manuscript using track changes.

Regarding corrections for multiple comparisons, we have added the following to the Results:

“We report results from both planned and exploratory analyses. For planned analyses, we interpret a priori effects when significant at p <.05 uncorrected, but for completeness we note whether all reported effects survive FDR correction (see Methods).”

Further, we have added the following to the Methods section:

“To correct for multiple comparisons, we first computed m, a number representing the number of tests conducted for a given hypothesis multiplied by the number of ROIs we examined (including the visual control). For example, to examine learning-driven changes in pattern similarity for items located at different distances on the same track, we tested for two interactions: distance and scan (Day 2 > Day 1), and distance and scan (Day 3 > Day 1). ROIs tested were hippocampus, left and right EC, vmPFC, and the visual control region (Calcarine). m thus represented 2 interactions * 5 ROIs = 10. We then controlled for the false discovery rate (FDR) by ordering the p-values for each hypothesis test from smallest to largest (P(min)…P(max)), and checking if the following was satisfied for each ordered p-value^97^: P(i)≤ α ×im, where ∝ =0.05 Unless otherwise specified, all p-values reported were uncorrected and we interpret a priori predicted effects at this level. For completeness, we also note whether the reported effects survived FDR correction throughout the text and in the Appendix tables.”

Further, as per Reviewer 1 (General Comments 1 and 2):

– We now report the denominator *m* for each model in the Appendix tables and clearly indicate which findings survive FDR correction. For example, we added the following text to the caption for Appendix – Table 1:

“To correct for multiple comparisons, we adjusted the α value for 18 tests. An asterisk (*) denotes findings that survived FDR correction.”

– We updated text throughout the manuscript to further clarify which results survived corrections for multiple comparisons.

– We emphasize in the Discussion that a number of our findings do not survive FDR correction and encourage the reader to interpret our results with caution:

“Moreover, some of our findings do not survive correction for multiple comparisons, and thus should be interpreted with caution.”

2) Please present the results in a clear and consistent way throughout the manuscript (in particular, see comments from Reviewers 1 and 2 for details).

We are grateful for the feedback we received from several reviewers with suggestions about how to improve the clarity and consistency of the presentation of our findings. Accordingly, we revised the figures to present findings from hippocampus and EC next to each other (regardless of significance) in the main text. To be consistent in reporting our findings for the local and global distance analyses, we now report similarity values (as opposed to model contrasts) for the link distance 1 and 2 (within-track) conditions in Figure 4.

3) Consider an alternate analytic analysis to the median split applied to the performance data. We would recommend a mixed effects model (see Reviewer 2).

We greatly appreciate the reviewers’ suggestions regarding alternative analytic approaches. We removed this section of the original manuscript and ran two models to address comments related to the median split: (1) a mixed-effects model predicting neural pattern similarity Post Local Navigation, with a continuous metric of task performance (each participant’s median path inefficiency for across-track trials in the first four test runs of Global Navigation) and link distance as predictors (as suggested by Reviewer 3); and (2) a mixed-effects model relating trial-wise navigation data to pairwise similarity values for each given pair of landmarks and fractals (as suggested by Reviewer 2). We report findings for both; the following has been added to the main text:

“The variability in the observed across-track hippocampal distance effect may reflect that some participants encoded global map knowledge during Local Navigation, whereas others did not (or did so less fully; Figure 6a). To the extent that this is the case, this would predict that the distance-related hippocampal pattern similarity effect Post Local Navigation should relate to navigational efficiency at the outset of performing the Global Task. Specifically, we predicted that more efficient navigators would have a negative distance function, such that pattern similarity would be greatest for the most proximal across-track landmarks and decrease with distance. To test this hypothesis, we first ran a mixed effects model predicting neural pattern similarity Post Local Navigation (Day 2), with path inefficiency (median path inefficiency for across-track trials in the first four test runs of Global Navigation on Day 2) and link distance as predictors. Indeed, we observed a significant interaction between path inefficiency and link distance (β = -0.001 ± 0.001; t = -1.983; p = 0.048, did not survive FDR correction; d = 0.43; Appendix – Table 24), but the direction of the effect was unexpected. Participants who did well from the beginning of the Global Task showed no effect of distance in hippocampal pattern similarity, whereas less efficient navigators showed a negative slope (Figure 6b). A similar interaction between path inefficiency and link distance was not observed when the model was fit to data from Day 1 (β = 0.001 ± 0.001; t = 1.523; p = 0.128; Appendix – Figure 3a; Appendix – Table 25) or to data from Day 3 (β = 0.001 ± 0.001; t = 1.379; p = 0.168; Appendix – Figure 3b; Appendix – Table 26).

Next, we examined single-trial navigation data in relation to pairwise neural similarity in the hippocampus (for each given pair of landmarks and fractals), using mixed-effects models to predict path inefficiency for each trial across the first four test runs of the Global Navigation Task (Day 2). The models included (a) neural similarity (Post Local Navigation) for a given pair of fractals or the nearby landmarks and (b) length of the optimal path for each trial as predictors, and a regressor indicating whether the trial was a within-track or across-track trial. Models were run for landmarks and fractals separately. Here we observed trend-level evidence that hippocampal pattern similarity (Post Local Navigation) for landmark pairs predicted trial-level subsequent Global Navigation performance (β = -41.245 ± 24.162; t = -1.707; p = 0.088; Appendix – Table 27), such that greater hippocampal pattern similarity was predictive of a more efficient path (Appendix – Figure 4). The length of the optimal path (β = -0.663 ± 0.183; t = -3.631; p = 0.0003; d = 0.79) and trial type (within-track vs. across-track; β = 20.551 ± 6.245; t = 3.291; p = 0.001; d = 0.71) significantly predicted navigation performance in the landmark model. There was no interaction between hippocampal pattern similarity for landmark pairs and trial type (β = 54.618 ± 43.489; t = 1.256; p = 0.210).

The length of the optimal path (β = -0.646 ± 0.184; t = -3.519; p = 0.0005; d = 0.77) and trial type (within-track vs. across-track; β = 27.289 ± 6.196; t = 4.404; p < 1.24e^-5^, survived FDR correction; d = 0.96; Appendix – Table 28) were also significant predictors of navigation performance in the model relating pattern similarity for fractal pairs to subsequent performance on the Global Task. However, hippocampal pattern similarity for fractal pairs did not significantly predict subsequent performance (β = 3.026 ± 27.687; t = 0.109; p = 0.913).”

4) The reviewers were skeptical of the explanation of the results linking behavior to neural similarity (where you interpret a "flatter" pattern of similarity values across distances as reflecting the integration of landmarks into a global map). The additional behavioral and neural analyses suggested in the reviews might help you arrive at a better-substantiated account of these results.

Based on the additional behavioral and neural analyses suggested by reviewers, we added the following to the Discussion section:

“We hypothesized that this variability in the neural data might relate to the behavioral variability we observed on the subsequent Global Navigation Task, and explored this relationship in two ways. First, we ran a mixed effects model predicting hippocampal pattern similarity Post Local Navigation, with across-track path inefficiency at the start of Global Navigation and link distance as predictors. Here we observed a significant interaction between path inefficiency and link distance, but the effect was not in the direction we expected. Our results indicate that the interaction between distance and scan (the negative slope in the center panel of Figure 5b) was driven by participants who were *less* efficient at the outset of Global Navigation, whereas we predicted a negative slope would be more apparent in highly efficient navigators. We initially hypothesized a negative distance-related similarity function would reflect global map learning, and thus expected to observe such a function in efficient navigators at the start of the Global Task on Day 2 and across all participants on Day 3. Such a function was absent following Global Navigation (the right panel of Figure 5b).

Why do we observe a flat distance-related similarity function after global map learning? It is possible that individual differences in navigational strategy or the particular learning processes utilized by efficient navigators in the present task organized the map in such a way. It is also possible our findings reflect a change in event boundaries. Evidence from a study examining temporal context found that hippocampal pattern similarity did not differ between items located nearby vs. further apart within a temporal event^61^. Hippocampal pattern similarity differed, however, between items that crossed an event boundary, such that nearby items had increased pattern similarity compared to items located further apart in time. While this study examined temporal events, spatial event boundaries may function similarly in that during retrieval, representations of other within-event items are reinstated. The flat slope observed on Day 3 could thus be a signature of an integrated map.

Our second approach to relating variability in the neural data to behavioral variability used a model to predict path inefficiency for each trial across the first four test runs of the Global Navigation Task (Day 2). While the present study was not designed to optimize power to detect trial-level effects, here we found trend-level evidence that hippocampal pattern similarity (Post Local Navigation) for landmark pairs predicted trial-level subsequent Global Navigation performance (Figure 7), providing additional suggestive evidence for a relationship between hippocampal pattern similarity and subsequent behavior.”

5) Fractals and landmarks should be analyzed in a more consistent way (if you did not have an a priori reason to favor one over the other, you should always present results from both).

In our original submission, we included findings with respect to fractals in all analyses except for the distance analyses. In the revision, we updated the relevant sections of the manuscript to include results from fractals. No significant interactions between distance and scan were observed when local and global distance models were run for fractal stimuli in the hippocampus, EC, and vmPFC. These findings have been added throughout the main text and the Appendix with variations of the following language:

“There were no significant interactions between distance and scan when similar models were run for fractal stimuli.”

Full model results are reported in the Appendix (Appendix – Tables 12, 15, 21-23, 36-37).

In addition, based on feedback from Reviewers 2 and 3, we added the following to the discussion:

“While we expected similar results for both fractal and landmark stimuli throughout our analyses, the null findings we observed across ROIs when local and global distance models were run with fractal stimuli were not completely surprising. Fractal and landmark stimuli differ in several key ways, which we believe explain the observed pattern of findings. For example, fractals are only visible in the environment for a minority of trials. During the majority of navigation, participants must rely on the landmark buildings to guide them. Fractals serve as pointers to the goal location to which a participant must navigate on a particular trial, but fractal information may not be used during route planning. Further, fractals are not necessary for participants to learn the layout of the environment; local and global maps can be built from landmarks alone.”

6) Present additional behavioral measures to more precisely examine navigational performance (see comments from Reviewer 3).

Following the guidance from Reviewer 3, we determined whether participants (a) switched to the correct track and (b) navigated in the correct direction once switching on each Global Navigation trial at the start (first four test runs) and end (last two test runs) of the task on Day 2, and at the end of Day 3 (last two test runs).

We added the following to the Results section:

“We also visually examined participants’ routes on each across-track Global Navigation Task trial at the start (first four test runs) and end (last two test runs) of the task on Day 2, and at the end (last two test runs) of Day 3, noting whether participants (a) initially switched to the correct track on the trial and (b) navigated in the correct direction after switching. At the start of the Global Task on Day 2, participants switched to the correct track on 65.78% (SD = 21.35%) of across-track trials. Of those trials, participants navigated in the correct direction after switching 77.75% (SD = 22.52%) of the time. By the end of Day 2, those numbers increased to 75.09% (SD = 20.33%) and 85.64% (SD = 16.91%) respectively. Performance continued to improve on Day 3, with participants switching to the correct track on 78.36% (SD = 21.77%) of across-track trials during the last two runs of the Global Navigation Task and navigating in the correct direction 90.48% (SD = 10.63%) of the time after switching tracks.”

7) Reconsider the hippocampus-vmPFC "connectivity" analysis (see comments from Reviewer 2).

We could not undertake a more traditional univariate functional connectivity analysis because we did not scan during navigation. We agree with the reviewers that the null results reported for the second-order similarity analysis in the initial manuscript are confusing and difficult to interpret. Based on the reviewer comments, we replaced the second-order similarity analysis with the following:

“While context- and distance-related effects on vmPFC and hippocampal pattern similarity did not significantly differ, such effects were significant in hippocampus but not in vmPFC. To further test whether distance-related similarity structures were similar between the regions, we modeled the relationship between vmPFC pattern similarity and hippocampal pattern similarity. Specifically, we ran a linear mixed-effects model predicting pairwise similarity values in vmPFC, with scan session, pairwise similarity values in the hippocampus (averaged across hemispheres) and pairwise similarity values in a visual control region as predictors. Here, we observed significant interactions between scan session and pairwise similarity values in the hippocampus Post Local (Day 2 > Day 1 × hippocampal pattern similarity, β = -0.094 ± 0.017; t = -5.713; p < 1.12e^-8^, survived FDR correction; d = 1.25) and Global Navigation (Day 3 > Day 1× hippocampal pattern similarity, β = -0.076 ± 0.017; t = -4.355; p < 1.35e^-5^, survived FDR correction; d = 0.95), but not in the visual control region (Day 2 > Day 1 × calcarine pattern similarity, β = 0.019 ± 0.013; t = 1.427; p = 0.154; Day 3 > Day 1 × calcarine pattern similarity, β = -0.020 ± 0.014; t = -1.433; p = 0.152; Appendix – Table 38). This pattern of findings suggests that functional connectivity between the hippocampus and vmPFC weakens over time.”

Finally, we clarified that the analyses undertaken in vmPFC were exploratory. While we believe this is a legitimate avenue for exploration, we moved the entire treatment of findings from vmPFC to the Appendix. Please note that while we moved the vmPFC findings to the Appendix so as to streamline the manuscript, this move did not change our approach to correction for multiple comparisons; the region was still included in the FDR correction.

Reviewer #1 (Recommendations for the authors):Here are some other points for the authors to consider:1. In the Introduction, the authors say, "we predicted that early evidence of global map learning during local navigation would depend on the integration and predict participants' ability to subsequently navigate across the environment". In the paper, it is not clear what the basis for this prediction was: Was this an a priori prediction arising from existing results and/or theories, or did the prediction arise after the authors observed the slope across the distance 2, 3, 4 conditions post-local-learning? It is also unclear whether the authors predicted the specific direction of this relationship (I actually find it somewhat counterintuitive: I would have thought that participants who show more sensitivity to location – i.e., a sharper slope across the distance 2, 3, and 4 conditions – would show better navigation performance). Lastly, it is unclear whether the authors originally planned to operationalize "global map learning" using the slope across the 2, 3, and 4 conditions, and whether they also considered other ways of relating neural data to navigation behavior. Specifying the answers more clearly in the paper will help readers to evaluate the results.

Thank you for this recommendation. We followed it by adding the following to the Results section so as to clarify our prediction with respect to the brain-behavior analysis:

“Specifically, we predicted that more efficient navigators would have a negative distance function, such that pattern similarity would be greatest for the most proximal across-track landmarks and decrease with distance. To test this hypothesis, we first ran a mixed effects model predicting neural pattern similarity Post Local Navigation (Day 2), with path inefficiency (median path inefficiency for across-track trials in the first four test runs of Global Navigation on Day 2) and link distance as predictors. Indeed, we observed a significant interaction between path inefficiency and link distance (β = -0.001 ± 0.001; t = -1.983; p = 0.048, did not survive FDR correction; d = 0.43; Appendix – Table 24), but the direction of the effect was unexpected. Participants who did well from the beginning of the Global Task showed no effect of distance in hippocampal pattern similarity, whereas less efficient navigators showed a negative slope (Figure 6b). A similar interaction between path inefficiency and link distance was not observed when the model was fit to data from Day 1 (β = 0.001 ± 0.001; t = 1.523; p = 0.128; Appendix – Figure 3a; Appendix – Table 25) or to data from Day 3 (β = 0.001 ± 0.001; t = 1.379; p = 0.168; Appendix – Figure 3b; Appendix – Table 26).”

We found the direction of this effect to be counterintuitive as well. Additional follow-up analyses provided further suggestive evidence that across-track path inefficiency relates to trial-level hippocampal pattern similarity (see response to Essential Revisions General comment #3). We offer the following interpretation of these analyses in the Discussion:

“Our results indicate that the interaction between distance and scan (the negative slope in the center panel of Figure 5b) was driven by participants who were *less* efficient at the outset of Global Navigation, whereas we predicted a negative slope would be more apparent in highly efficient navigators. We initially hypothesized a negative distance-related similarity function would reflect global map learning, and thus expected to observe such a function in efficient navigators at the start of the Global Task on Day 2 and across all participants on Day 3. Such a function was absent following Global Navigation (the right panel of Figure 5b).

Why do we observe a flat distance-related similarity function after global map learning? It is possible that individual differences in navigational strategy or the particular learning processes utilized by efficient navigators in the present task organized the map in such a way. It is also possible our findings reflect a change in event boundaries. Evidence from a study examining temporal context found that hippocampal pattern similarity did not differ between items located nearby vs. further apart within a temporal event^61^. Hippocampal pattern similarity differed, however, between items that crossed an event boundary, such that nearby items had increased pattern similarity compared to items located further apart in time. While this study examined temporal events, spatial event boundaries may function similarly in that during retrieval, representations of other within-event items are reinstated. The flat slope observed on Day 3 could thus be a signature of an integrated map.”

2. Issues with results presentation: The way that the results are presented makes it very challenging to do apples-to-apples comparisons of different brain regions. For example, Figure 3 shows the "within – across context similarity" measure for EC and Figure 4 shows the "link distance 1 – link distance 2 similarity" measure for the hippocampus. Featuring different measures for different areas makes it hard to compare them. I know that the corresponding apples-to-apples measures are in various supplements but readers should not have to dig for them. If the authors had a priori reasons to feature these different dependent measures for different regions, that might justify the results presentation strategy used here, but – as things stand – it is unclear why some results were featured in the main text and others were relegated to the supplement. In the absence of an a priori justification for the present way of displaying results, I think it would be useful if the authors showed corresponding results for hippocampus and EC right next to each other in the main paper, regardless of significance.

We appreciate this comment and agree that the organization of the paper makes it difficult to compare results between regions. Accordingly, we revised the figures to present findings from the hippocampus and EC next to each other. Please see our above response to Essential Revisions General comment #2 for additions to the revised manuscript.

3. Additional analyses to connect to other findings in the literature: I think it would be useful for the authors to consider whether there are additional exploratory analyses they could run to connect better to other findings in the literature. For example, many of the extant studies on MTL integration and differentiation separately investigate hippocampal subfields. If there are reasons to expect different results in different subfields, it might be worth doing this (but if the scans are not well suited for this purpose, or the authors don't think that there are good reasons to do so, that is fine). Also, the "hippocampal-vmPFC interaction" analysis is unconventional. Usually, this involves some kind of functional connectivity but here it was a second-order similarity analysis. Is there a reason that the authors took this approach? Do the authors think that their paradigm lends itself to interesting predictions relating to the more standard functional connectivity approach? Lastly, there are other ROIs that are known to be involved in spatial learning, e.g., RSC. Why did the authors focus on vmPFC instead of RSC in their analyses?

The exploratory analysis in vmPFC was flagged by multiple reviewers, and we now take an alternative approach in the resubmission. Please see our response above to Essential Revisions General comment #7 for full details. Because we cannot undertake a more traditional univariate functional connectivity analysis (we did not scan during navigation, and it is thus not clear what would be modulating vmPFC activity during the fMRI task), we have moved the vmPFC section to the Appendix.

Finally, we agree that additional exploratory analyses (for example, in RSC) have merit. We are currently undertaking an analysis in RSC and hope to publish these findings in the future; however, in this work we focus on a priori hypotheses in EC and hippocampus. Unfortunately, our scanning protocol was not optimized for subfield segmentation.

4. I think that it might be useful to show the similarity values for the 1, 2 within-track conditions alongside the 2, 3, and 4 (across-track) conditions. This may not be enough to resolve the question about whether integration is taking place, but looking at these results together (instead of presenting them in separate figures/analyses) might provide a more clear picture of what the regions are doing.

Following this guidance and other related comments, we revised the figures in the main text to show similarity values for the 1 and 2 within-track conditions (Figure 4), in addition to the 2, 3, and 4 across-track conditions (Figure 5). Please see our above response to Essential Revisions General comment #2 for additions to the revised manuscript.

Reviewer #2 (Recommendations for the authors):I have a few more suggestions to help with readability/interpretation.1. It was difficult to keep track of the results as they related to the different ROIs, within vs. across tracks, distances, and time (related to my comment #5 above). It would be helpful to display all relevant plots in the same way throughout the manuscript, especially as the same mixed-effects models were applied (e.g., the same panels for different regions in the same order). Even if the findings are not significant, it is helpful to see the results in the same figure rather than comparing them with supplementary figures.For example, Figure 4B shows data for same-track landmarks collapsed across link distance, and Figure 4C shows data for different-track landmarks split by link distance. This makes it difficult to directly compare these values, even though they were entered into the same model.

We appreciate this comment and agree that the organization of the paper makes it difficult to compare results across regions. Accordingly, we revised the figures to present findings from the hippocampus and EC next to each other (regardless of significance). Please see our above response to Essential Revisions General comment #2 for additions to the revised manuscript.

2. More broadly, the reporting of the results was somewhat difficult to follow. While I appreciate the importance of accounting for all factors in the same model, the order in which different factors were reported sometimes varied across models, making it difficult to keep in mind how the different models compared.

We appreciate this guidance and revised the manuscript with a focus on increasing the clarity of our results, with an emphasis on how the reported results relate to our a priori hypotheses.

3. As I was reading the results, I made a note to suggest a distance analysis only to realize it was carried out later on (p. 16) – it would be helpful to include more signposting to motivate and help the reader anticipate this analysis.

We appreciate this feedback and added additional language to the beginning of the Results, under the section entitled “fMRI assays of learning-driven representational change”:

“we extracted voxel-level estimates of neural activity for each stimulus from EC and hippocampus regions-of-interest (ROIs; see Methods) and used pattern similarity analysis to probe whether learning resulted in representational differentiation or integration as a function of (a) the experienced relations between stimuli in the environment (e.g., same vs. different track, see Hippocampus and entorhinal cortex learn to separate the three tracks; distance within and across tracks, see The hippocampus represents both local and global distance) and (b)…”

4. Figure 5B was difficult to follow. The solid lines represent similarity prior to the global navigation task for both groups, but the dashed line only represents similarity post-global navigation for the less efficient navigators.

We removed the median-split analysis from the manuscript and the dashed line from this figure (now Figure 6B). As noted above, our analysis of the across-participant brain-behavior relationship was now conducted using a linear mixed-effects model. However, to qualitatively visualize the findings from the linear model, we retained presentation of the split-half data in Figure 6, and we revised the caption as follows:

“To qualitatively visualize the relationship between pattern similarity, link distance, and path inefficiency, we split participants into two groups – More Efficient and Less Efficient – based on their median path inefficiency on across-track trials in the first four test runs of the Global Task on Day 2. (A) Path inefficiency (%) for each across-track trial during the first four test runs of the Global Navigation Task, plotted for each participant and colored by performance group. (B) We used a linear mixed-effects model to formally test this relationship (see main text for details). The linear model revealed a significant interaction between path inefficiency and link distance, with the direction of the effect being unexpected. To qualitatively depict this effect, we plot hippocampal pattern similarity for landmarks on different tracks prior to the Global Navigation Task for the Less Efficient and More Efficient median-split data. Data are split by participants’ subsequent navigation performance as shown in (A). (Error bars denote SE of the estimates. More Efficient, n = 11; Less Efficient, n = 10).”

Reviewer #3 (Recommendations for the authors):1. The path inefficiency metric conflates the effects of local and global knowledge of the environment. For example, when participants switch tracks correctly in the global condition but fail to respond precisely on the goal track, this could result in a relatively high error despite efficient global navigation performance. While the comparison to the local condition is helpful of course, the report of additional behavioral measures would be desirable in my view. Relevant measures could quantify how frequently participants (a) manage to switch tracks correctly and (b) how frequently they choose to navigate in the correct direction after switching tracks.

We thank Dr. Bellmund for the proposed suggestions. In the revised manuscript, we determined whether participants (a) switched to the correct track and (b) navigated in the correct direction once switching on each Global Navigation trial at the start (first four test runs) and end (last two test runs) of the task on Day 2, as well as at the end of Day 3 (last two test runs). We agree that the addition of these data provides a more complete window onto task performance. Please see the above response to Essential Revisions General comment #6 for full details.

2. It is unclear to me why the distance-based analyses (Figure 4) are based on the landmark images only. I would expect a similar effect to be present for the fractal stimuli as participants were asked to memorize the fractal locations. Thus, I think demonstrating a similar effect for the fractals as for the landmarks would substantially strengthen the finding of a hippocampal representation of local and global distances.

Again, we are grateful for the feedback on how to improve the consistency of results reporting. In the revision, we now include results for fractal stimuli in the revised manuscript; please see our above response to Essential Revisions General comment #5 for full details.

3. Again related to the analysis of distance representations in the hippocampus, I am wondering why the authors chose the link distance as their distance measure. I think the paper would benefit from a justification for why this measure was chosen.

Please see our above response to Dr. Bellmund’s General Comment #3.

4. With respect to the formation of cognitive maps of space there is also evidence for the representation of (remembered) Euclidean distances in the hippocampus (e.g. Howard et al., J Neurosci, 2011; Deuker et al., eLife, 2016). I would be curious to see if the authors can detect similar representations of Euclidean distances in their data in an additional exploratory analysis.

We appreciate Dr. Bellmund’s suggestion; however, our task was not designed with hypotheses regarding Euclidean distances in mind (i.e., there are no distal cues in the environment that participants might use to orient themselves). We ran a small behavioral pilot study that asked participants to draw a map of the environment following Global Navigation, with mixed success. While some participants were able to do this, others were unable to conceptualize a top-down view of the environment. Further, a number of participants thought the tracks were circular instead of oval-shaped, which would distort their estimates of Euclidean distances. For these reasons, we ultimately decided against conducting the proposed exploratory analyses.

5. A surprising and somewhat puzzling aspect of the reported results is that the global distance effect can only be detected after the local, but not after the global navigation task. Further, this effect seems to be driven by participants who take less efficient paths. I cannot quite follow the authors' interpretation in the Results section that the "acquisition of a more fully integrated global map" would lead to this pattern of results. Further, the authors state in the Discussion section that a "negative distance-related similarity function reflects restricted learning of the global map, hindering performance on the Global Task". In my view, a negative relationship between distance and representational similarity could rather point towards the formation of an integrated, map-like representation of the environment, where nearby landmarks share more similar representations than those that are separated by larger distances. I would appreciate some clarification on why the authors think such an effect would hinder task performance and why it might disappear with the "building of a more global cognitive map".

As noted above, we appreciate Dr. Bellmund’s input here. We revised and clarified the Discussion based on reviewer comments. Please see our above response to Essential Revisions General comment #4.

6. I am wondering about why the authors chose a median split to analyze interindividual differences. As dichotomizing a continuous variable such as navigation performance is often problematic (as discussed in detail e.g. in MacCallum et al., Psychological Methods, 2000), I would like to know whether the effect also holds when using an approach based on correlation/regression.

As noted above, we are grateful for this input. Following this guidance, we replaced the median-split analysis with a mixed-effects model predicting neural pattern similarity Post Local Navigation, with a continuous metric of task performance (each participant’s median path inefficiency for across-track trials in the first four test runs of Global Navigation) and link distance as predictors. Please see our above response to Dr. Bellmund’s General Comment 5.

7. I tried to access the analysis code, but the GitHub repository referred to in the Data availability statement is currently empty.

We remedied this oversight. The analysis code is now available at https://github.com/coreyfernandez/RID. fMRI data will be uploaded to Open Neuro by the time of publication.